# Specialized post-arterial capillaries facilitate adult bone remodelling

Vishal Mohanakrishnan [1], Kishor K. Sivaraj [1], Hyun-Woo Jeong [2], Esther Bovay[1], Backialakshmi Dharmalingam [1], M. Gabriele Bixel [1], Van Vuong Dinh[1], Milena Petkova[3], Isidora Paredes Ugarte[1], Yi-Tong Kuo[1], Malarvizhi Gurusamy[1], Brian Raftrey[4,5,6,14], Nelson Tsz Long Chu[7,8], Soumyashree Das[4,5,6,15], Pamela E. Rios Coronado[4,5,6], Martin Stehling[9], Lars Sävendahl [10], Andrei S. Chagin [7,8], Taija Mäkinen [3,11,12], Kristy Red-Horse [4,5,6,13] & Ralf H. Adams [1]✉

The vasculature of the skeletal system is crucial for bone formation, homoeostasis and fracture repair, yet the diversity and specialization of bone-associated vessels remain poorly understood. Here we identify a specialized type of post-arterial capillary, termed type R, involved in bone remodelling. Type R capillaries emerge during adolescence around trabecular bone, possess a distinct morphology and molecular profile, and are associated with osteoprogenitors and bone-resorbing osteoclasts. Endothelial cell-specific overexpression of the transcription factor DACH1 in postnatal mice induces a strong increase in arteries and type R capillaries, leading to local metabolic changes and enabling trabecular bone formation in normally highly hypoxic areas of the diaphysis. Indicating potential clinical relevance of type R capillaries, these vessels respond to anti-osteoporosis treatments and emerge during ageing inside porous structures that are known to weaken compact bone. Our work outlines fundamental principles of vessel specialization in the developing, adult and ageing skeletal system.

Blood vessels form a hierarchically organized network for the transport of different cargoes, including hormones, gases, nutrients, waste products and circulating cells. Moreover, endothelial cells (ECs), forming the innermost layer of blood vessels, provide important factors acting on perivascular cell populations in a paracrine (angiocrine) fashion. In the skeletal system, angiogenic blood vessel growth is indispensable for embryonic and postnatal osteogenesis[1–3] but also fracture repair[4].

Previous work has provided some insight into the heterogeneity of bone ECs. Fms-related tyrosine kinase 4 (Flt4/VEGFR3) is a marker of sinusoidal endothelium in bone marrow (BM), whereas arterial ECs lack VEGFR3 but express Stem cell antigen-1 (Sca1/Ly6A)[5]. Specialized capillary ECs in the metaphysis, termed type H (CD31$^{high}$EMCN$^{high}$) because of high expression of the markers CD31 and endomucin (EMCN), are tightly associated with Runx2[+] and OSTERIX[+] osteoprogenitors. By contrast, sinusoidal ECs of the BM, which express lower levels of CD31 and EMCN, have been termed type L (CD31$^{low}$EMCN$^{low}$)[2]. Embryonic and early postnatal long bones contain a type-H-related, transiently existing EC subpopulation (type E) that strongly facilitates osteogenesis[6]. Fate-tracking experiments have established a hierarchy of bone ECs with type E ECs giving rise to type H cells, which, in turn, can generate arterial and type L endothelium[6]. In addition to arteries, afferent blood flow is provided by small periosteal (also termed cortical or transcortical) vessels, which cross the cortex of the diaphyseal bone shaft[7–10].

Bone loss occurs during healthy ageing but can progress to osteoporosis, a disease characterized by bone weakness, chronic pain and increased fracture risk. Type H vessels and perivascular osteoprogenitors decline strongly in ageing animals and in the ovariectomy model of osteoporosis[3,11]. Similar changes occur in ageing humans

and osteoporosis patients[12,13]. Existing interventions for the preservation of bone mass include antiresorptive drugs, such as oral bisphosphonates, which inhibit osteoclasts but cause microfractures over time[14,15]. Anabolic treatments, acting primarily on osteoblasts, include active fragments of parathyroid hormone (PTH), which requires daily administration and has adverse effects. While it is appreciated that anti-osteoporosis treatments lead to changes in the bone vasculature[16–18], the effects on specific EC populations and the resulting molecular changes remain little understood.

## Results

### Endothelial cell heterogeneity in bone

To extend previous single-cell RNA sequencing (scRNA-seq) studies (Supplementary Fig. 1) and gain better insight into endothelial heterogeneity in long bone and age-related changes, we isolated ECs from the tibia and femur of juvenile (3-week-old), adult (12-week-old) and aged (75-week-old) *Cdh5-mTnG* mice[19]. Following the depletion of haematopoietic cells and flow cytometric enrichment of ECs, scRNA-seq was performed (Fig. 1a and Extended Data Fig. 1a). Clustering of a total of 23,215 ECs in the resulting dataset indicates five major subpopulations, which are found at all stages in different relative ratios (Fig. 1b–e). BM ECs (bmECs), representing sinusoidal, type L ECs from BM[2], show high expression of markers like the regulator of G-protein signalling-4 (*Rgs4*), G-protein-coupled receptor 126 (*Gpr126*), vascular cell adhesion molecule 1 (*Vcam1*) and the transmembrane receptor stabilin-2 (*Stab2*). Another large subpopulation are metaphyseal ECs (mpECs), corresponding to type H ECs[2]. Enriched markers in this population include receptor activity modifying protein 3 (*Ramp3*), the Piezo-type mechanosensitive ion channel component 2 (*Piezo2*) and the laminin subunit β-1 (*Lamb1*) (Supplementary Table 1). Two further subpopulations represent proliferating ECs (pECs), expressing marker of proliferation Ki-67 (*Mki67*) and arterial ECs (aECs), characterized by transcripts for the tyrosine kinase Bmx (*Bmx*) and connexin-37 (*Gja4*) (Supplementary Table 1). Another distinct EC subcluster was termed remodelling endothelial cells (rECs) or type R endothelium because of its association with remodelling bone, as is shown further below. rECs show expression of the secreted factor C1q and tumour necrosis factor-related protein 9 (*C1qtnf9*), the enzyme flavin-containing dimethylaniline monooxygenase 2 (*Fmo2*) and the water-selective membrane channel aquaporin 7 (*Aqp7*) (Supplementary Table 1). Expression of the G-protein-coupled receptor for Apelin (*Aplnr*) is shared by rECs and mpECs. Furthermore, rECs express caveolin-1 (*Cav1*), a main component of caveolae, but this marker is also prominent in aECs (Fig. 1e,f). All EC subpopulations show the expected expression of VE-cadherin (*Cdh5*), a component of endothelial cell junctions, whereas transcripts for mesenchymal markers (*Pdgfra* and *Pdgfrb*) or lymphatic endothelial (*Prox1* and *Lyve1*) are low or absent (Fig. 1g). The analysis also shows that rECs express transcripts for *Emcn* in addition to *Cav1* but lack expression of *Flt4* (Fig. 1h).

Analysis of isolated sample groups shows that rECs are rare in juvenile bone relative to adult samples (Fig. 2a,b). To validate this finding, we performed immunostaining of femur cryosections with a combination of antibodies against caveolin-1 (CAV1), VEGFR3 and endomucin. This approach distinguishes CAV1⁺VEGFR3⁻EMCN⁻ aECs,

CAV1⁻VEGFR3⁺EMCN⁺ bmECs and CAV1⁺VEGFR3⁻EMCN⁺ rECs. The latter are located around adult but not juvenile trabecular bone (Fig. 2c,d). VEGFR3⁻EMCN⁺ rECs and VEGFR3⁺EMCN⁺ bmECs can be easily distinguished in the distal diaphysis and in the transition zone adjacent to the metaphysis in adult (22-week-old) and aged (55-week-old) mice (Extended Data Fig. 1a,b). VEGFR3⁻EMCN⁺ rECs emerge and expand gradually in the period between 3 weeks and 12 weeks after birth (Fig. 2d–g). In contrast to rECs, the mpECs of type H capillaries are abundant near the growth plate in 6-week-old femur but are strongly reduced at 22 or 55 weeks (Extended Data Fig. 1b). The same analysis reveals that proliferating ECs (pECs) in the metaphysis express the transcription factor *Sox11* (Extended Data Fig. 1c,d). Type L bmECs are characterized by high levels of *Cxadr* transcripts encoding the CAR cell adhesion molecule. Mucosal vascular addressin cell adhesion molecule 1 (MadCAM1) is another marker distinguishing bmECs from other subpopulations at both transcript and protein level (Extended Data Fig. 1c,e,f). These data show that postnatal bone growth involves a substantial expansion of the sinusoidal network in BM but also of type R capillaries around trabecular bone, whereas type H vessels gradually decline during the transition to adulthood (Extended Data Fig. 2a,b).

### Location and origin of rECs

VEGFR3⁻EMCN⁺ type R vessels are located in the proximity of trabecular bone and seem substantially thinner than the surrounding VEGFR3⁺EMCN⁺ sinusoidal vasculature (Fig. 2e). Injection of high-molecular-weight (2,000 kDa) tetramethylrhodamine isothiocyanate (TRITC)-labelled dextran confirms that type R vessels are perfused (Fig. 3a). Furthermore, analysis of sectioned long bones from 12-week-old *Efnb2-H2BGFP* reporter mice expressing a fusion protein of nuclear histone H2B and green fluorescent protein (GFP) under control of the gene encoding ephrin-B2, an arterial marker, shows that *Efnb2*⁺EMCN⁺ type R vessels are connected to *Efnb2*⁺EMCN⁻ arterioles and represent post-arterial capillaries (Fig. 3b). Type R capillaries are associated with the extracellular matrix, as indicated by immunostaining for the fibrillar collagens 3 and 4 (Fig. 3c,d). Indicating that type R vessels are actively expanding in juvenile and young adult bone, rECs extend filopodial protrusions and express the proliferation marker Ki-67 (Fig. 3e,f).

Next, we investigated the origin of rECs in postnatal bone by conducting genetic fate tracking with tamoxifen-inducible Cre lines. Analysis of *Bmx-CreERT2*, a transgenic line expressed in arteries[20], in combination with the *R26-mTmG* Cre reporter[21] yields the expected GFP-labelling of aECs but does not cover type R capillaries (Extended Data Fig. 2c–e). Due to VEGFR3 (*Flt4*) expression in bmECs but not in arteries or rECs, we performed fate tracking with an *Flt4-CreERT2* allele[22] (Fig. 3g). In 8-week-old animals and 48 h after 4-hydroxy-tamoxifen (4-OHT) administration, rECs display very limited GFP expression, whereas bmECs are abundantly labelled (Fig. 3g–i). Four weeks after 4-OHT at the age of 12 weeks, the proportion of GFP⁺-type R capillaries is strongly increased, indicating that rECs are derived from surrounding sinusoidal vessels (Fig. 3h–k). Apelin receptor (*Aplnr*) expression marks rECs and mpECs but is absent in CAV1⁺ arteries (Figs. 1e and 4a,b). Long-term fate tracking of recombined, GFP⁺ cells in *Aplnr-CreERT2*

**Fig. 1 | scRNA-seq analysis of bone endothelial cells. a**, Schematic overview of scRNA-seq work flow. **b**, Uniform Manifold Approximation and Projection (UMAP) visualization of total bone ECs, with colour-coded subclusters of bmECs (*n* = 12,002), mpECs (*n* = 6,307), aECs (*n* = 2,809), rECs (*n* = 1,768) and pECs (*n* = 329) with a total *n* = 23,215 cells. **c**, UMAP plot of total ECs separated by age group (juvenile = 7,973, adult = 10,721 and aged = 4,521). **d**, Bar plots showing the proportion of cells in each subcluster per age group. **e**, Feature plots show markers for subclusters of bone endothelial cells: bmECs/type L ECs expressing *Rgs4*, *Gpr126*, *Vcam1*, *Adamts5* and *Stab2*; mpECs/type H ECs expressing *Apln*, *Piezo2*, *Ramp3*, *Aplnr* and *Lamb1*; aECs expressing *Sema3g*, *Gja4*, *Fbln5*, *Vegfc* and *Bmx*; rECs/type R expressing *C1qtnf9*, *Cav1*, *Fmo2* and *Aqp7*; and

proliferating pECs expressing *Top2a*, *Neil3*, *Hist1h2ae* and *Mki67*. **f**, Heatmap showing the expression of the top four markers for each cluster. **g**, Heatmap depicting expression of *Cdh5* and the absence of BM stromal cell markers (*Pdgfra* and *Pdgfrb*) as well as absence of lymphatic EC markers (*Prox1* and *Lyve1*) in bone EC subclusters. **h**, Heatmap showing selected markers distinguishing EC subclusters. Black box highlights combination of markers allowing the identification of rECs (*Flt4⁻ Emcn⁺Cav1⁺*). Colour bars illustrate the expression level (**e**, **g**) and expression level (log₂ fold change) (**f**, **h**). Data in **b**, **d**–**h** are derived from an integrated scRNA-seq dataset, whereas data in **c** show individual samples from different age groups.

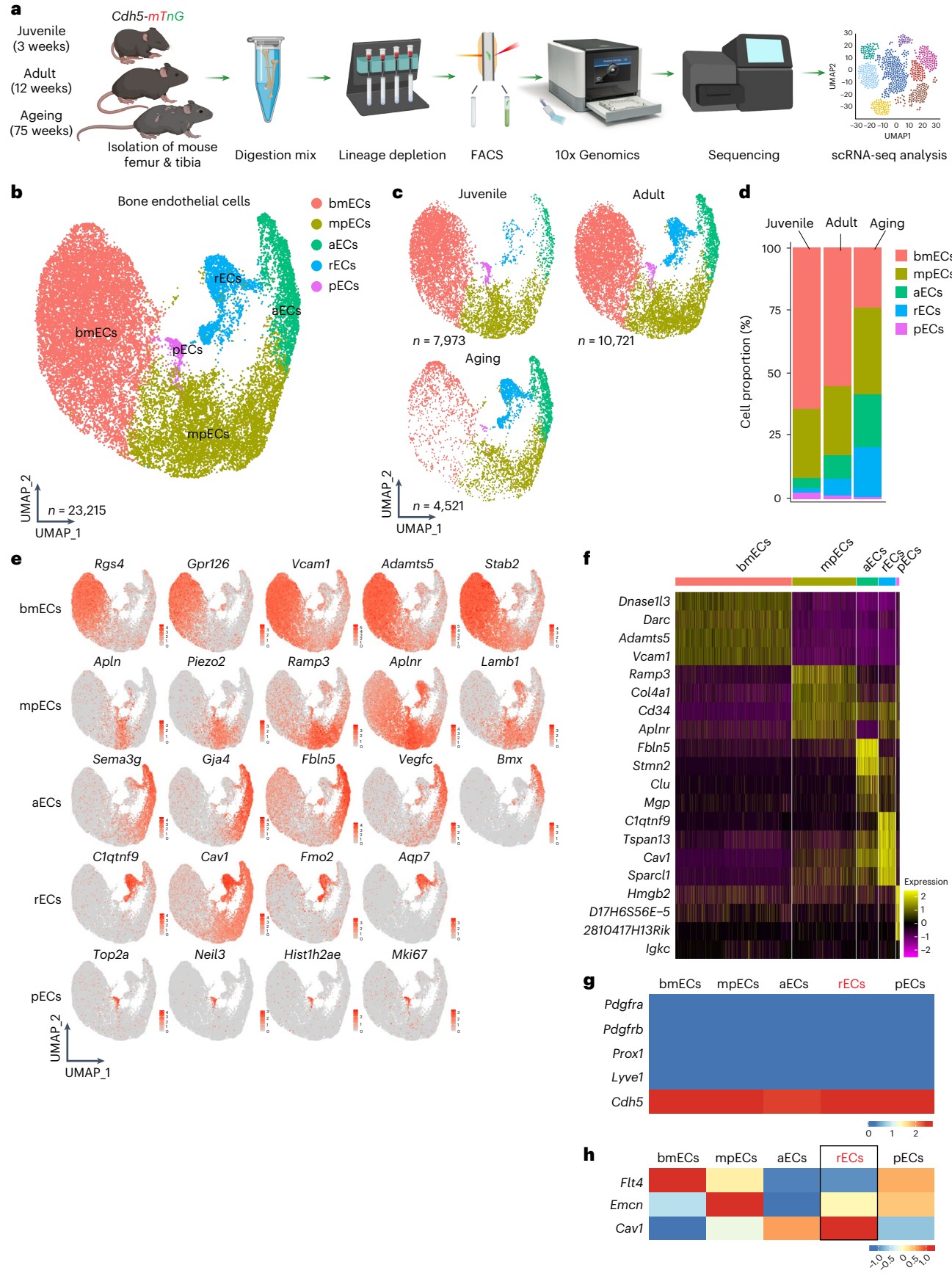

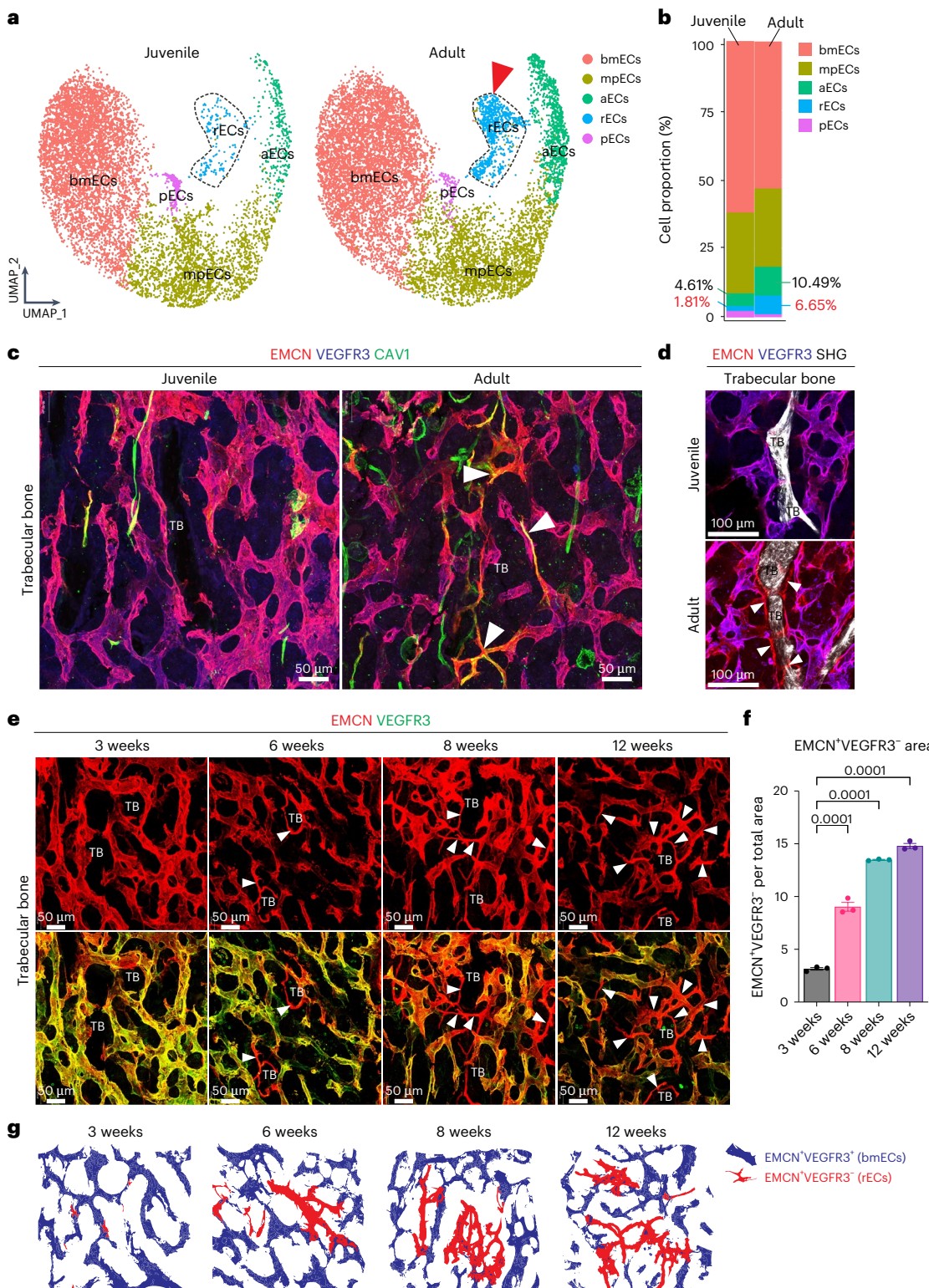

**Fig. 2 | Identification of remodelling endothelial cells. a**, UMAP plots showing colour-coded subclusters in juvenile and adult bone ECs. Dashed black line encompasses rEC clusters. Red arrowhead indicates expansion of type R ECs in adult. **b**, Bar plots showing colour-coded subclusters in juvenile and adult bone ECs. Relative representation (in %) of rECs and aECs is indicated. **c**, High-magnification images of femurs immunostained for EMCN (red), VEGFR3 (blue) and CAV1 (green) showing the emergence of EMCN⁺VEGFR3⁻CAV1⁺ (yellow) rECs (white arrowheads) around trabecular bone (TB). **d**, Two-photon microscopic image of immunostained EMCN⁺VEGFR3⁻ rECs (white arrowheads) around TB visualized by second-harmonic generation. **e**, Representative confocal images of 3-, 6-, 8- and 12-week-old femurs immunostained for EMCN (red) and VEGFR3 (green). White arrowheads indicate increasing age-dependent abundance of EMCN⁺VEGFR3⁻ rECs around TB. **f**, Quantitation of rECs (EMCN⁺VEGFR3⁻ area) showing the age-dependent increase in the 3-, 6-, 8- and 12-week-old TB relative to samples from 3 weeks. $n = 3$ mice per group. Mean ± s.e.m. $P$ values, one-way analysis of variance (ANOVA). **g**, Schematic representation of type R capillary expansion in postnatal and adult long bone. Data in **a** show individual samples from different age groups, whereas **b** is based on integrated scRNA-seq data.

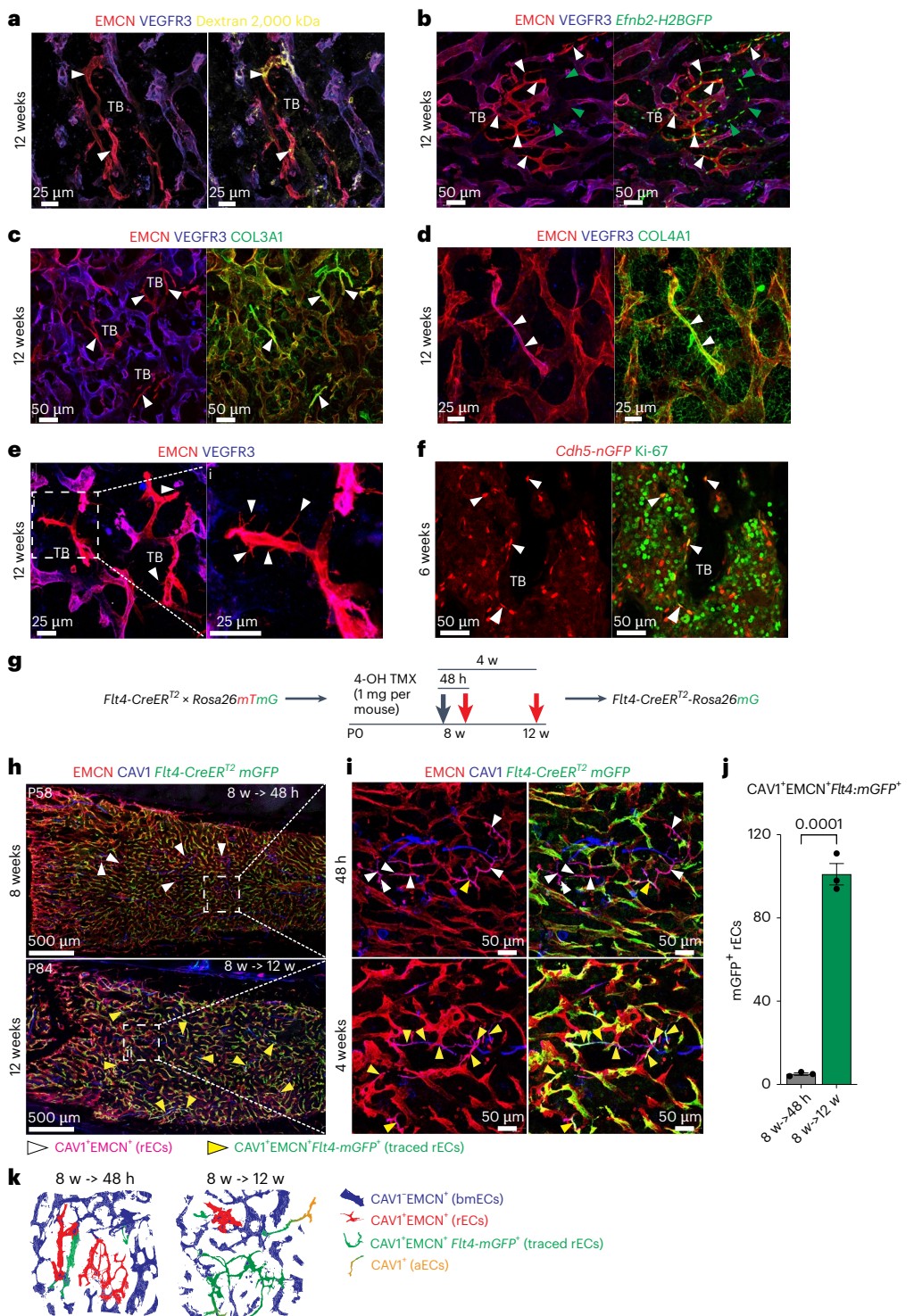

**Fig. 3 | Properties and origin of type R capillaries. a**, Perfusion of EMCN⁺VEGFR3⁻ type R capillaries (arrowheads) near TB demonstrated by injected 2,000 kDa TRITC dextran (yellow) in 12-week-old wild-type femur. **b**, Representative confocal images of 12-week-old *Efnb2-H2B-GFP* (green) femur section co-stained for EMCN (red) and VEGFR3 (blue). *Efnb2*⁺EMCN⁺ rECs (white arrowheads) are connected to *Efnb2*⁺EMCN⁻ arterioles and arteries (green arrowheads). **c,d**, Type III collagen (COL3A1) (**c**) and type IV collagen (COL4A1) (**d**) are tightly associated with EMCN⁺VEGFR3⁻ type R capillaries (arrowheads) in 12-week-old wild-type femur, whereas the surrounding sinusoidal vessels show a loose reticular fibre network. **e**, High-magnification images showing filopodia (arrowheads) extending from EMCN⁺VEGFR3⁻ rECs around 12-week-old TB. **f**, Proliferating rECs (white arrowheads) near 6-week-old TB. *Cdh5-mTnG* reporter (nGFP, red) shows EC nuclei co-stained with KI-67 (green). **g**, Scheme of genetic fate-mapping strategy. 4-OHT administration (1 mg per mouse) is indicated by black arrow and red arrows mark time points of analysis. **h,i**, Tile-scan confocal images (**h**) and higher magnification of insets (**i**) showing fate-tracked *Flt4-CreERT2 R26-mTmG* (GFP, green)-labelled ECs in femur at 8 weeks (48 h after Cre induction) and 12 weeks (4 weeks after Cre induction). White arrowheads indicate EMCN⁺CAV1⁺ rECs and yellow arrowheads mark GFP⁺ traced rECs. **j**, Quantitative analysis of GFP⁺ rECs (EMCN⁺CAV1⁺*FLT4-GFP*⁺) at 48 h and 4 weeks post-induction, respectively. *n* = 3 mice per group. Mean ± s.e.m. *P* values were obtained using an unpaired two-tailed *t*-test. **k**, Schematic illustration of genetic fate mapping of rECs in *Flt4-CreERT2 R26-mTmG* femur.

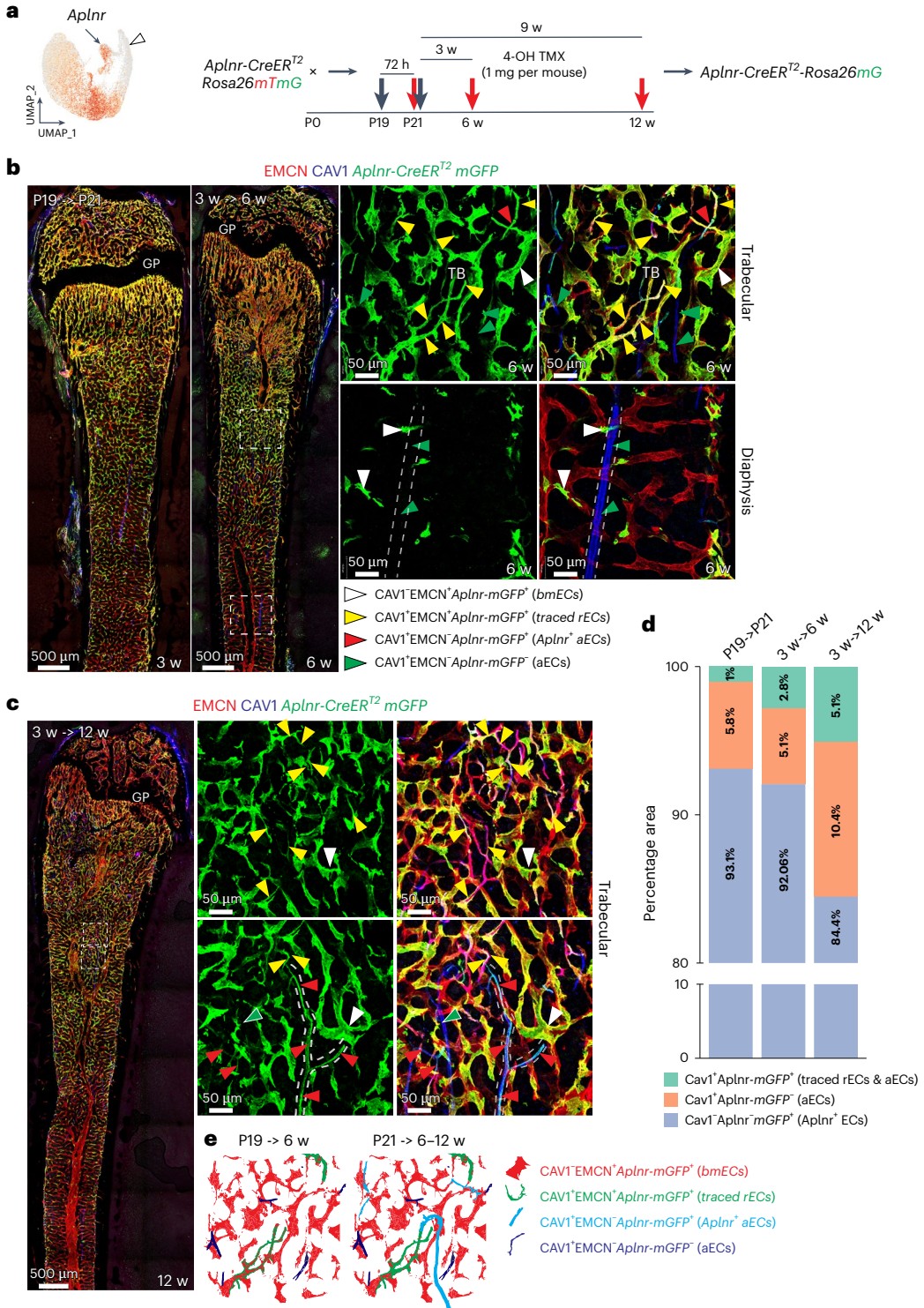

**Fig. 4 | Fate mapping of *Aplnr*+ ECs. a**, UMAP plot (derived from integrated scRNA-seq data of all age groups) illustrating *Aplnr* expression in rECs (arrow) but not aECs (arrowhead). Scheme of genetic fate-mapping strategy with *Aplnr-CreERT2* mice. 4-OHT administration (1 mg per mouse) is indicated by black arrows and red arrows mark time points of analysis. **b,c**, High-resolution confocal images and tile-scan overview images of fate-tracked *Aplnr-CreERT2 R26-mTmG* (GFP, green) ECs in femur at 3 weeks (72 h after 4-OHT administration and Cre activation), 6 weeks (3 weeks after Cre activation) (**b**) and at 12 weeks (9 weeks after Cre activation) (**c**). Small panels show higher magnifications of TB and diaphyseal area. White arrowheads mark GFP+EMCN+ bmECs, yellow arrowheads GFP+CAV1+EMCN+ rECs, green arrowheads GFP−CAV1+ aECs and red arrowheads fate-tracked GFP+CAV1+EMCN− ECs inside arteries. White dashed lines in **b** and **c** indicate arteries and arterioles. **d**, Bar plots showing the proportion of area covered by CAV1+GFP+ rECs and aECs, CAV1+GFP− aECs and CAV1−GFP+ ECs at P21, 6 weeks and 12 weeks post-induction. **e**, Schematic illustration of genetic fate mapping of rECs in *Aplnr-CreERT2 R26-mTmG* femur.

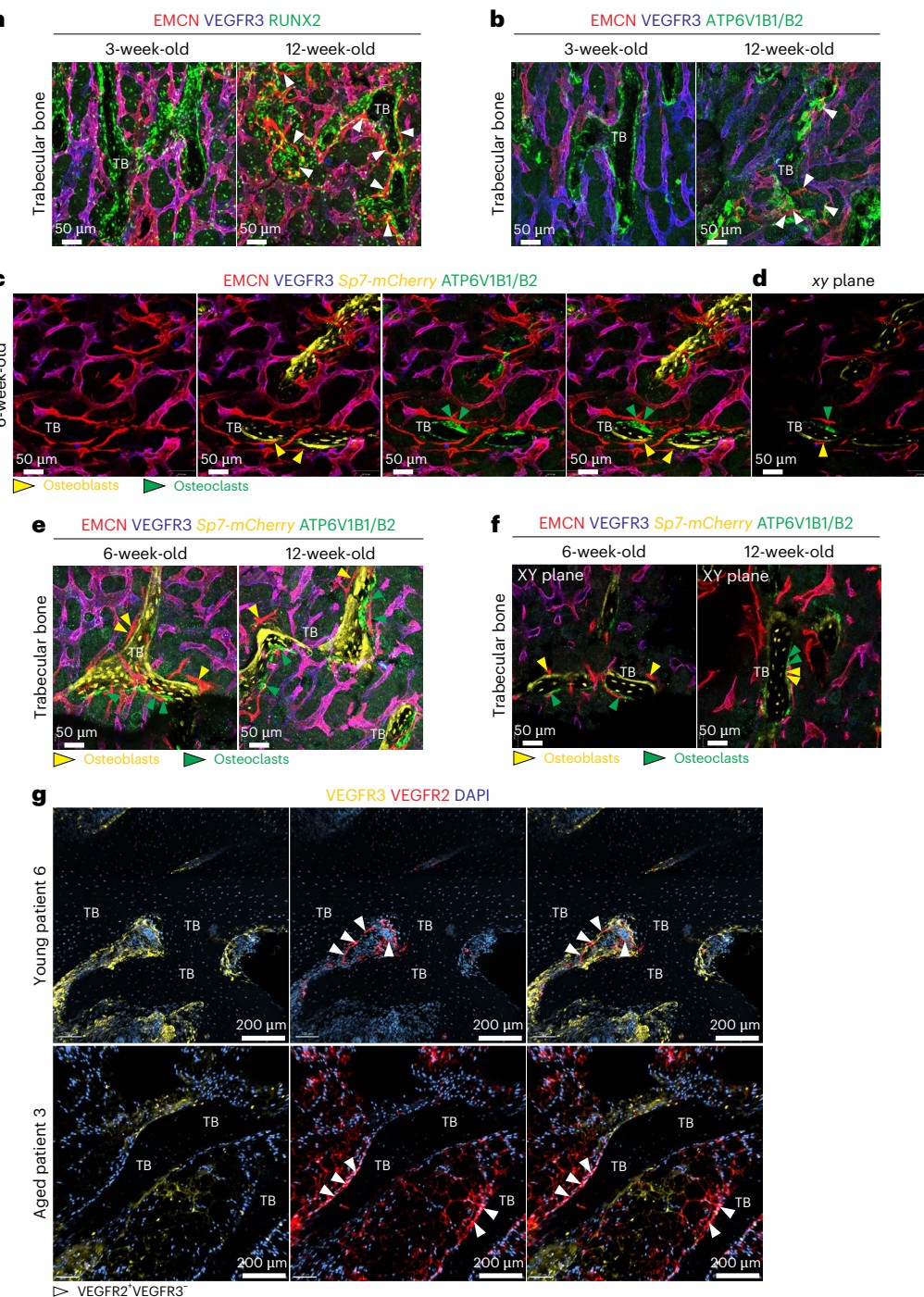

**Fig. 5 | Coupling of rECs and remodelling bone. a,b**, High-resolution confocal images showing EMCN⁺VEGFR3⁻ rECs (arrowheads) around TB in relation to RUNX2⁺ osteoprogenitors (green) (**a**) and ATP6V1B1/B2⁺ osteoclasts (green) (**b**). **c,d**, Maximum-intensity projection showing *Sp7-mCherry*⁺ osteoblasts (yellow) and ATP6V1B1/B2⁺ osteoclasts (green) in in relation to EMCN⁺VEGFR3⁻ rECs around 6-week-old femoral TB (**c**). Single *xy* plane (**d**). Green arrowheads mark rECs near osteoclasts and yellow arrowheads near osteoblasts. **e,f**, Distribution of EMCN⁺VEGFR3⁻ type R vessels in relation to *Sp7-mCherry*⁺ osteoblasts (yellow arrowheads) and ATP6V1B1/B2⁺ osteoclasts (green arrowheads) around 6-week-old and 12-week-old TB (**e**). Isolated *xy* plane (**f**). **g**, Confocal images showing VEGFR3⁻VEGFR2⁺ ECs (white arrowheads) around TB in young and aged patient samples. VEGFR3 (yellow), VEGFR2 (red) and nuclei (DAPI, blue). DAPI, 4,6-diamidino-2-phenylindole.

(ref. 23) mice shows that rECs contribute to CAV1⁺ arteries (Fig. 4a–d). Trajectory analysis of cells in our scRNA-seq data and pseudo-time analysis of marker genes is consistent with the conversion of *Flt4*⁺ bmECs into *Aplnr*⁺ rECs and also shows *C1qtnf9*⁺ rECs in proximity of *Bmx*⁺aECs (Extended Data Fig. 3a–d). The sum of this data defines major bone EC subpopulations with distinct molecular profiles (Extended Data Fig. 3e).

**Type R capillaries are associated with remodelling bone**

Immunostaining shows that type R capillaries are located around trabecular bone in adult (12-week-old) but not in 3-week-old femur (Fig. 5a,b and Extended Data Fig. 4a). Notably, type R vessels are positioned near RUNX2⁺ and OSTERIX⁺/*Sp7-mCherry*⁺ osteoprogenitor cells but also close to mature osteoclasts expressing vacuolar ATPase (Fig. 5a–f and Extended Data Fig. 4b,c). Double-fluorescence bone

labelling by consecutive injection of Calcein green and Alizarin Red supports that trabecular bone is actively remodelling in young adult mice, suggesting that type R vessels might participate in this process (Extended Data Fig. 4d).

Examination of human biopsy samples from adolescent patients shows that type R-like VEGFR2$^+$VEGFR3$^-$ ECs are located near the bone surface and surrounded by sinusoidal VEGFR2$^+$VEGFR3$^+$ vessels (Fig. 5g and Extended Data Fig. 5a). VEGFR2$^+$VEGFR3$^-$ ECs are also visible near trabecular bone in elderly patients, despite a different appearance of the adjacent, adipocyte-rich marrow (Fig. 5g and Extended Data Fig. 5b).

Sexual dimorphism is very pronounced in the skeletal system and loss of trabecular bone during adult life and ageing is much greater in female mice[24]. Immunostaining of EMCN$^+$VEGFR3$^-$ type R vessels in 12-week-old adult bone reveals a higher abundance of type R vessels in males relative to females, which correlates with trabecular bone abundance (Extended Data Fig. 5c–e). Thus, type R vessel abundance might be linked to the sexual dimorphism of bone.

### Loss of *Dach1* affects type R vessels and trabecular bone
scRNA-seq analysis shows that transcripts encoding the transcription factor DACH1 (dachshund family transcription factor 1), a regulator of coronary artery growth[25,26], are sparsely expressed in many organs (Extended Data Fig. 6a). In contrast, *Dach1* expression is enriched in rECs, which is confirmed by immunostaining of femur sections (Fig. 6a,b). We generated EC-specific *Dach1* loss-of-function mutants by combining the *Cdh5-CreERT2* transgene[27] with loxP-flanked (conditional) *Dach1* alleles[25]. Following postnatal administration of tamoxifen, analysis of *Dach1*$^{i\Delta EC}$ mutants at 12 weeks shows a reduction of the EMCN$^+$ total area, CAV1$^+$ arteries as well as VEGFR3$^-$EMCN$^+$ and CAV1$^+$EMCN$^+$ type R capillaries (Fig. 6c–e). The density of vessels in retina and intestine remains unaffected (Extended Data Fig. 6b–e). *Dach1*$^{i\Delta EC}$ mutant intestine, however, shows a reduction of villus size and therefore of total vascular area per villus (Extended Data Fig. 6d,e).

Immunostaining of bone sections confirms that EC-specific loss of *Dach1* leads to a strong reduction of type R capillaries. Residual *Dach1*$^{i\Delta EC}$ VEGFR3$^-$EMCN$^+$ vascular structures resemble sinusoidal vessels and lack the slender morphology of normal type R capillaries (Fig. 6d–f). *Dach1* loss of function also leads to a reduction of type H capillaries in proximity of the growth plate (Fig. 6f). Analysis of trabecular bone by micro-computed tomography (μCT) shows a decrease in mutant bone volume relative to tissue volume, which is much more pronounced in *Dach1*$^{i\Delta EC}$ males. Trabecular bone thickness is only reduced in male but not in female mutants relative to sex-matched littermate controls (Fig. 6g,h). Bone loss in *Dach1*$^{i\Delta EC}$ males is accompanied by a shortening of the region containing trabecular bone, an increase in the number of vATPase$^+$ osteoclasts and reduction of OSX$^+$ cells. However, connectivity density and cortical thickness remain unaffected relative to control (Extended Data Fig. 6f–i). Consistent with a role of DACH1 in artery and type R vessel specification, siRNA-mediated knockdown of the transcription factor in cultured human umbilical vein endothelial cells (HUVECs) reduces the expression of multiple arterial markers, including the Notch ligand Delta-like 4 (*DLL4*) (Extended Data Fig. 6j,k).

### DACH1 overexpression expands type R vessels and arteries
Next, we generated gain-of-function mutants (*Dach1*$^{OE}$) overexpressing DACH1 in ECs under control of the *Aplnr-CreERT2* line[23,26] (Fig. 7a). Following postnatal tamoxifen administration, *Dach1*$^{OE}$ femurs show a notable expansion of CAV1$^+$ arteries and type R vessels at postnatal day (P) 30 (Fig. 7b–d). *Dach1*-induced premature formation of type R vessels is most prominent in the distal diaphysis, a region that is normally devoid of trabecular bone (Fig. 7b,c). Analysis of mineralized bone by μCT shows an increase in trabecular bone number, volume relative to tissue volume and connectivity density, changes that are most obvious for the *Dach1*$^{OE}$ distal diaphysis. In contrast, cortical bone thickness is not significantly altered (Fig. 7e,f). Immunostaining confirms the expansion of OSX$^+$ cells associated with trabecular bone in the *Dach1*$^{OE}$ distal diaphysis. The area covered by vATPase$^+$ osteoclasts is also increased, indicating active bone turnover (Extended Data Fig. 7a–c). Double-fluorescence bone labelling by consecutive injection of Calcein green and Alizarin Red shows increased trabecular mineral apposition rates (MARs) and bone formation rates in *Dach1*$^{OE}$ mutants relative to control littermates, whereas MAR of the compact bone remains unchanged (Extended Data Fig. 7d–f). These results demonstrate that *Dach1* overexpression in *Aplnr*$^+$ ECs induces the expansion of arteries and type R capillaries, leading to increased trabecular bone formation.

To gain deeper insight into the alterations induced in *Dach1*$^{OE}$ mutants, we performed scRNA-seq analysis of non-haematopoietic cells from long bone at P30 (Fig. 7g). *Dach1* overexpression in ECs increases the fraction of osteoblasts, whereas metaphyseal mesenchymal stromal cells (mpMSCs), which are similar to previously reported osteogenic Osteo-CAR/OLC cells[28,29], are not substantially changed (Fig. 7h–k). Diaphyseal mesenchymal stromal cells (dpMSCs), which are sinusoidal vessel-associated bone mesenchymal stromal cells (BMSCs) similar to the previously reported Adipo-CAR/Lepr-MSC cells[28,29], are reduced (Fig. 7h–k). *Dach1*$^{OE}$ BMSCs as a whole show upregulation of transcription factors associated with osteogenesis, such as OSTERIX (encoded by the *Sp7* gene) and MEF2C (*Mef2c*), whereas PPARG (peroxisome proliferator activated receptor-γ, *Pparg*) and other regulators of adipogenesis are downregulated (Fig. 7l).

### Metabolic and angiocrine roles of type R vessels
Previous work has established that large parts of BM are highly hypoxic with the exception of the metaphysis and the endosteal region at the inner cortical bone surface[2,30]. Low oxygen tension and hypoxia-inducible factor (HIF) signalling are important for the localization and function of haematopoietic stem cells but also regulate osteoclast properties and their resorptive activity[31–33]. In addition, appropriate oxygenation is important for osteogenesis, as both low and high levels of oxygen can inhibit this process[34–36]. Type R vessel formation during postnatal development coincides with changes in tissue oxygenation. Hypoxia-inducible factor 1α (HIF1α) immunostaining in sections shows strong signals in 6-week-old murine femoral trabecular bone where it partially overlaps with *Sp7-mCherry* expression. In the equivalent region of 12-week-old mice, type R vessels are present and HIF1α staining is strongly diminished (Extended Data Fig. 7g). Further arguing for a role of type R post-arterial capillaries in local

**Fig. 6 | Loss of *Dach1* reduces rEC number and trabecular bone. a**, UMAP plot (integrated scRNA-seq dataset of all age groups) showing *Dach1* expression in rECs (arrow). Colour bar illustrates the expression level. **b**, Representative confocal images of DACH1 immunostaining (green) in 6-week-old *Cdh5-mTnG* (red) reporter femur. DACH1$^+$ rECs near TB (green) are marked by white arrowheads. **c**, Scheme of tamoxifen-induced EC-specific *Dach1* inactivation. **d**, Tile-scan confocal images of EMCN$^+$VEGFR3$^-$CAV1$^+$ rECs (white arrowheads) and EMCN$^-$VEGFR3$^-$CAV1$^+$ aECs (yellow arrowheads) in 12-week-old *Dach1*$^{i\Delta EC}$ loss-of-function and littermate control femur. Growth plate (GP) is indicated. **e**, Quantitation of EMCN$^+$ vessel density, EMCN$^+$VEGFR3$^-$ vessel density, and number of CAV1$^+$ arteries in 12-week-old *Dach1*$^{i\Delta EC}$ and control femur (*n* = 3–4 female mice per group). Mean ± s.e.m. *P* values, unpaired two-tailed *t*-test. Emcn$^+$ vessel density plotted with Welch's correction. **f**, High-magnification images of metaphysis near GP (left) and of EMCN$^+$VEGFR3$^-$ vessels (white arrowheads; right) around TB in 12-week-old male *Dach1*$^{i\Delta EC}$ and control femurs. White dashed lines in **d** and **f** indicate type H area. **g**, Representative 3D reconstruction of μCT measurements of 12-week-old *Dach1*$^{i\Delta EC}$ and control femoral bone. Dashed yellow lines indicate area analysed. **h**, Quantitation shows relative bone volume, represented as bone volume/tissue volume (BV/TV) and trabecular thickness (per mm) (*n* = 3–4 female mice per group and *n* = 3 male mice per group). Mean ± s.e.m. *P* values were plotted using an unpaired two-tailed *t*-test.

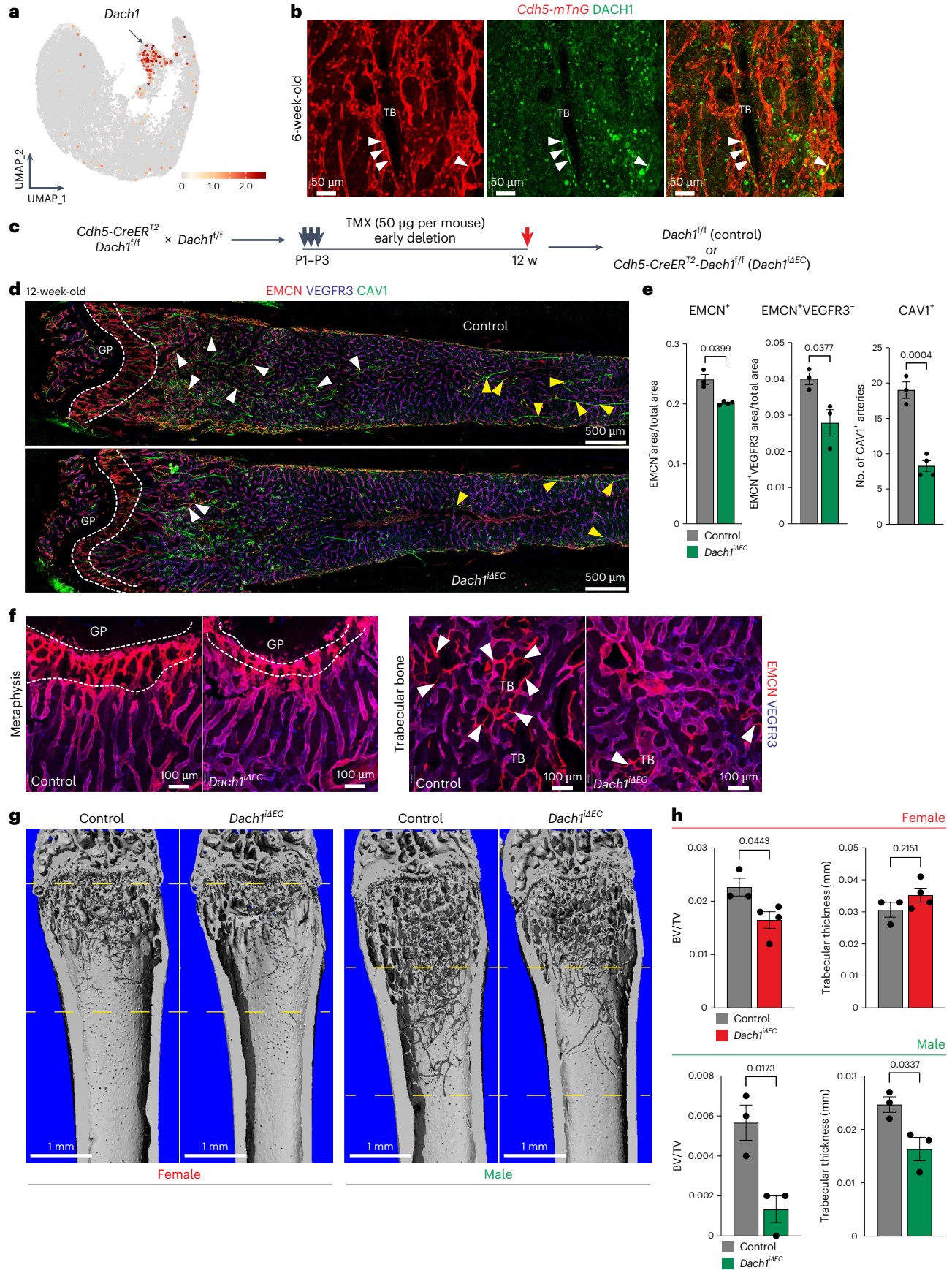

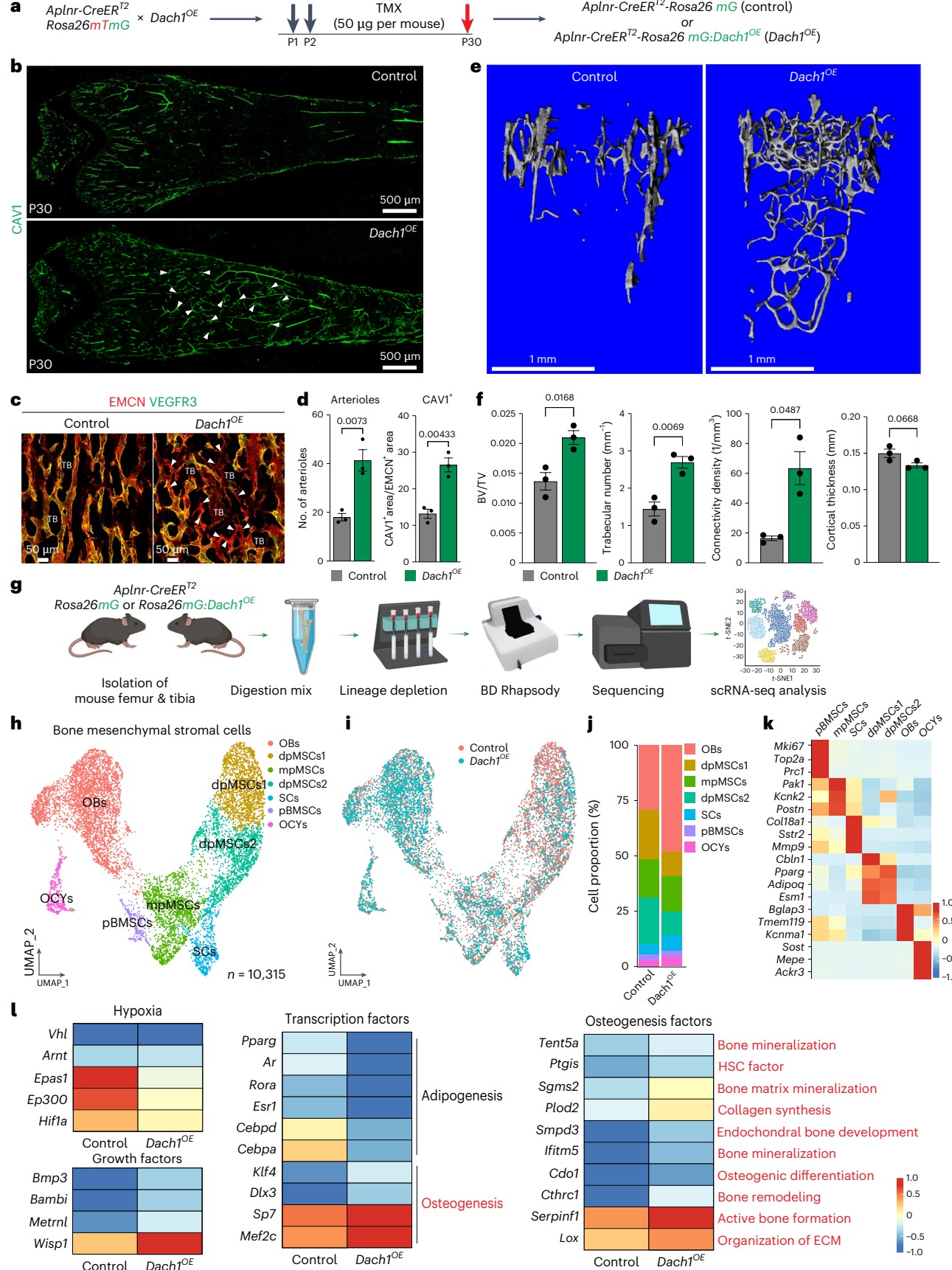

**Fig. 7 | EC-specific Dach1 overexpression enhances trabecular bone formation. a**, Schematic representation of tamoxifen-inducible *Dach1* overexpression in *Aplnr*⁺ ECs. **b**, Tile-scan confocal images showing expansion of CAV1 (green, white arrowheads) immunostained arteries and type R capillaries in the Dach1 gain-of-function (*Dach1*^OE) femur relative to littermate control at postnatal day 30 (P30). **c**, High-magnification confocal images of EMCN (red) and VEGFR3 (green) immunostained femurs showing the expansion of type R capillaries near TB in *Dach1*^OE mutants relative to littermate control. White arrowheads indicate type R capillaries. **d**, Quantitation of arteries/arterioles and CAV1⁺ area in control and *Dach1*^OE mutants. *n* = 3 mice per group. Mean ± s.e.m. *P* values were obtained by unpaired two-tailed *t*-test. **e**, Representative three-dimensional (3D) reconstruction of μCT measurements of P30 *Dach1*^OE and control femoral TB. **f**, Quantitation of parameters: bone volume/tissue volume (BV/TV), trabecular number represented by number of trabeculae per millimetre, connectivity density (connectivity density per cubic millimetre) and cortical

thickness (mm). *n* = 3 mice per group. Mean ± s.e.m. *P* values were obtained by an unpaired two-tailed *t*-test. **g**, Schematic overview of scRNA-seq workflow for non-haematopoietic cells from *Dach1*^OE and littermate control long bone. **h,i**, UMAP plots of BMSCs (*n* = 10,315) with colour-coded subclusters, namely osteoblasts (OBs), osteocytes (OCYs), septoclasts (SCs), proliferating BMSCs (pBMSCs), mpMSCs and dpMSCs (dpMSCs1 and dpMSCs2) (**h**). UMAP distribution of *Dach1*^OE and control BMSCs, as indicated by colour (**i**). **j**, Bar plots showing proportion of cells in *Dach1*^OE and control BMSC subclusters. **k**, Heatmap showing the top three marker genes for each BMSC subcluster. **l**, Heatmap illustrating differentially expressed genes in *Dach1*^OE and control BMSCs related to hypoxia, growth factors, adipogenic and osteogenic transcription factors, and regulators of osteogenesis. Texts in red are the areas of interest. Colour bars in **k** and **l** illustrate the expression level (log₂ fold change). Data in **h–l** are derived from an integrated scRNA-seq dataset of two conditions.

oxygenation, immunostaining for the hypoxia-controlled gene products glucose-6-phosphate Isomerase (GPI), ecto-5′-nucleotidase (CD73) and haem oxygenase-1 (Hmox1) is higher in and around trabecular bone in juvenile compared with adult femur (Extended Data Fig. 7h).

Arguing that tissue oxygenation is increased after EC-specific *Dach1* overexpression, *Hif1a*, *Epas1* (HIF2α) and *Ep300*, which encodes the p300 histone acetyltransferase and transactivator of HIF signalling[37,38], are downregulated in *Dach1*^OE BMSCs relative to control (Fig. 7l). Furthermore, transcripts associated with osteogenic differentiation, bone development, extracellular matrix and bone mineralization are increased in BMSCs after EC-specific *Dach1* overexpression (Fig. 7l).

The analysis of EC subpopulations in *Dach1*^OE mutants and littermate controls reveals the same clusters seen in our data of FACS-isolated bone ECs, encompassing metaphyseal mpECs (type H), diaphyseal bmECs (type L), proliferating pECs, arterial aECs and type R rECs (Extended Data Fig. 7i–m and Fig. 1a–c). Consistent with the observed increase in CAV1⁺ vessels (Fig. 7b), *Dach1*⁺ ECs are strongly enriched in the *Dach1*^OE mutant aEC and rEC clusters. In addition, a large fraction of *Dach1*⁺ cells is retained in the bmEC subcluster, which might reflect incomplete or ongoing conversion into rECs/aECs (Extended Data Fig. 7l). Alternatively, not all bmECs might be able to become arteries and post-arterial capillaries.

Our scRNA-seq data shows that *Dach1* overexpression lowers the expression of *Hif1a* and *Angptl4* (encoding angiopoietin-like protein 4), which is another hypoxia-inducible gene[39], in *Dach1*^OE ECs. Conversely, genes related to better tissue oxygenation, namely *Egfl7* (epidermal growth factor-like domain 7) and *Eln* (elastin), are upregulated in *Dach1*^OE ECs (Extended Data Fig. 7n). Likewise, *Dach1*^OE femurs show a profound reduction of HIF1α immunostaining around trabecular bone but also in the adjacent BM and metaphysis (Extended Data Fig. 7o). Expression of several bmEC markers, such as *Stab2*, *Flt4* and *Gpr182*, is decreased in *Dach1*^OE ECs, whereas arterial markers, such as transcripts for the transcription factor Sox17, the receptor Notch4 and the ligand Delta-like 4 (*Dll4*), are increased (Extended Data Fig. 8a,b). Increased Dll4 expression in the *Dach1*^OE vasculature is confirmed

by immunostaining (Extended Data Fig. 8c). Conversely, EC-specific deletion of *Dll4* following postnatal administration of tamoxifen at 6 weeks, a period of active type R vessel expansion, leads to profound reduction of EMCN⁺VEGFR3⁻ type R vessels and CAV1⁺ arteries at 12 weeks (Extended Data Fig. 8d–g).

We also investigated whether rECs might control bone turnover through the release of paracrine acting molecules. The analysis of our scRNA-seq data indicated high rEC expression of secreted factors associated with osteoclast or osteoblast function, such as the chemokine Cxcl12 (ref. 40), endothelin 1 (ref. 41) or bone morphogenetic protein 6 (BMP6)[42], but also of other molecules that are less established as regulators of osteogenesis (Extended Data Fig. 8h). We tested semaphorin 7A (*Sema7a*), platelet-derived growth factor D (*Pdgfd*), neurotrophin 3 (*Ntf3*) and complement C1q and tumour necrosis factor-related protein 9 (*C1qtnf9*) in cell culture. Treatment of human MSCs with recombinant PDGFD or SEMA7a enhances osteogenic differentiation, as indicated by increased Alizarin Red-positive calcified nodules (Extended Data Fig. 8i,j). Conversely, the differentiation of BM-derived cells into osteoclasts is improved by SEMA7a, NTF3 and PDGFD (Extended Data Fig. 8k,l). Based on these data, type R vessels might influence bone remodelling through secreted factors in addition to metabolic regulation of perivascular cell populations.

## Type R vessels in therapeutic and age-related remodelling

Our previous work has shown that the vasculature of long bones undergoes striking changes during adulthood and ageing[2]. Analysis by scRNA-seq of ECs sorted from aged (75-week-old) long bone reveals that rECs are increased relative to adult samples (Fig. 8a,b). Immunostaining shows that trabecular bone at 75 weeks remains surrounded by EMCN⁺VEGFR3⁻ type R vessels in close association with RUNX2⁺ osteoprogenitors and vATPase⁺ osteoclasts (Fig. 8c). Thinning of cortical bone and increased cortical porosity are hallmarks of bone ageing both in humans and mice, which predominantly affects females[43,44]. Notably, cortical bone pores are associated with vascular growth (Fig. 8d–f,h). The vessels inside porous cortical bone exhibit hallmarks of type R capillaries, namely expression of EMCN but not

**Fig. 8 | Age-associated changes in bone remodelling and regeneration. a**, UMAP visualization of bone ECs from 75-week-old bone with colour-coded subclusters. **b**, Bar plot showing the proportion of cells from each EC subcluster in adult and aged bone. Percentage (%) differences are indicated for rECs and aECs. Data are representative of an individual sample (Methods) (**a**) and integrated scRNA-seq data from all age groups (**b**). **c,d**, High-magnification confocal images of TB (**c**) and compact bone (CB) (**d**) immunostained for EMCN (red), VEGFR3 (blue) RUNX2 (yellow arrowheads, marking osteoprogenitors) and ATP6V1B1B2 (green arrowheads, marking osteoclasts). **e**, High-magnification two-photon microscopy images of CB immunostained for EMCN (red) and VEGFR3 (blue) together with second-harmonic generation (white) showing the changes during ageing. **f**, Confocal tile-scan images showing EMCN⁺VEGFR3⁻

rECs in untreated 12-week-old and 75-week-old mice femur and in response to treatment with alendronate (ALN) or PTH, as indicated. Note expansion of EMCN⁺ VEGFR3⁻ vessels in CB in response to ageing or treatments (arrowheads). **g**, Representative confocal images showing EMCN⁺VEGFR3⁻ rECs and OSTERIX (green) immunostaining in 75-week-old control, ALN-treated or PTH-treated CB (white arrowheads, marking type R capillaries during ageing). White dashed lines in **d–g** indicate compact bone (CB) area. **h,i**, Quantitation of EMCN⁺CAV1⁺ vessel density across age groups (**h**), EMCN⁺VEGFR3⁻ vessel density in 75-week-old control, ALN-treated or PTH-treated groups, and number of OSTERIX⁺ cells in CB after ALN and PTH treatment compared with respective controls (**i**). *n* = 3 mice per group. Mean ± s.e.m. *P* values were obtained using ordinary ANOVA.

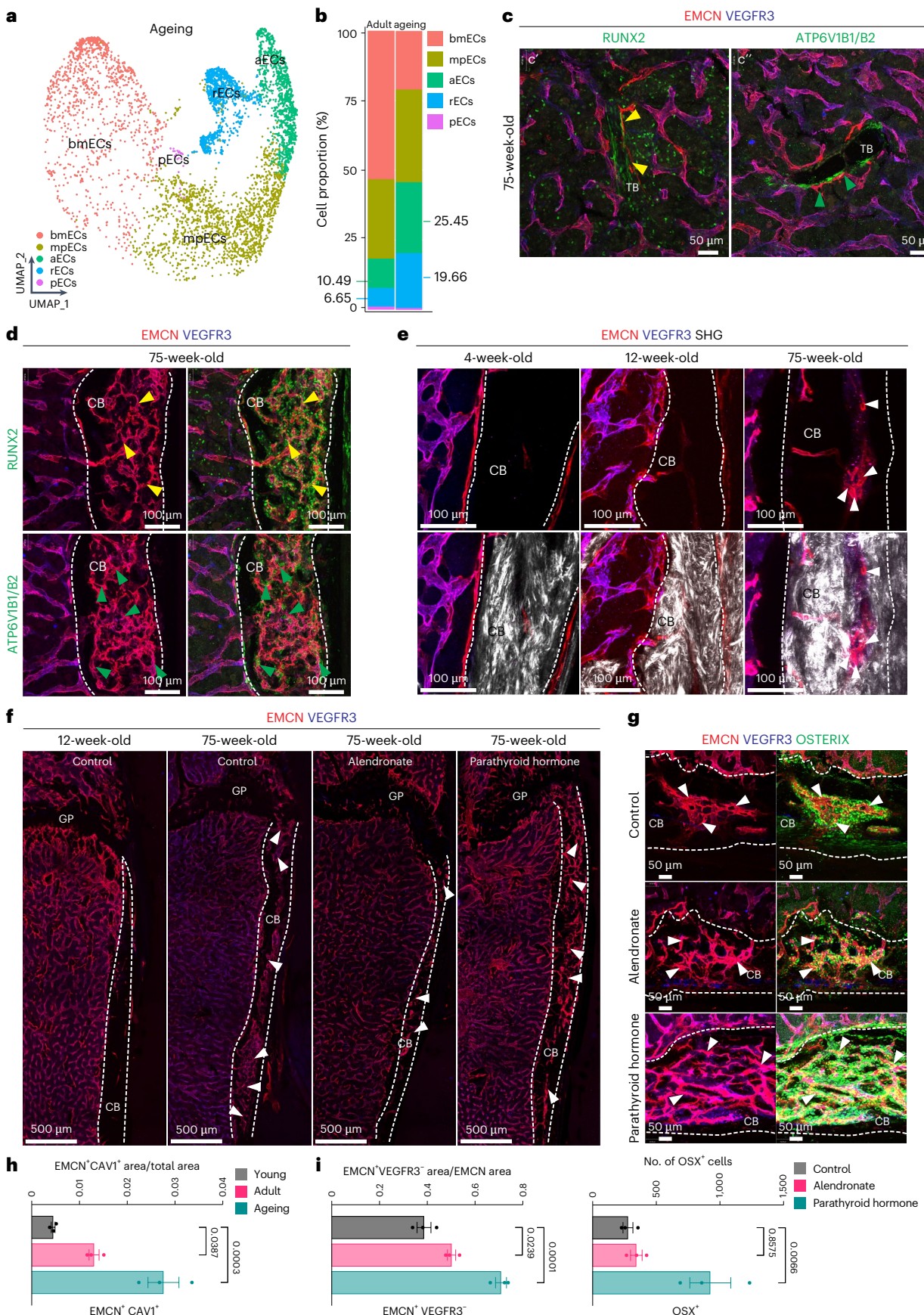

VEGFR3 as well as association with RUNX2[+] osteoprogenitors and vATPase[+] osteoclasts (Fig. 8d,e). The abundance of type R ECs is not only increased by ageing but also by anti-osteoporosis treatments (Extended Data Figs. 9a–e and 10a,b). Administration of the bisphosphonate alendronate, a potent inhibitor of osteoclast-mediated bone resorption[45], results in strong increases of trabecular bone thickness, connectivity density and bone volume over tissue volume (BV/TV) (Extended Data Fig. 10a). Alendronate also increases the fraction of type R ECs in scRNA-seq data and leads to an expansion of EMCN[+]VEGFR3[−] capillaries in association with osteoprogenitors around trabecular bone despite inhibited osteoclast activity (Extended Data Fig. 9c–f). This suggests that type R capillaries respond to an increase in trabecular bone even when osteoclast-mediated resorption and thereby bone turnover are pharmacologically blocked.

Active peptide fragments of parathyroid hormone, such as PTH1-34 (ref. 46), are potent inducers of osteogenesis. Daily PTH1-34 administration over a period of 4 weeks leads to significant increases in trabecular bone thickness, connectivity density and BV/TV (Extended Data Fig. 10b). These changes involve notable increases in osteoblast-lineage cells but also type R capillaries (Extended Data Fig. 9e–g). Treatment of aged (75-week-old) mice with alendronate or PTH also increases type R capillaries, quantified as CAV1[+]EMCN[+] and VEGFR3[−]EMCN[+] vessels, in long bone (Fig. 8g,i). Inside aged cortical bone, PTH1-34 induces a strong expansion of type R capillaries but also of vessel-associated osteoblast-lineage cells (Fig. 8g–i). Together, these data show that type R ECs respond to clinically relevant anti-osteoporosis treatments and are associated with active but also inhibited bone remodelling.

## Discussion

The heterogeneity and functional specialization of cells in bone is an important question with relevance for haematopoiesis but also for bone formation, homeostasis, age-related osteopenic bone loss and osteoporosis. As several studies have investigated the cellular composition of bone and marrow with scRNA-seq approaches, it is helpful to compare EC heterogeneity in different datasets. Tikhonova et al.[47] have sorted genetically labelled ECs and other cells from adult mice. Within the 4,669 cells of the *Cdh5*[+] EC cluster, *Stab2*[+] sinusoidal capillary ECs and *Ly6a*[+] aECs were defined. Our analysis of this data (Supplementary Fig. 1a) shows that only a few cells in the arterial subcluster express the type R markers *Cav1*, *Dach1* and *C1qtnf9* (Supplementary Table 1). Baryawno et al.[29] used a combination of flushing and crushing of bone samples to obtain 5,703 *Cdh5*[+] ECs belonging to sinusoidal, arteriolar and aEC subclusters. The latter of those contains a clearly identifiable group of *Cav1*[+], *Dach1*[+] and *C1qtnf9*[+] ECs (Supplementary Fig. 1b). Baccin et al.[28] obtained a total of 7,497 cells from crushed bone samples. While numbers for the EC fraction are not provided, two clusters, namely sinusoidal and aECs, were identified. This dataset contains a group of cells that might represent rECs based on gene expression (Supplementary Fig. 1c). Our own results, based on a robust number of 62,298 ECs from different datasets, support the existence of type H, sinusoidal and arterial subpopulations in addition to type R ECs (Supplementary Fig. 1d). Specifics of the bone sample processing protocols might be, at least in part, responsible for the differences in the EC subpopulations identified in the studies mentioned above. Flushing of BM, a method that is well established for the isolation of haematopoietic cells, often involves the removal of the metaphysis, which contains trabecular bone together with type H and type R capillaries. Flushing also fails to isolate endosteal vessels[2]. By contrast, crushing of bone can release bone-associated EC populations but might increase the fraction of damaged cells that will not pass quality control steps. Given these limitations, future methodological advancements and increased sequencing depth might enable the identification of further EC subpopulations in health and disease. In this context, it is noteworthy that recent work has identified another new capillary subtype (termed type S) in the epiphysis, the rounded distal end of long bone, which is covered by articular cartilage[48]. As the epiphysis was not included in our analysis, future work will have to establish the relationship between type S cells and other bone EC subpopulations.

Our findings establish that type R vessels are post-arterial capillaries, which control oxygen availability and thereby local metabolic properties of perivascular cell populations at the surface of remodelling bone. Oxygenation levels inside bone are not homogeneous and, direct measurement in murine calvarial BM in vivo has shown that values range from 0.6–2.8% oxygen despite the abundance of sinusoidal vessels[30]. In the endosteum covering the inner bone surface, a region reported to contain mostly smaller vessels, oxygenation is slightly higher (2.9%) compared with 1.3–2.4% in areas further away from bone[30]. These values indicate that cells inside bone are exposed to substantial levels of hypoxia. Studies on cultured BMSCs have revealed that both atmospheric oxygen (20–21%) or continuous hypoxia can impair osteogenic differentiation[35,36]. By contrast, transient or mild hypoxia (2–3% oxygen), similar to what has been measured for the endosteum in vivo, can facilitate osteogenesis in cell culture[35,49,50]. Increasing metabolic demand during postnatal bone growth and size expansion might be the leading cause of type R capillary formation during the transition from postnatal growth to adulthood. Arteries directly feed into the vasculature of the metaphysis and endosteum[17], which, together with transcortical blood flow[10], might be sufficient to supply relatively small postnatal and adolescent skeletal elements. By contrast, direct supply of arterial blood to trabecular bone might be required inside the larger adult long bone. Supporting this concept, DACH1-mediated expansion of arteries and type R capillaries enables osteogenesis in an area of diaphyseal BM that is normally devoid of trabecular bone. We therefore propose that type R post-arterial capillaries generate a local tissue microenvironment that facilitates trabecular bone formation.

Increased oxygen demand and elevated metabolic needs might also explain the higher abundance of type R vessels in response to pharmacological inhibition of osteoclast activity, a condition that increases bone mass but suppresses bone turnover. The role of oxygenation is less clear for osteoclasts. Published reports indicate that oxygen can positively or negatively influence osteoclastogenesis and bone resorption[51–54]. However, our scRNA-seq data indicate that rECs express many known regulators of osteoblast-lineage cells or osteoclasts at higher levels than other bone ECs. Thus, future work will have to address how metabolic parameters, cytokines, growth factors or other type R capillary-derived factors participate in the regulation of bone remodelling.

While the DACH1 genetic data and the treatment with anti-osteoporosis drugs suggest that type R capillaries have a positive influence on osteogenesis and bone mass, the emergence of these vessels in ageing cortical bone argues for a potential role in bone loss. Cortical bone thinning and porosity are key features of age-associated osteopenia in human participants[44,55]. Our study shows that cortical pores in ageing mice contain type R capillaries together with associated osteoprogenitors and osteoclasts. Thus, crosstalk between ECs and other cell types might influence whether pores expand through osteoclast-mediated resorption or get filled by newly formed bone. We propose that type R capillaries are important mediators not only of bone growth but also of physiological and pathological bone remodelling with potential relevance for ageing and osteoporosis.

## Online content

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

[1]Max Planck Institute for Molecular Biomedicine, Department of Tissue Morphogenesis, Münster, Germany. [2]Max Planck Institute for Molecular Biomedicine, Single Cell Multi-Omics Laboratory, Münster, Germany. [3]Department of Immunology, Genetics, and Pathology, Uppsala University, Uppsala, Sweden. [4]Department of Biology, Stanford University, Stanford, CA, USA. [5]Stanford Cardiovascular Institute, Stanford University School of Medicine, Stanford, CA, USA. [6]Institute for Stem Cell Biology and Regenerative Medicine, Stanford University, Stanford, CA, USA. [7]Department of Physiology and Pharmacology, Karolinska Institute, Stockholm, Sweden. [8]Centre for Bone and Arthritis Research, Institute of Medicine, Sahlgrenska Academy at University of Gothenburg, Gothenburg, Sweden. [9]Max Planck Institute for Molecular Biomedicine, Flow Cytometry Unit, Münster, Germany. [10]Department of Women's and Children's Health, Karolinska Institutet and Pediatric Endocrinology Unit, Karolinska University Hospital, Stockholm, Sweden. [11]Wihuri Research Institute, Helsinki, Finland. [12]Translational Cancer Medicine Program and Department of Biochemistry and Developmental Biology, University of Helsinki, Helsinki, Finland. [13]Howard Hughes Medical Institute, Stanford University, Stanford, CA, USA. [14]Present address: Department of Stem Cell and Regenerative Biology, Harvard University, Cambridge, MA, USA. [15]Present address: National Centre for Biological Sciences, Tata Institute of Fundamental Research, Bengaluru, India. ✉e-mail: ralf.adams@mpi-muenster.mpg.de

## Methods

### Animal models

C57BL/6J female mice at the indicated age were used for the analysis of wild-type femurs. For labelling and scRNA-seq analysis of bone ECs, *Cdh5-mTomato-nGFP* (*Cdh5-mT/nG*) transgenic reporter mice were used[19], labelling of osteoblasts in vivo utilized *Sp7-mCherry* reporter mice[56]. *Efnb2-H2BGFP*[57] knock-in animals express H2B-coupled GFP under control of the gene encoding ephrin-B2.

Tamoxifen or 4-OHT was administered as indicated in the figures. Lineage-tracing experiments were performed by mating *Bmx-CreERT2* (ref. 20), *Flt4-CreERT2* (ref. 22) or *Aplnr-CreERT2* (ref. 23) with *R26-mT/mG* reporter animals[21]. Cre activity and GFP expression were induced by intraperitoneal injection at the indicated doses.

Inducible and EC-specific *Dach1* overexpression (*Dach1^OE^*)[26] was accomplished by interbreeding with *Aplnr-CreERT2* (ref. 23) and *R26-mT/mG* mice. Resulting offspring was analysed at P30 after tamoxifen administration on P1 and P2. For inactivation of *Dach1 or Dll4*, *Cdh5(PAC)-CreERT2* transgenic mice[27] were bred to animals carrying Dach1 (*Dach1*^flox/flox^) alleles[25], which were derived from *Dach1*^tm1a(KOMP)Wtsi^ ES cells (https://www.komp.org/ProductSheet.php?cloneID=689703) or loxP-flanked Dll4 (*Dll4*^flox/flox^)[58], respectively. Cre-negative offspring was used as a littermate control. All pups were injected with tamoxifen as indicated and analysed at the age of 12 weeks. Tamoxifen (Sigma, T5648) stocks were prepared by dissolving 50 mg in 500 µl of 100% ethanol (Fisher Scientific, BP2818-500 (200 proof)), vortexing for 10 min, and adding an equal volume of Kolliphor EL (Sigma, C5135). One-milligram aliquots were stored at −20 °C and dissolved in PBS before injection. 4-OHT (Sigma, H7904) stocks were prepared by dissolving 25 mg in 500 µl of 100% ethanol, vortexing for 10 min and adding an equal volume of Kolliphor EL. One-milligram aliquots were stored at −20 °C and dissolved in PBS before injection.

For treatment with anti-osteoporosis drugs, mice received daily intraperitoneal injections of PBS or PTH (1-34, Bachem, 4095855) (100 µg kg^−1 body weight) or alendronate (Tocris, MK 217, 4014) (2 mg kg^−1 body weight) every other day. Aged mice received PTH (150 µg kg^−1 body weight) daily or alendronate (3 mg kg^−1).

No randomization was used. Data collection and analysis were not performed blind to the conditions of the experiments.

*Flt4-CreERT2* lineage-tracing experiments were performed at Uppsala University and approved by the Uppsala Laboratory Animal Ethical Committee. The remaining animals were housed in an animal facility at the Max Planck Institute for Molecular Biomedicine in specific pathogen-free conditions in individually ventilated cages with a consistent light–dark cycle, free access to food and water and controlled temperature and humidity. Mice were routinely genotyped by PCR using allele-specific primers. All animal experiments were conducted following the 3Rs (replacement, reduction and refinement) and in accordance with institutional guidelines and laws, following protocols (84-02.04.2016.A160, 81-02.04.2018.A171, 81-02.04.2020.A212, 81-02.04.2020.A416 and 81-02.04.2022.A198) approved by the Landesamt für Natur, Umwelt und Verbraucherschutz of North Rhine-Westphalia, Germany.

### Sorting and sequencing of bone endothelial cells from *Cdh5-mT/nG* mice

For isolation of bone ECs, single-cell suspensions were prepared from juvenile (*n* = 2 male, 2 female mice), adult (*n* = 6 female mice) or aged (*n* = 3 female mice) *Cdh5-mT/nG*[19] reporter femurs and tibias. Surrounding muscle tissue was removed and the epiphysis was detached. Cleaned samples were collected in a solution of collagenase types I and IV (2 mg ml^−1) before bones were cut into small pieces and crushed using mortar and pestle. After 20 min of digestion at 37 °C, crushed samples were transferred to 70-µm strainers in 50-ml tubes to obtain single-cell suspensions, which were resuspended in blocking solution (1% BSA and 1 mM EDTA in PBS without Ca^{2+}/Mg^{2+}), followed by centrifugation

at 300*g* for 5 min. Cells were washed thrice with ice-cold blocking solution, filtered through 50-µm strainers and resuspended in an appropriate volume of blocking solution. For fluorescence-activated cell sorting (FACS) of ECs, haematopoietic cells were removed with the help of a lineage cell depletion kit (Miltenyi Biotec, 130-090-858). The remaining cells were resuspended in 0.5% BSA in PBS and processed on a FACS Aria II cell sorter (BD Biosciences) using a 100-µm nozzle. Flow cytometry data were analysed using FACSDiva (v.8.0.2, BD Biosciences) and FlowJo (v.10.8.1) software. Sorted nGFP and mTomato double-positive cells were collected in a 0.05% blocking buffer for scRNA-seq.

Single-cell suspensions were subjected to droplet-based scRNA-seq using a chromium controller (10x Genomics). Libraries were prepared using a Chromium single cells 3′ reagent kit (10x Genomics) following the manufacturer's protocol. scRNA-seq libraries were evaluated and quantified using an Agilent Bioanalyzer with a High Sensitivity DNA kit (Agilent Technologies, 5067-4626) and Qubit dsDNA HS Assay kit (Thermo Fisher Scientific, Q32851). Libraries were sequenced using a NextSeq500 sequencer (Illumina) with a High Output kit (150 cycles) (Illumina, TG-160-2002).

### Single-cell RNA sequencing of non-haematopoietic cells from *Dach1^OE^* bone

*Dach1^OE^* single-cell suspensions were generated by bone preparation, digestion and lineage depletion of control (*n* = 2 mice) or *Dach1^OE^* (*n* = 2 mice) as described above. Lineage-negative cells were further incubated with CD45 microbeads (Miltenyi Biotec, 130-052-301) and CD117 microbeads (Miltenyi Biotec, 130-091-224) to maximize the removal of haematopoietic cells. Following magnetic-activated cell sorting, cells were resuspended at a final concentration of 10^6 cells per ml in 0.05% BSA in PBS and loaded onto a microwell cartridge of the BD Rhapsody Express system (BD Biosciences) following the manufacturer's instructions. Libraries were prepared using the BD Rhapsody WTA kit (BD Biosciences, 633802) and sequenced on an Illumina NextSeq500 using a High Output kit (150 cycles, Illumina) for control or *Dach1^OE^* samples.

### Sample preparation and immunostaining

Femurs and tibias were isolated and fixed in ice-cold 2% PFA, typically around 6–24 h at 4 °C, based on the age group. After fixation, bones were subjected to 0.5 M EDTA decalcification under constant agitation for 1–4 days at 4 °C. Bones were immersed in 20% sucrose and 2% polyvinylpyrrolidone (PVP) solution for 12–24 h, embedded in 8% gelatin using 20% sucrose and 2% PVP and frozen at −80 °C. For immunostaining, sections (60–100-µm thickness) were generated using a Leica 3050 cryostat with low-profile blades. Sections were air-dried, permeabilized with 0.3% Triton X-100 in PBS for 15 min at room temperature (RT) and blocked using 6% donkey serum prepared with 0.3% Triton X-100 for 45 min at RT. Slides were washed with PBS three times at 10-min intervals and incubated overnight at 4 °C with primary antibodies diluted in a solution containing 5% donkey serum in PBS. Primary antibodies were: rat monoclonal anti-endomucin (V.7C7) (Santa Cruz, sc-65495, 1:100 dilution), goat polyclonal anti-Vegfr3 (R&D Systems, AF743, 1:50 dilution), rabbit polyclonal anti-Cav1 (Cell Signalling, 3238, 1:50 dilution), chicken polyclonal anti-GFP (Abcam, ab13970, 1:200 dilution), rabbit polyclonal anti-OSTERIX (Abcam, ab22552, 1:300 dilution), rabbit monoclonal anti-Runx2 (Abcam, ab192256, 1:200 dilution), rabbit monoclonal anti-Ki-67 (Cell Signalling, 12202, 1:200 dilution), rabbit monoclonal anti-vATPase (Abcam, ab200839, 1:200 dilution), rabbit polyclonal anti-collagen IV (AbD Serotec, 2150-1470, 1:100 dilution), rabbit polyclonal anti-collagen IIIA1 (Abcam, ab7778, 1:100 dilution), goat polyclonal anti-Hif1a (R&D Systems, AF1935, 1:100 dilution), rabbit polyclonal anti-Hif1a (Santa Cruz, sc-10790, 1:100 dilution), rabbit monoclonal anti-GPI (Cell Signalling, 94068, 1:200 dilution), rabbit monoclonal anti-CD73 (Cell Signalling, 13160, 1:200 dilution), rabbit monoclonal anti-Hmox1 (Cell Signalling, 86806, 1:200 dilution), rabbit polyclonal anti-Dach1 (Proteintech, 10914-1-AP, 1:200

dilution), rabbit polyclonal anti-NG2 (Millipore, AB5320, 1:200 dilution), goat polyclonal anti-Cxadr (R&D Systems, AF2654, 1:50 dilution), rabbit monoclonal anti-Sox11 (Abcam, ab134107, 1:800 dilution), rat monoclonal anti-Madcam1 (Abcam, ab80680, 1:100 dilution), rabbit polyclonal anti-Fabp4 (Abcam, ab13979, 1:100 dilution), mouse monoclonal anti-Mct4 (Santa Cruz, AF647, 1:100 dilution), rabbit monoclonal anti-Glut1 (Cell Signalling, 12939, 1:100 dilution), goat polyclonal anti-Dll4 (R&D Systems, AF1389, 1:50 dilution) and chicken polyclonal anti-Mct1 (Millipore, AB1286-I, 1:100 dilution). Following overnight incubation, sections were washed three times with ice-cold PBS and incubated with DAPI and suitable secondary antibodies: anti-rat Alexa Fluor 488 (Thermo Fisher Scientific, A21208, 1:100–1:300 dilution), anti-rabbit Alexa Fluor 488 (Thermo Fisher Scientific, A21206, 1:100–1:300 dilution), anti-chicken Alexa Fluor 488 (Jackson Laboratories, 703-545-155, 1:100–1:300 dilution), anti-goat Alexa Fluor 546 (Thermo Fisher Scientific, A11056, 1:100–1:300 dilution), anti-rabbit Alexa Fluor 546 (Thermo Fisher Scientific, A10040, 1:100–1:300 dilution), anti-rat Alexa Fluor 594 (Thermo Fisher Scientific, A21209, 1:100–1:300 dilution), anti-rabbit Alexa Fluor 594 (Thermo Fisher Scientific, A21207, 1:100–1:300 dilution), anti-goat Alexa Fluor 594 (Thermo Fisher Scientific, A11058, 1:100–1:300 dilution), anti-goat Alexa Fluor 647 (Thermo Fisher Scientific, A21447, 1:100–1:300 dilution), anti-rabbit Alexa Fluor 647 (Thermo Fisher Scientific, A31573, 1:100–1:300 dilution) and anti-rat Alexa Fluor 647 (Jackson Laboratories, 712-605-153, 1:100–1:300 dilution). Following secondary staining, sections were washed three times with PBS at RT and mounted on cover slips using FluroMount-G (Southern Biotech, 0100-01).

Retinas were fixed with 4% PFA for 1 h at RT, washed with PBS and permeabilized with 0.5% Triton X-100 and blocked with 5% donkey serum overnight on a rotating platform at 4 °C. Tissues were incubated with primary antibodies rat anti-CD31 (BD Pharmingen, 553370; 1:100 dilution) followed by secondary antibody anti-rat Alexa-594 (Thermo Fisher Scientific, A21209, 1:300 dilution) overnight on a rotating platform at 4 °C and mounted using FluroMount-G.

Intestine immunostaining followed a previously published protocol[59]. Mice were perfused with PBS, followed by 4% PFA at RT, and the upper jejunum was collected in ice-cold PBS, cut open and washed thoroughly with ice-cold PBS. The cleaned jejunum was fixed with 4% PFA overnight on a rotating platform at 4 °C, washed with ice-cold PBS and incubated overnight with 10% sucrose in PBS followed by 20% sucrose and 10% glycerol in PBS. For staining, samples were permeabilized with 0.5% Triton X-100 and blocked with 5% donkey serum. Tissues were incubated with primary antibodies rabbit anti-DACH1 (Proteintech, 10914-1-AP, 1:500 dilution), goat anti-VEGFR2 (R&D Systems, AF644; 1:100 dilution) followed by secondary antibodies anti-rabbit Alexa-488 (Thermo Fisher Scientific, A21206; 1:300 dilution) and anti-goat Alexa-594 (Thermo Fisher Scientific, A11058; 1:300 dilution) overnight on a rotating platform at 4 °C, incubated and mounted with Histodenz (Sigma, D2158) in DAPI.

## Human bone sample preparation and immunostaining
Adolescent and aged human subchondral bones were collected from patients undergoing epiphysiodesis surgery at Karolinska University Hospital or femoral head replacement surgery in Ortopedkliniken Sodersjukhuset in Stockholm, Sweden, respectively. The study involved three young patients (11–14 years old) and three aged patients (77–78 years old). Collection of human material was conducted with informed consent by patients/parents and with approval from the Swedish Ethical Research Authority and the National Board of Health and Welfare (ethical permissions numbers 2014/276-31/2 and 97-214). No financial compensation was provided to participants.

For adolescent patients, subchondral bone samples were obtained using a BM biopsy needle targeting the proximal tibia and distal femur growth plate. Subchondral bones adjacent to the growth plate cartilage were fixed with 4% PFA in PBS for 16 h. Femoral heads from aged patients

were dissected into smaller tissue sections (<1.5 cm in each dimension) and fixed in 4% PFA/PBS at 4 °C with agitation for 16 h before two washes with PBS (1 h each) and decalcification in Morse solution (20% formic acid/10% citrate) for 24 h at 4 °C with agitation. Decalcified samples were washed twice with PBS (1 h each), incubated successively with 15% sucrose (1 h) followed by 30% sucrose for 24 h and embedded in OCT before 100-µm-thick cryosections were mounted on a glass slide.

For immunolabelling, human bone sections were washed three times for 10 min each with PBS and 0.1% Tween-20. Sections were incubated in blocking buffer (PBS, 0.2% CHAPS, 5% donkey serum, 10% DMSO and 25 mM EDTA, pH 8) for 1 h before staining with primary antibodies mouse anti-VEGFR3 (R&D Systems, MAB3491, 1:100 dilution) and goat anti-VEGFR2 (R&D Systems, AF357; 1:100 dilution) overnight. Next, sections were washed thrice with PBS and 0.1% Tween-20 (10 min each) and incubated for 24 h with secondary antibodies Cy3 AffiniPure goat anti-mouse IgG (Jackson ImmunoResearch, 115-165-003; 1:400 dilution), donkey anti-goat Alexa-647 (Thermo Fisher Scientific, A32849; 1:400 dilution) and DAPI (1:500 dilution). Sections were washed three times with PBS and 0.1% Tween-20 for 10 min each and mounted with FluoroShield (F6057, Sigma). Imaging of young human bones was performed using an LSM980 confocal microscope (Zeiss), whereas aged human samples were imaged using a Yokogawa Confocal microscope (Nikon).

## Dextran permeability and bone histomorphometry
To assess vascular permeability, dextran (tetramethylrhodamine 2,000,000 MW, lysine fixable, Thermo Fisher, D7139) was injected into mice retro-orbitally. At 1 h after injection, mice were killed and femurs were processed for imaging as described above.

For histomorphometry analysis of bone turnover, mice received two intraperitoneal injections of bone-staining dyes with an interval of 1 week between injections. Calcein green (40 mg kg$^{-1}$; Sigma, C0875) and Alizarin Red (40 mg kg$^{-1}$; Sigma, A3882) were prepared in 2% sodium bicarbonate (Sigma, S4019) to label existing and new bone fronts, respectively. Four days later, femurs were dissected and fixed with 2% PFA overnight at 4 °C, followed by three PBS washes. Bones were placed in 20% sucrose and 2% PVP solution for 72 h, embedded in 8% gelatin using 20% sucrose and 2% PVP and frozen at −80 °C. Bones were carefully cryosectioned (at 20-µm thickness) and mounted as described above. For *Dach1*$^{OE}$, histomorphometry analysis was performed with an interval of 5 days and missing values were imputed with a minimum MAR of 0.1 µm d$^{-1}$ for the trabecular bone[60].

## HUVEC cell culture, siRNA transfection and immunostaining
HUVECs (Gibco, C-003-5C, lot no. 2219451) were cultured in EBM-2 endothelial cell medium (Lonza, CC-3156) supplemented with EGM-2 Single Quots (Lonza, CC-4176). For RNA and immunostaining, cells were seeded in 0.1% gelatin-coated six-well plates and µ-slide eight-well plates (Ibidi, 80826).

Cells (passage 4) were transfected with negative control (Ambion, 4390844) or *DACH1*-targeting (Thermo Fisher Scientific, s3899) siRNAs using Lipofectamine RNAiMAX (Invitrogen, 13778). Knockdown efficiency was validated using qPCR. After transfection, the medium was replaced after 4 h and the cells were incubated for 48 h before analysis.

After 48 h, cells were fixed with 4% PFA for 10 min at RT. All further steps were performed at RT unless otherwise mentioned. After fixation, cells were washed three times with PBS for 5 min each and permeabilized with ice-cold 0.1% Triton X-100 in PBS for 10 min at 4 °C. Cells were blocked with 5% donkey serum in PBS for 30 min. Primary antibodies, rabbit polyclonal anti-DACH1 (Proteintech, 10914-1-AP, 1:500 dilution) and goat polyclonal anti-Cdh5 (R&D Systems, AF938, 1:100 dilution) were then incubated in blocking buffer for 1 h. Following three 10-min washes with PBS, cells were incubated with DAPI (Sigma, D9542) and secondary antibodies: anti-mouse Alexa Fluor 488 (Thermo Fisher Scientific, A21202, 1:500 dilution) and anti-rabbit Alexa Fluor 546

(Thermo Fisher Scientific, A10040, 1:500 dilution) for 30 min. After three additional PBS washes for 10 min each, FluroMount-G (Southern Biotech, 0100-01) was introduced into the μ-slide well (Ibidi, 80826). Imaging was performed using an LMS780 confocal microscope (Zeiss).

### Culture and differentiation of human mesenchymal stem cells

Human mesenchymal stem cells (HMSCs) (Lonza, PT-2501) were expanded in MSCGM (Lonza, PT-3001) at 37 °C in 5% $CO_2$ in a humidified hypoxic incubator (1% $O_2$). Cells were expanded to passage 6 before use.

For osteogenic differentiation, HMSCs were seeded in 12- and 24-multiwell plates for 40–50% confluency with MSCGM and incubated overnight at 37 °C in 5% $CO_2$ in a humidified incubator. The culture medium was replenished with a StemPro Osteogenesis Differentiation kit (Thermo Fisher Scientific, A1007201) supplemented with human CTRP9 (R&D Systems, 6537-TN), NT-3 protein (R&D Systems, 267-N3), PDGFD protein (R&D Systems, 1159-SB) or semaphorin 7A (R&D Systems, 2068-S7) at a final concentration of 100 ng ml$^{-1}$. Cells were cultured at 37 °C, 5% $CO_2$ in a humidified incubator for up to 21 days with the medium changed every 3–4 days. Cells were fixed using 4% PFA for 10 min at RT followed by 2% Alizarin Red staining (Sigma, A5533) for 1 h. Calcified nodules were visualized under a light microscope (Zeiss) and absorbance at 405 nm was measured with a spectrophotometer.

### In vitro osteoclast formation assay

BM cells were collected from 6-week-old C57BL/6 femurs and red blood cells were lysed with red blood cell lysis buffer (Sigma, 11814389001). The remaining cells were cultured in minimum essential medium α (MEMα, Thermo Fisher Scientific, 12571063) containing 10% FBS and macrophage colony stimulating factor (M-CSF, 10 ng ml$^{-1}$) overnight to obtain BM-derived monocytes/macrophages (BMMs). Non-adherent BMMs were collected, seeded in 100-mm culture dishes and incubated for 2 days with MEMα containing M-CSF (30 ng ml$^{-1}$) to generate osteoclast precursor cells (OCPs). OCP-derived osteoclasts were generated by seeding $2 \times 10^4$ cells per well in 96-well plates in MEMα-based osteoclast differentiation medium, including 10% FCS, 30 ng ml$^{-1}$ M-CSF (Peprotech, 315-02) and 50 ng ml$^{-1}$ RANKL (Peprotech, 315-11) at 37 °C, 5% $CO_2$ for 7–10 days. Differentiation medium was changed every 3 days and starting at day 3, the medium was supplemented with human CTRP9, NT-3, PDGFD or semaphorin 7A at 100 ng ml$^{-1}$. Cells were checked daily and incubation was stopped once large multinucleated cells emerged. TRAP-positive cells, labelled with a leukocyte acid phosphatase staining kit (Sigma, 387A), containing three or more nuclei and a full actin ring were counted as osteoclasts under a light microscope.

### Quantitative PCR

After transfection, total RNA was isolated from HUVECs using the Monarch Total RNA Miniprep kit (NEB, T2010S), according to the manufacturer's instructions and cDNA was prepared using the iScript cDNA synthesis kit (Bio-Rad, 1708890). Quantitative PCR was performed using gene-specific TaqMan probes (Thermo Fisher) for *DACH1* (Hs00974297_m1), *CA4* (Hs00426343_m1), *EFNB2* (Hs00187950_m1), *GJA4* (Hs01098016_m1), *DLL4* (Hs00184092_m1), *VEGFC* (Hs01099203_m1) and *GJA5* (Hs00979198_m1), together with SsoAdvanced Universal Probes Supermix (Bio-Rad). Gene expression was normalized to that of *GAPDH* (Hs99999905_m1, (Fisher, 4319413E) on a CFX96 Touch Real-Time PCR Detection System (Bio-Rad). The ΔΔCT method was used to quantify gene expression.

### scRNA-seq data analysis

FASTQ-format sequencing raw data was initially processed using UMI-tools (v.1.1.2). The data were then aligned to the mouse reference genome mm10 (https://www.ncbi.nlm.nih.gov/datasets/genome/GCF_000001635.20/) using STAR (v.2.7.10a) and quantified with

Subread featureCounts (v.2.0.3) to generate an expression matrix. For bone non-haematopoietic cells, the FASTQ-format sequencing raw data were processed with BD Rhapsody WTA Analysis pipeline (v.1.0) on SevenBridges Genomics online platform (SevenBridges) and expression matrix was used for further data analysis. Data normalization, dimensionality reduction and visualization were performed using the Seurat package (v.4.3.0), unless otherwise specified.

Cells were filtered based on the criteria of number of genes per cell (nFeature_RNA) between 500 and 6,000 and percentage of mitochondrial genes (percent.mito) <25. Additionally, genes were filtered to include only those present in a minimum of three cells. After filtering, matrices were normalized using the NormalizeData function with the LogNormalize method and a scale factor of 10,000. Variable genes were identified using the FindVariableFeatures function, selecting the top 2,000 genes with the variance stabilizing transformation method, while excluding genes related to the cell cycle (GO:0007049). Data integration was achieved through the FindIntegrationAnchors and IntegrateData functions with default options. The integrated data were further processed using the ScaleData and RunPCA functions. Statistically significant principal components were identified using the JackStraw method, which were then used for UMAP non-linear dimensional reduction.

For unsupervised hierarchical clustering analysis, the FindClusters function in the Seurat package was applied. Various resolutions between 0.1 and 0.9 were tested and the final resolution was selected based on the most stable and relevant outcome, utilizing the clustree R package (v.0.5.1) and considering previous knowledge. Cellular identity for each cluster was determined by finding cluster-specific marker genes using the FindAllMarkers function. Haematopoietic cells expressing *Ptprc* (Encoding CD45$^+$) were filtered out as part of quality control. Marker genes were considered with a minimum fraction of cells expressing the gene over 25% (min.pct = 0.25), and comparisons were made against established cell-type-specific marker genes. Monocle (v.2.30.0) was used for pseudo-time trajectory analysis. We performed dimensionality reduction using the DDRTree method with parameters max_components = 2 and norm_method = 'log'. Cluster-specific markers determined by the FindAllMarkers function of Seurat with min.pct = 0.25 were used as ordering genes for the trajectory. The resulting trajectory was then visualized using plot_cell_trajectory or plot_genes_in_pseudotime functions with default settings.

### Image processing and quantifications

Bone sections were imaged using a confocal microscope (Zeiss LSM780 and LSM980). The images were processed using Adobe Illustrator (Adobe), Adobe Photoshop (Adobe), ImageJ2 (v.1.53u; Fiji; http://fiji.sc/), Imaris (Oxford Instruments), Ilastik (https://www.ilastik.org/) and Volocity (v.6.3, Quorum Technologies) in compliance with the *Nature Cell Biology* guidelines for digital images. Brightness-contrast modifications were uniformly applied to the original images using Volocity to improve image quality and visualization.

Batch processing of images was carried out using a Fiji macro with identical parameters and thresholding. Cell number, length and area were quantified using Fiji. The number of arterioles and arteries was calculated based on CAV1 staining. Osteoblast numbers were calculated based on OSX staining, using the nuclei counter macro created by Fiji. Osteoclast area was calculated based on ATP6V1B1/B2 staining of bone sections normalized to the total selected area. EMCN$^+$CAV1$^+$, EMCN$^+$VEGFR3$^-$, CAV1$^+$, CD31$^+$, VEGFR2$^+$ and GFP$^+$ areas were normalized to their respective staining, as mentioned in the figures and legends.

Images are representative and were derived from various experiments, each employing consistent laser excitation and confocal scanner detection settings within the same experiment. All figures show maximum-intensity projections, and we analysed at least three mice

for each condition, unless explicitly mentioned otherwise. The illustrations in Figs. 1a and 7g and Extended Data Fig. 3e were created using BioRender (https://biorender.com).

## Micro-computed tomography

Femurs were fixed in 2% PFA overnight at 4 °C and washed with PBS. The samples were scanned using cabinet cone-beam μCT (100; SCANCO Medical) with a voxel size of 6 μm at 70 kVp, 86 μA and 600 ms integration time. The volume of interest was set between the trabecular section distal to the femur and above the growth plate, and the evaluation length was 2.5 mm for $Dach1^{i\Delta EC}$, 3 mm for $Dach1^{OE}$ and 5 mm for alendronate-treated femurs. Cortical thickness was evaluated manually using Fiji using the line selection tool.

## Statistics and reproducibility

Statistical analyses were performed using GraphPad Prism v.8 or v.10. All data are presented as mean ± s.e.m. unless indicated otherwise. Statistical significance was analysed using one-way ANOVA and unpaired two-tailed $t$-tests, as indicated in the legends. Statistical significance was set at $P < 0.05$. If variances between the groups differed significantly, an unpaired two-tailed $t$-test with Welch's correction was applied. Sample sizes are described in the figure legends. Experiments using primary cells were performed with at least three replicates and independently repeated three times. Reproducibility was ensured by independent biological samples or independent experiments. Three biological independent samples were used in Figs. 2c,d, 3a–f, 4b,c, 5a–g, 6b,f, 7c and 8c–e and Extended Data Figs. 1b,d–f, 2b,d,e, 4a–c, 5a,b,e, 6f, 7g–h,o, 8c and 9b,e–g. scRNA-seq experiments were performed once with pooled animals for each age and treatment group. Integration of these samples led to robust and reliable results across conditions. No statistical method was used to predetermine sample size and no animals were excluded from analysis. Data distribution was assumed to be normal, but this was not formally tested.

## Reporting summary

Further information on research design is available in the Nature Portfolio Reporting Summary linked to this article.

## Data availability

Sequencing data that support the findings of this study have been deposited at Gene Expression Omnibus (https://www.ncbi.nlm.nih.gov/geo/) under the accession code GSE239627. This is a SuperSeries record providing access to all related data, namely bone EC characterization and $Dach1^{OE}$ non-haematopoietic cells. The mouse reference genome, mm10 (https://www.ncbi.nlm.nih.gov/datasets/genome/GCF_000001635.20/) was used for mapping reads in this study. All other information supporting the findings of this study is available within this article or can be obtained from the corresponding author upon request. Source data are provided with this paper.

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

## Acknowledgements

We thank S. Ortega for *Flt4-CreERT2* mice and K. Müller for technical assistance. R.H.A. is supported by the Max Planck Society, the European Research Council (AdG 101139772, PROTECT) and the Deutsche Forschungsgemeinschaft (CRC 1366, project number 394046768). K.R.-H. is supported by the NIH (R01-HL128503) and is a Howard Hughes Medical Investigator. P.E.R.C. is supported by the National Institute of General Medical Sciences of the NIH (NIH T32GM007276) and National Science Foundation Graduate Research Fellowship Program (DGE-1656518). T.M. is supported by the Knut and Alice Wallenberg Foundation (2020.0057) and the Swedish Research Council (2020-02692). A.S.C. is supported by the Swedish Research Council (2020-02298) and the Swedish state under the agreement between the Swedish government and the county councils, the ALF agreement, in Gothenburg (ALFGBG-966178). V.M., V.V.D. and Y.-T.K. are part of the Cells in Motion-International Max Planck Research School and the University of Münster, Germany.

## Author contributions

V.M., K.K.S. and R.H.A. designed the study and interpreted results. V.M. generated and characterized mouse mutant lines and conducted all animal experiments, including bone sectioning, qPCR, immunostaining, cell biology, confocal imaging, quantification and scRNA data visualization. K.K.S., V.M. and B.D. prepared samples for scRNA-seq. V.V.D. analysed PTH μCT data. H.-W.J., V.M. and K.K.S. analysed scRNA-seq data. M.P. and T.M. performed *Flt4-CreERT2* experiments. K.R.-H., B.R., S.D. and P.E.R.C. provided critical mouse models and bone samples for preliminary experiments. E.B. processed retina and intestine samples. Y.-T.K. and M.G. performed cell culture experiments. I.P.U. contributed to image acquisition. N.T.L.C., L.S. and A.S.C. generated the human data. M.G.B. performed two-photon microscopy. M.S. provided critical input for flow cytometry.

## Funding

## Competing interests

The authors declare no competing interests.

## Additional information

**Extended data** is available for this paper at https://doi.org/10.1038/s41556-024-01545-1.

**Correspondence and requests for materials** should be addressed to Ralf H. Adams.

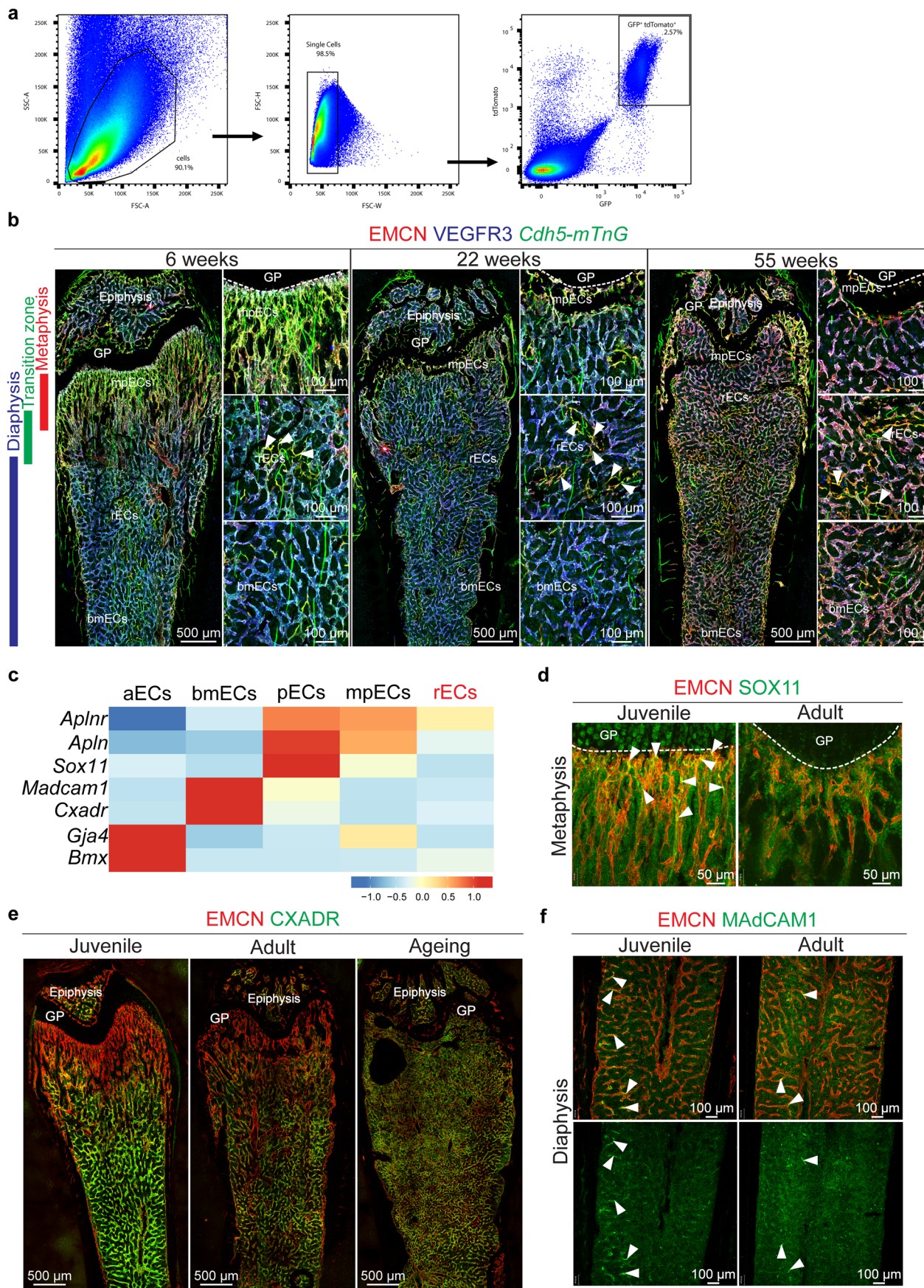

**Extended Data Fig. 1 | See next page for caption.**

**Extended Data Fig. 1 | Molecular heterogeneity of endothelial cells in long bone. a**. Representative FACS plots showing gating strategies for the isolation of ECs from *Cdh5-mTnG* reporter bone. **b**. Tile scan confocal images showing sections of 6, 22 or 55-week-old *Cdh5-mTnG* reporter bone co-stained for EMCN (red) and VEGFR3 (blue). The avascular growth plate (GP) and regions containing bmECs, mpECs and rECs are indicated. Small panels show higher magnifications of these areas. Arrowheads mark type R capillaries. Scale bars, 100μm (overview images) and 500μm (small panels). **c**. Heatmap showing the expression of selected marker genes for each EC subcluster (based on integrated scRNA-seq data for all age groups). **d**. Representative confocal images showing immunostaining for EMCN (red) and SOX11 (green, white arrowheads), marking pECs in the young but not adult metaphysis. **e**. CXADR (green) staining of bmECs in young, adult, and aging femur. Growth plate (GP) and epiphysis are indicated. **f**. MAdCAM1 (green, white arrowheads) expression in diaphyseal bmECs in young and adult femur. Scale bars, 50μm (**d**), 500μm (**e**), 100μm (**f**).

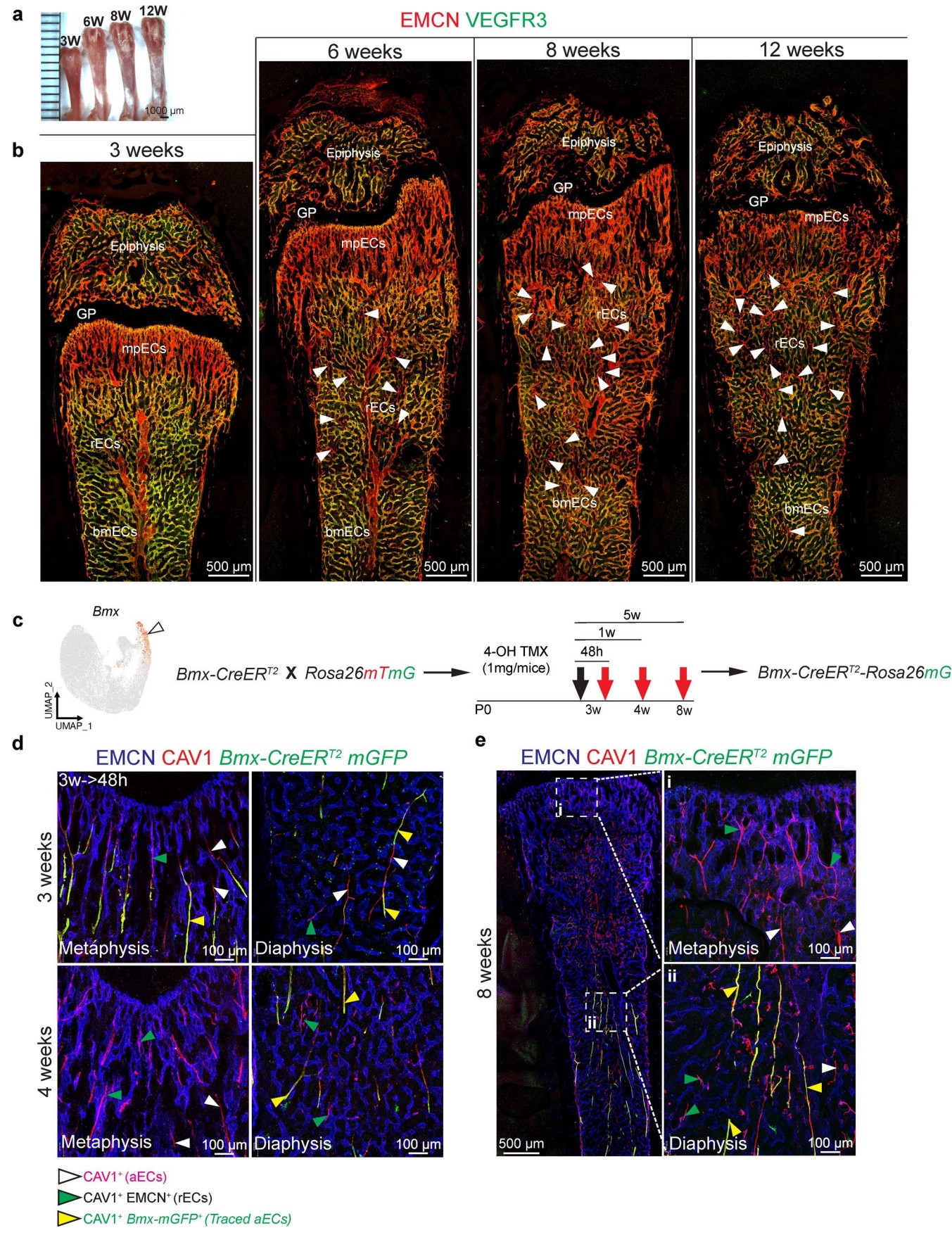

**Extended Data Fig. 2 | See next page for caption.**

**Extended Data Fig. 2 | Expansion of rECs in adolescent femur. a, b**, Freshly isolated wild-type femurs at the age of 3, 6, 8 or 12 weeks (W), as indicated. Tile scan confocal images of EMCN⁺ VEGFR3⁻ ECs (arrowheads) in sectioned femur at the indicated ages (**b**). Epiphysis and growth plate are labelled. Scale bars, 1000µm (**a**), 500µm (**b**). **c**. The UMAP plot (integrated scRNA-seq data of all age groups) showing *Bmx* expression in arterial endothelial cells (aECs), marked by arrowhead. Scheme of *Bmx-CreERT2* genetic fate mapping. 4-hydroxy tamoxifen (4-OHT) administration (1mg/mouse) is indicated by black arrow and red arrows mark time points of analysis. **d, e**, High-resolution confocal images and tile scan overview images of fate-tracked *Bmx-CreERT2 R26-mTmG* (GFP, green) in femur at the indicated time points after 4-OHT administration. Insets in (**e**) show metaphysis (i) and diaphysis (ii), respectively. White arrowheads mark CAV1⁺ EMCN⁻ aECs and yellow arrowheads GFP⁺ CAV1⁺ EMCN⁻ aECs. Green arrowheads indicate EMCN⁺ CAV1⁺ rECs, which are devoid of GFP signal. Scale bars, 100µm (**d**) and 500µm or 100µm (**e**).

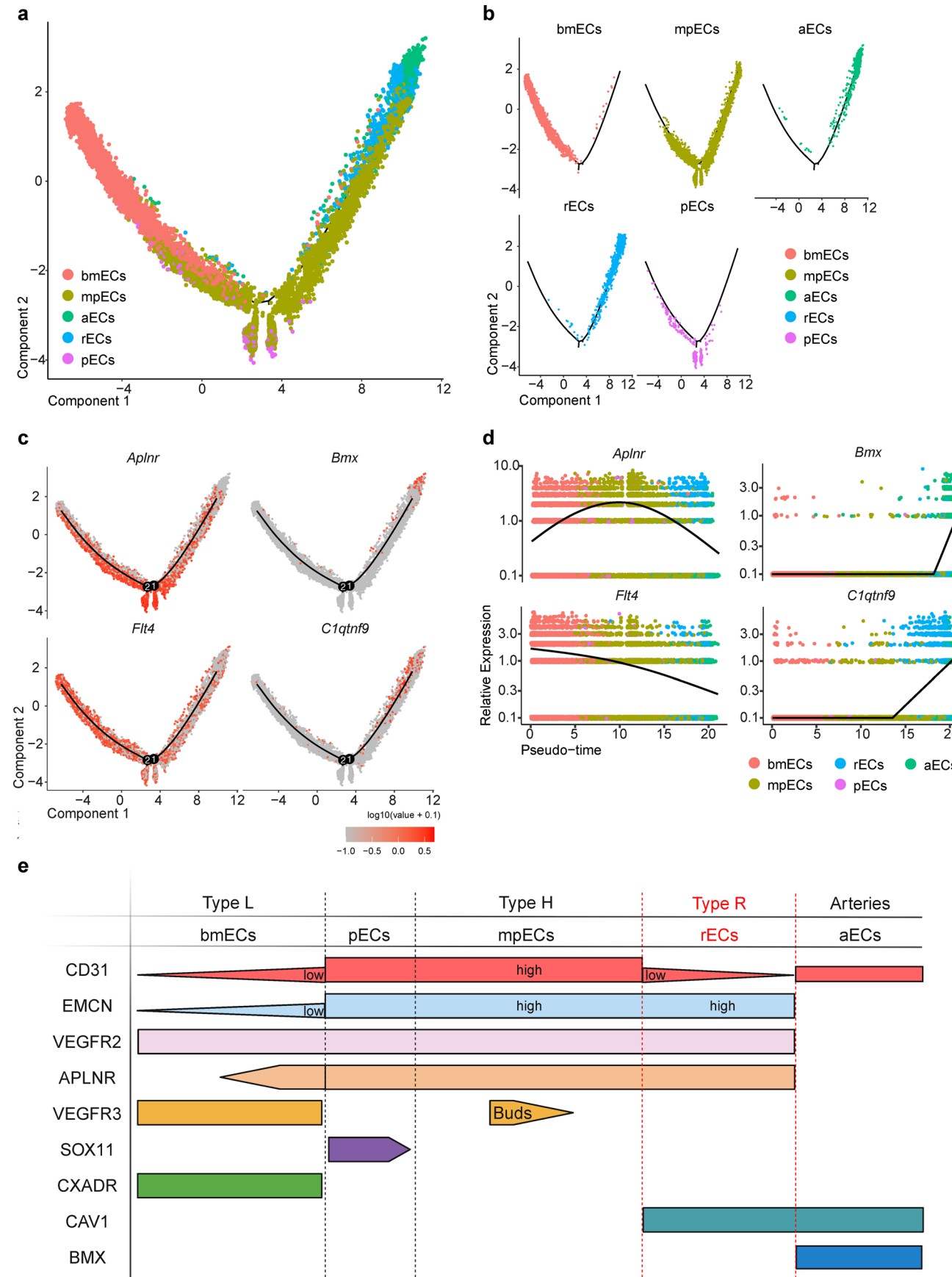

**Extended Data Fig. 3 | Molecular features of bone endothelial cells.**
**a-d**, Monocle trajectory analysis of bone EC subpopulations with coloured cell clusters (**a**), representation of individual clusters in trajectory (**b**), and cell type-specific relative gene expression shown in pseudo-time (**c, d**). Results are derived from integrated scRNA-seq data of all age groups. **e**, Schematic representation of major bone EC subpopulations along with designated markers.

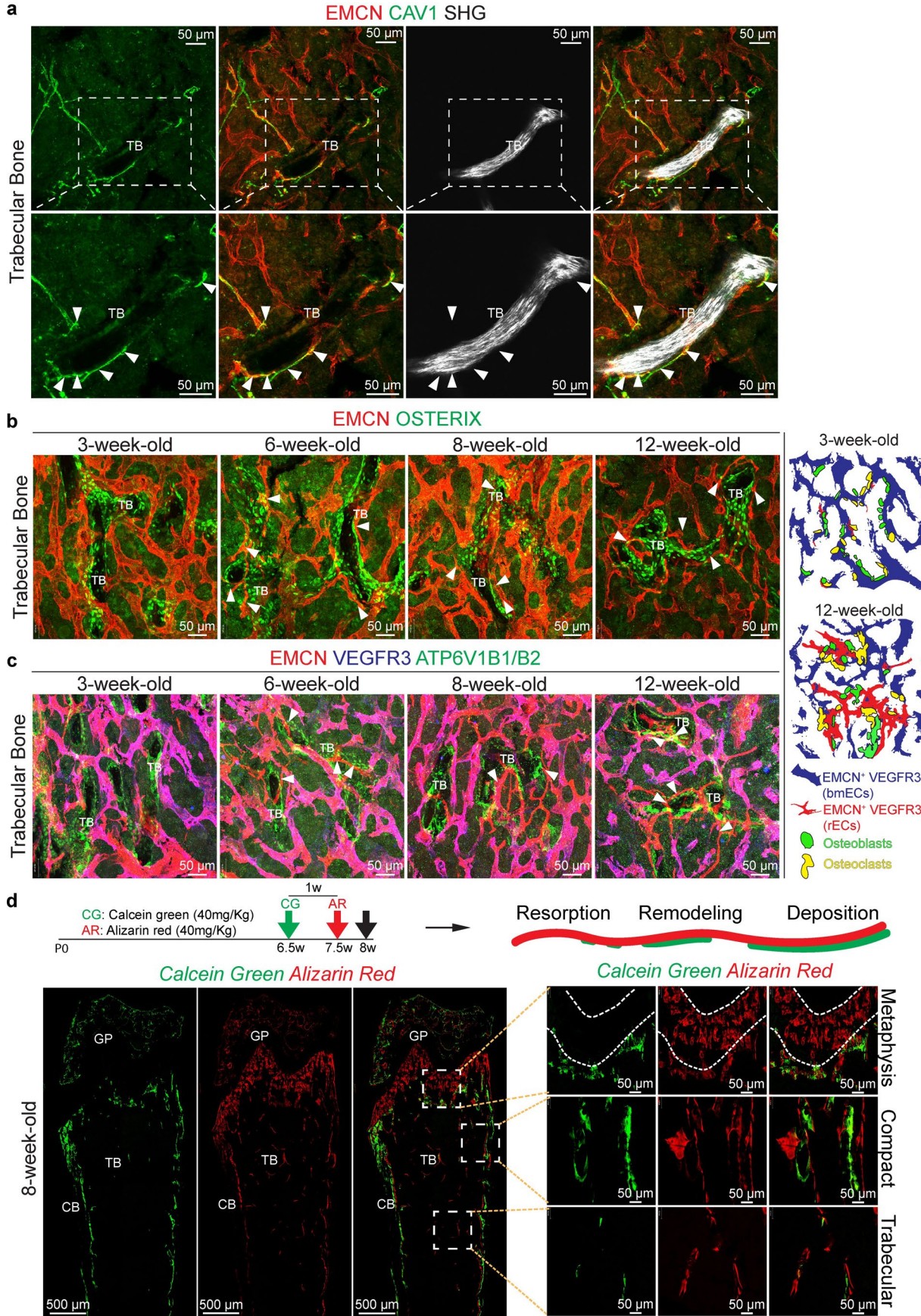

**Extended Data Fig. 4 | See next page for caption.**

**Extended Data Fig. 4 | Type R vessels are associated with remodelling bone. a**, High-magnification two-photon microscopy images of compact bone immunostained for EMCN (red) and CAV1 (blue) together with second-harmonic generation (SHG, white). Type R capillaries (arrowheads) and trabecular bone (TB) are indicated. Scale bars, 50μm. **b, c**, Maximum-intensity projections of sections from 3, 6, 8 or 12-week-old femur immunostained for EMCN (red) and OSTERIX (green) (**b**) or EMCN (red), VEGFR3 (blue) and ATP6V1B1/B2 (green) (**c**). Arrowheads mark type R capillaries near trabecular bone (TB). Scale bars, 50μm. Diagram on the right depicts association of rECs with osteoblast-lineage cells and osteoclasts. **d**. Injection scheme and time points of imaging after labelling of wild-type mice with Calcein Green (CG) and Alizarin Red (AZ). Tile scan imaging of sectioned femur from 8-week-old wild-type after double labelling. Insets depict active resorption in the metaphysis with $CG^{-} AR^{high}$ labelling (top row), active osteogenesis ($CG^{high} AR^{high}$) in compact bone (middle row), and active remodelling ($CG^{low} AR^{high}$) of trabecular bone (bottom row). n=3 mice. Scale bars, 500μm (overview images) and 50μm (insets).

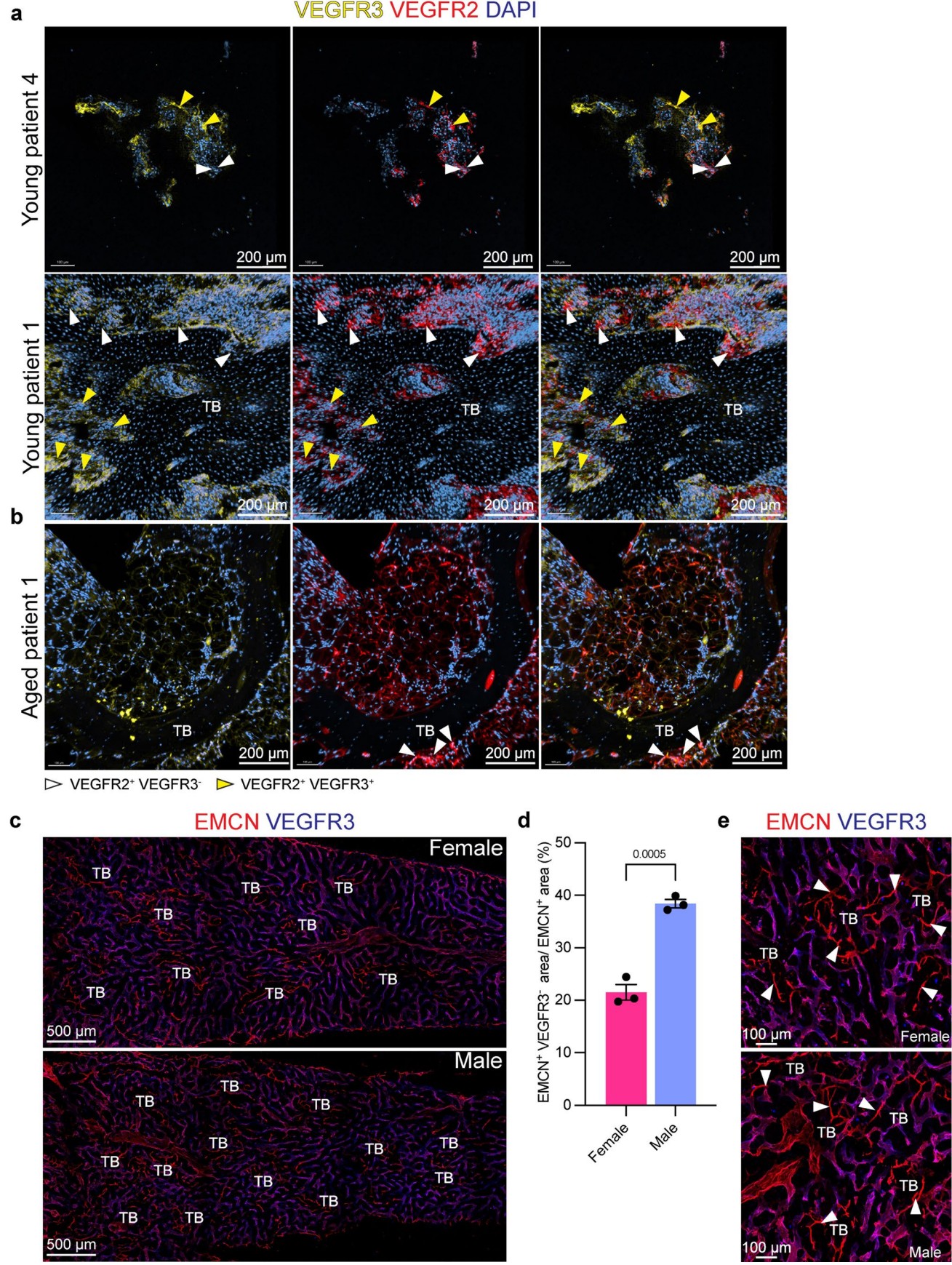

**Extended Data Fig. 5 | See next page for caption.**

**Extended Data Fig. 5 | Features of type R vessels. a, b**. High-resolution confocal images showing VEGFR3 (yellow) and VEGFR2 (red) immunostaining together with DAPI (blue). VEGFR3⁻ VEGFR2⁺ capillaries (white arrowheads) around trabecular bone (TB) and nearby VEGFR3⁺ VEGFR2⁺ vessels (yellow arrowheads) in samples from young (**a**) and aged patients (**b**) are indicated. Scale bars, 200µm. **c**. Tile scan confocal images of EMCN⁺ VEGFR3⁻ ECs in 12-week-old female and male murine femur. Trabecular bone (TB) is indicated. Scale bars, 500µm. **d**. Quantitation of EMCN⁺ VEGFR3⁻ vessel density in 12-week-old female and male femur (n =3 mice per group). Mean ± SEM. P values, unpaired two-tailed t-test. **e**. EMCN (red) and VEGFR3 (blue) immunostaining showing the presence of VEGFR3⁻ EMCN⁺ vessels (white arrowheads) around trabecular bone (TB) in female and male femur. Scale bars, 100µm.

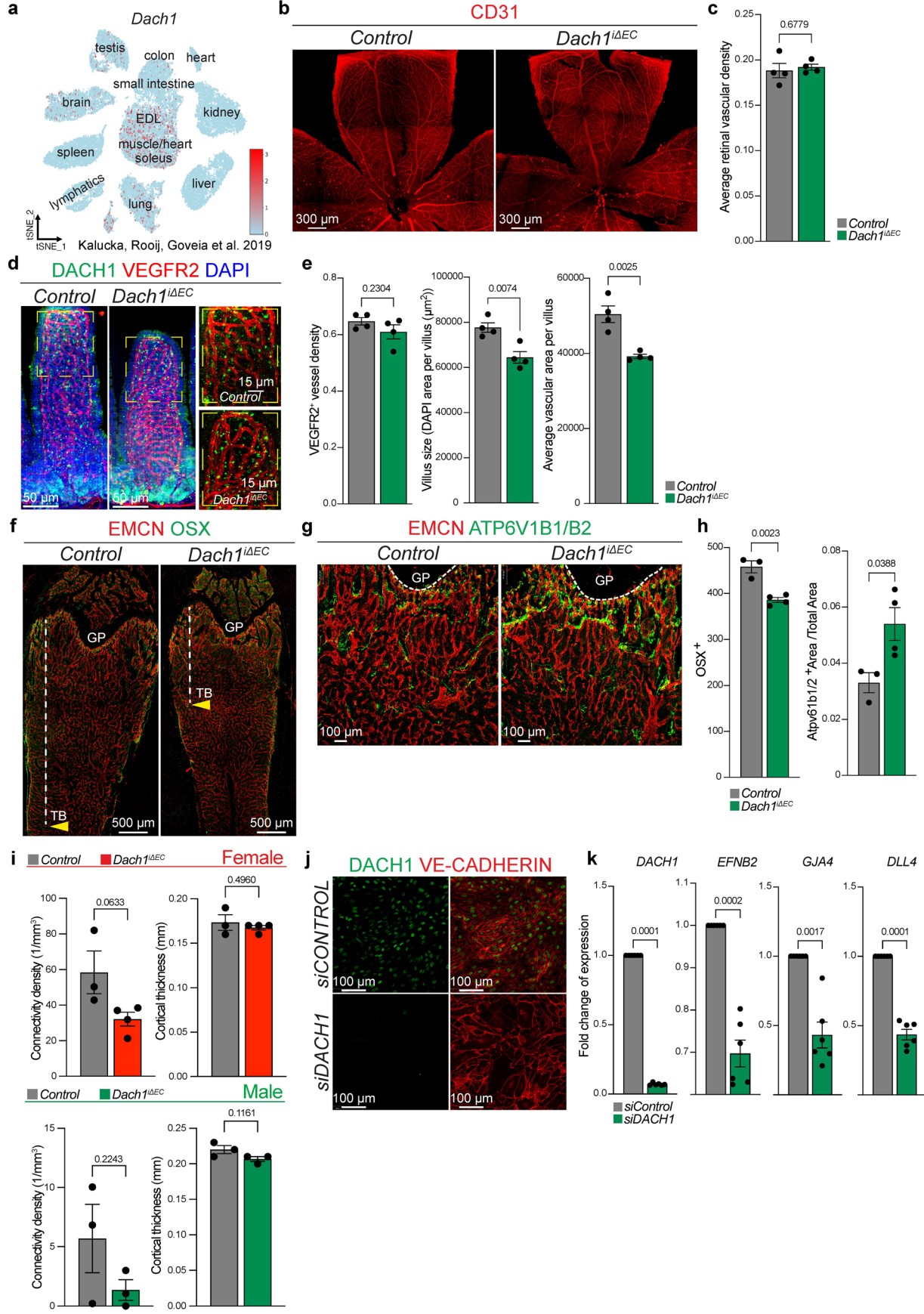

**Extended Data Fig. 6 | See next page for caption.**

**Extended Data Fig. 6 | Endothelial-specific inactivation of *Dach1*. a**, *t*-SNE visualization of *Dach1* gene expression in ECs of different organs. **b**, Confocal tile scan overview images of 12-week-old *Dach1*[iΔEC] and control retina stained for CD31 (red). Scale bars, 300μm. **c**, Quantitation of CD31[+] vessel density in 12-week-old *Dach1*[iΔEC] retinas relative to control (n =4 mice per group). Mean ± SEM. P values, unpaired two-tailed t-test. **d**, High-resolution confocal images of *Dach1*[iΔEC] and control intestinal villi stained for DACH1 (green), VEGFR2 (red), and DAPI (blue). Scale bars, 50μm and insets 15μm. **e**, Quantitation of average VEGFR2[+] vessel density, villus size, and average vascular area per villus in 12-week-old *Dach1*[iΔEC] and littermate control (n =4 mice per group). Mean ± SEM. P values, unpaired two-tailed t-test. **f**, Confocal tile scan overview images of 12-week-old female *Dach1*[iΔEC] and control femur stained for EMCN (red) and OSTERIX (green, OSX). White dashed lines and yellow arrowheads indicate

reduction of region containing trabecular bone (TB) in *Dach1*[iΔEC] mutants. Scale bars, 500μm. **g**. High-magnification images showing EMCN (red) and ATP6V1B1/B2 (green) immunostained female *Dach1*[iΔEC] and control femur. Growth plate (GP) is indicated. Scale bars, 100μm. **h**. Quantitation of OSX[+] cells and ATPV6B1B2 area in control and *Dach1*[iΔEC] mutant (n=3 and 4 mice per group). Mean ± SEM. P values, unpaired two-tailed t-test. **i**. Quantitation shows connectivity density and cortical thickness (per mm), (n = 3-4 female mice per group and n=3 male mice per group). Mean ± SEM. P values, plotted using unpaired two-tailed t-test. **j**. DACH1 (green) co-stained with VE-CADHERIN (red) in *siCONTROL* and *siDACH1* cultured HUVECs. **k**. Downregulation of arterial marker genes in *siDACH1* HUVECs relative to *siCONTROL*. n=6 (3 independent experiments) for each group. Mean ± SEM. P values, unpaired two-tailed t-test with Welch's correction.

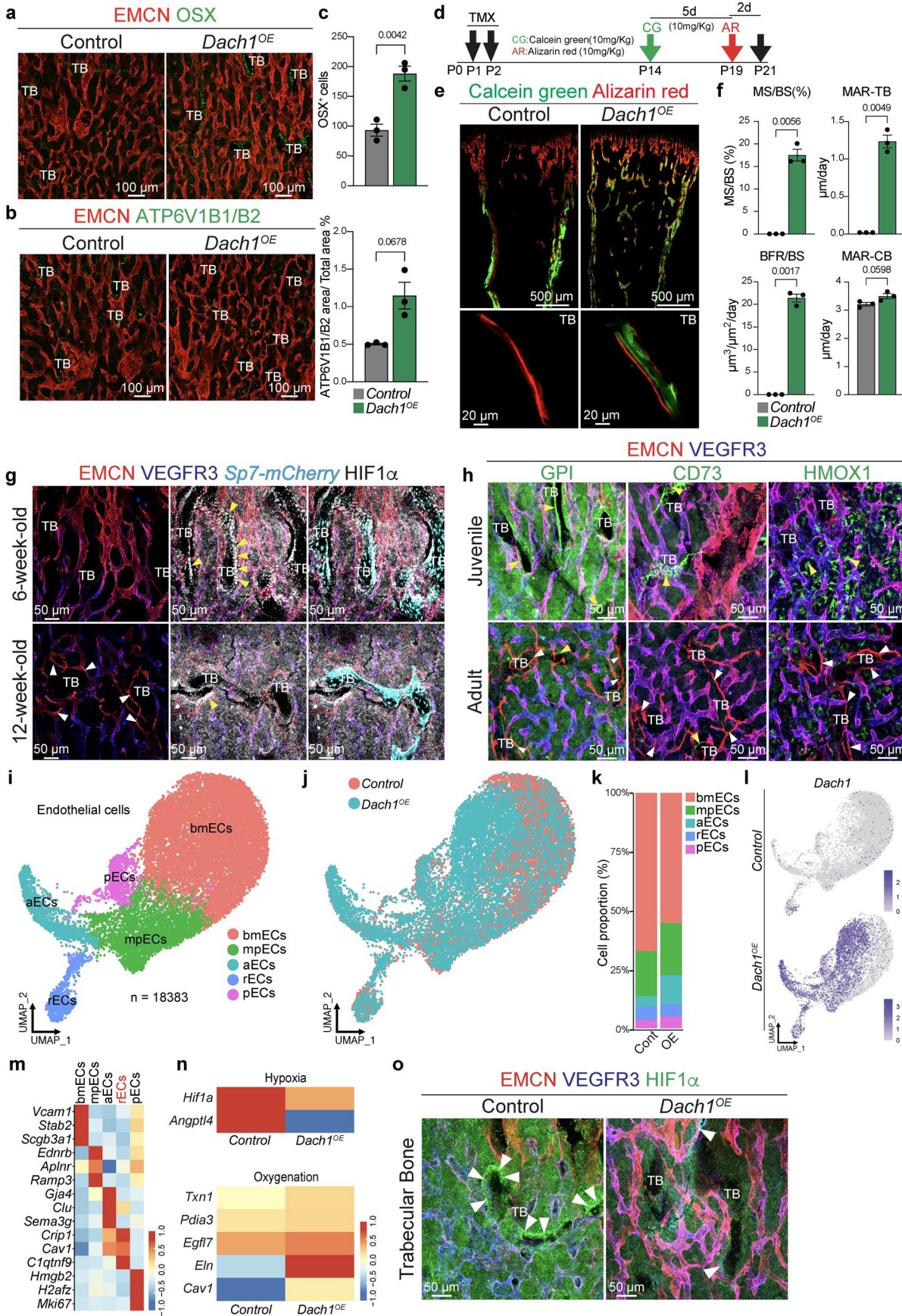

**Extended Data Fig. 7 | See next page for caption.**

**Extended Data Fig. 7 | Effect of DACH1 overexpression in endothelial cells.**
**a, b**. Confocal images of femur sections immunostained for EMCN (red) together with OSTERIX (OSX, green) (**a**) or ATP6V1B1/B2 (green) (**b**). Scale bars, 100μm.
**c**. Quantitation of OSX$^+$ osteoblast-lineage cells and ATP6V1B1/B2$^+$ osteoclasts in *Dach1*$^{OE}$ and control femur (n = 3 in each group). Mean ± SEM. P values, unpaired two-tailed test with Welch's correction. **d-f**. Injection scheme and time points of imaging after labelling of *Dach1*$^{OE}$ and control with Calcein Green (CG) and Alizarin Red (AZ) (**d**) and tile scan imaging of sectioned femur from P30 *Dach1*$^{OE}$ and control after double labelling (**e**). Scale bars, 500μm. Bottom panels in (**e**) show stained trabecular bone. Scale bars, 20μm. **f**. Quantitation showing extent of bone surface actively mineralizing as mineralizing surface/bone surface (MS/BS) in percentage, mineral apposition rate (MAR) as per micrometre/day, and bone formation rate (BFR) (n=3 mice per group). Mean ± SEM. P values, unpaired two-tailed t-test with Welch's correction for MS/BS, MAR-trabecular bone and BFR. **g**. Maximum-intensity projections showing prominent HIF1α$^+$ immunostaining (yellow arrowheads) in 6-week-old *Sp7-mCherry*$^+$ trabecular osteoblasts, whereas HIF1α$^+$ is strongly decreased after emergence of EMCN$^+$ VEGFR3$^-$ type R vessels at 12 weeks. Scale bars, 50 μm. **h**. Maximum-intensity

projections of 12-week-old femurs showing immunostaining for hypoxia-related markers in relation to EMCN$^+$ VEGFR3$^-$ type R vessels. Shown are immunostaining for Glucose-6-Phosphate Isomerase (GPI), 5′-nucleotidase/CD73 and Haem Oxygenase-1 (HMOX1)-expressing macrophages (yellow arrowheads), all of which are elevated near juvenile trabecular bone relative to the equivalent region in adult. White arrowheads mark EMCN$^+$ VEGFR3$^-$ type R vessels. **i,j**. UMAP plots of bone ECs (n=18383) with colour-coded subclusters (**f**), namely sinusoidal bmECs, metaphyseal mpECs, arterial aECs, remodelling rECs, and proliferating pECs. UMAP distribution of *Dach1*$^{OE}$ and control ECs, as indicated by colour (**g**). **k**. Bar plots showing proportion of cells in *Dach1*$^{OE}$ and control samples. **l**. UMAP plots comparing *Dach1* expression in *Dach1*$^{OE}$ (bottom) and control (top) EC subclusters. **m**. Heatmap of differentially expressed genes between EC subclusters. **n**. Heatmap of differentially expressed genes related to hypoxia and, tissue oxygenation in *Dach1*$^{OE}$ and control bone ECs. **o**. Confocal images showing reduced HIF1α immunostaining (white arrowheads) in *Dach1*$^{OE}$ metaphysis and around trabecular bone relative to littermate control. Scale bars, 50μm. Results in panels **i, j, m**, and **n** show the integrated mutant and control scRNA-seq data, whereas panels **k** and **l** show separated samples derived from the integrated data.

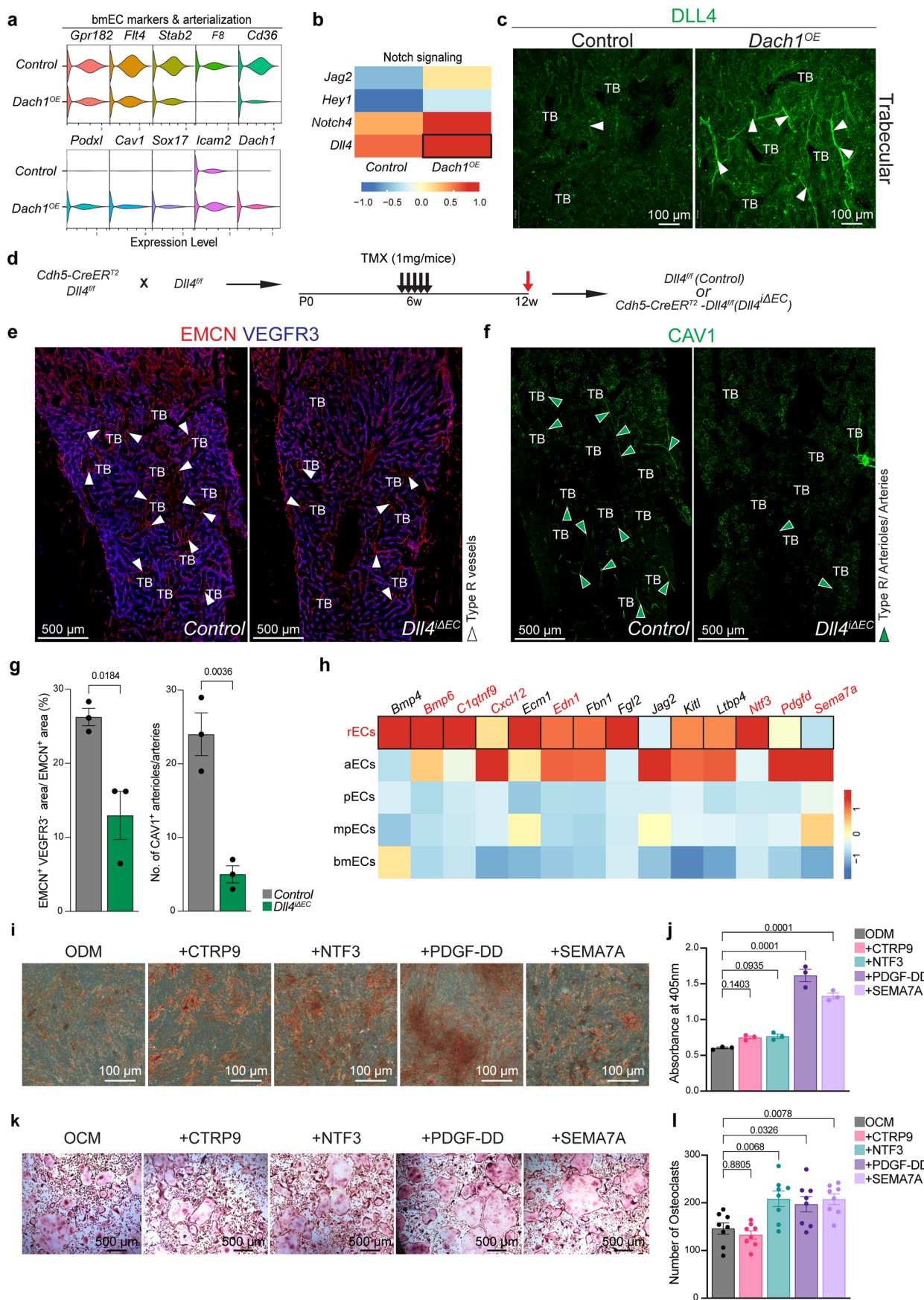

**Extended Data Fig. 8 | See next page for caption.**

**Extended Data Fig. 8 | Molecular signalling and metabolic regulation of rECs.**
**a**. Violin plots showing selected genes related to reduction of bmEC markers and increased arterialization in *Dach1*^OE and control. **b**. Heatmap of differentially expressed genes related to Notch signalling in *Dach1*^OE and control bone ECs. **c**. Representative confocal images showing increased Delta-like 4 (DLL4) (green, white arrowheads) expression in vessels around *Dach1*^OE trabecular bone (TB). Scale bars, 100μm. **d**. Scheme of tamoxifen-induced (TMX) EC-specific *Dll4* inactivation with *Cdh5-CreERT2* line. **e, f**, Tile scan confocal images of EMCN^+ VEGFR3^– ECs (white arrowheads) **(e)** and CAV1^+ aECs (green arrowheads) **(f)** in 12-week-old *Dll4*^iΔEC loss-of-function and control femur. Trabecular bone (TB) is indicated. Scale bars, 500μm. **g**, Quantitation of EMCN^+ VEGFR3^– vessel density (left), and number of CAV1^+ arteries (right) in 12-week-old *Dll4*^iΔEC and control femur (n =3 female mice per group). Mean ± SEM. P values, unpaired two-tailed t-test. **h**. Heatmap showing differentially expressed transcripts for secreted factors in bone EC subpopulations. **i, j**, Representative images **(i)** and quantitation **(j)** of calcified nodules in human mesenchymal stem cells

(HMSC) cultured in osteogenic differentiation medium (ODM) supplemented with 100ng/ml Complement C1q and Tumour Necrosis Factor-Related Protein 9 (CTRP9), Neurotrophin 3 (NTF3), Platelet-Derived Growth Factor D (PDGFD), or Semaphorin 7A (SEMA7A) (n=3 independent experiments for each group). Scale bars, 100μm. Graph shows quantitation of absorbance at 405nm. Mean ± SEM. P values, ordinary one-way ANOVA and plotted with Dunnett's multiple comparisons test. **k, l**, Representative images **(k)** and quantitation **(m)** of multinucleated TRAP+ osteoclasts generated from bone marrow-derived monocytes/macrophages treated with RANKL in osteoclast medium (OCM) supplemented with 100ng/ml CTRP9, NTF3, PDGFD, or SEMA7A n=8 (4 independent experiments) for each group. Scale bars, 500μm. Graph shows quantitation of TRAP^+ multinucleated cells between OCM and treated conditions. Mean ± SEM. P values, ordinary one-way ANOVA and plotted with Dunnett's multiple comparisons test. Panels **a** and **b** show separated conditions derived from the integrated mutant and control data, whereas panel **h** is based on the integrated scRNA-seq data.

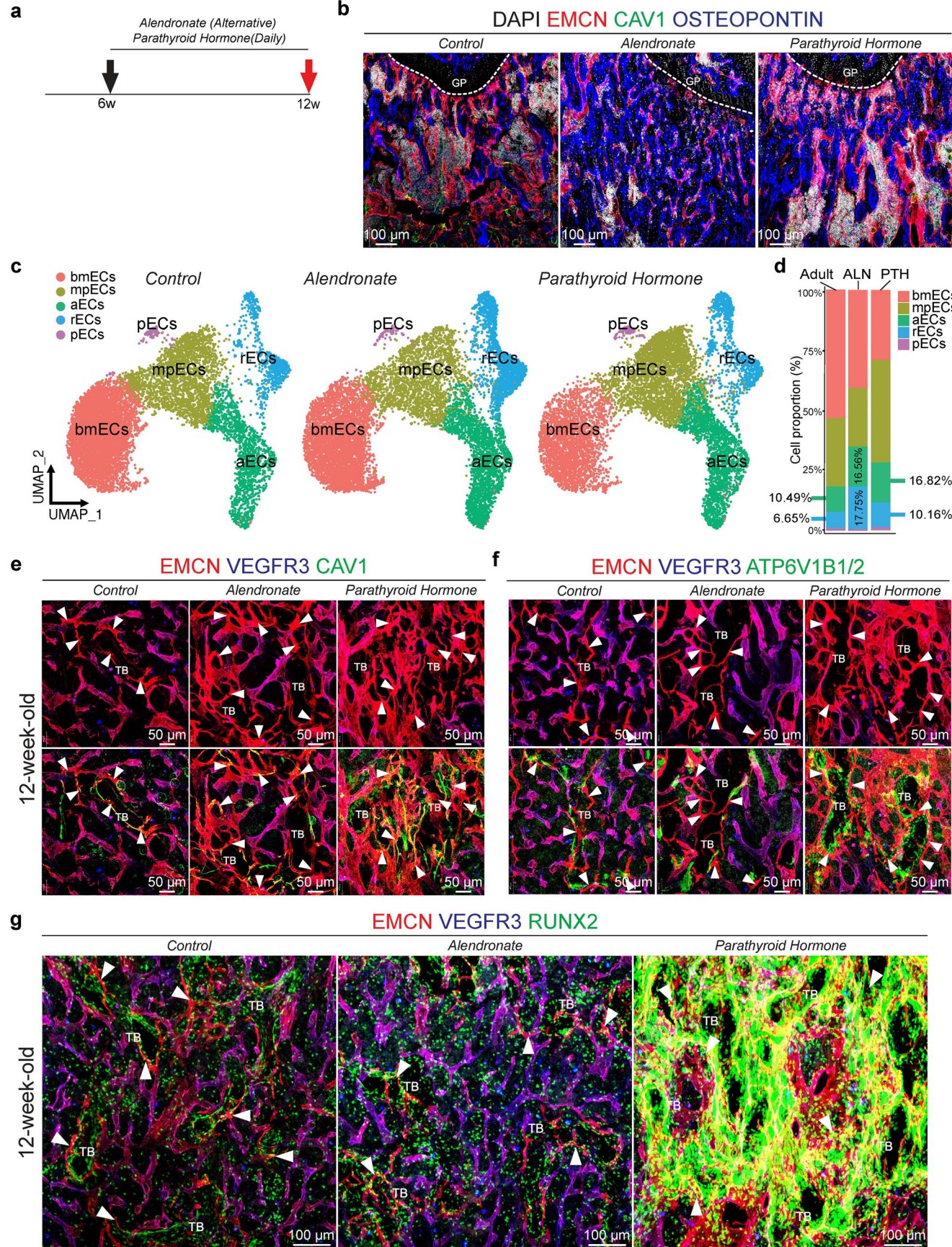

**Extended Data Fig. 9 | See next page for caption.**

**Extended Data Fig. 9 | Modulation of type R vessels by anti-osteoporosis treatments. a**, Scheme for the treatment of adult mice with parathyroid hormone (PTH) or Alendronate. Black arrows indicate onset of treatment, red arrows time points of imaging. **b**, Confocal images of OSTEOPONTIN (blue) immunostaining of the metaphysis near the growth plate (GP) together with EMCN (red), CAV1 (green) and DAPI (white) to show the response to treatments. Scale bars, 100μm. **c**, UMAP visualization of sorted bone ECs from control and treatment groups with colour-coded subclusters. **d**, Bar plots showing the proportions of EC subpopulations in 12-week-old mice. Differences in percentage of rECs and aECs compared to control are indicated. Panels **c** and **d** show the individual samples from an integrated scRNA-seq dataset of three conditions. **e, f**, Confocal images showing EMCN$^+$ VEGFR3$^-$ CAV1$^+$ rECs (white arrowheads) around trabecular bone (TB) (**e**) and association of ATP6V1B1/B2$^+$ (green) osteoclasts with EMCN$^+$ VEGFR3$^-$ type R capillaries (**f**) in 12-week-old control and indicated treatment conditions. Scale bars, 50μm. **g**, Effect of treatments on EMCN$^+$ VEGFR3$^-$ rECs (white arrowheads) and RUNX2$^+$ (green) osteoprogenitors around trabecular bone (TB). Scale bars, 100μm.

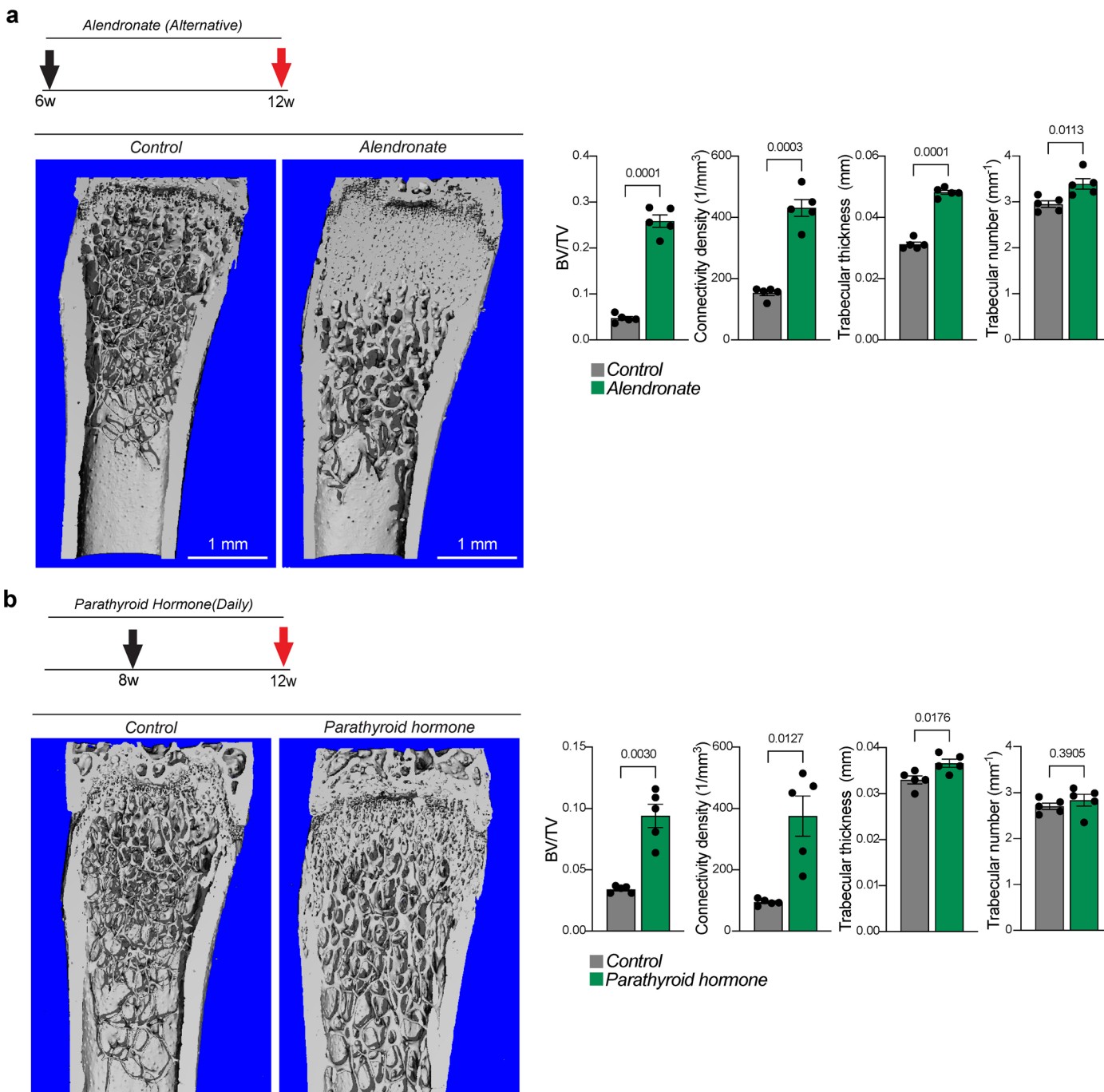

**Extended Data Fig. 10 | Validation of anti-osteoporosis treatments.**
**a, b**, Experimental design, 3D reconstruction of μCT measurements and quantitation of key parameters for the treatment of adult mice with Alendronate (**a**) and Parathyroid hormone (PTH) (**b**). PTH results have been previously reported[18]. Quantitation shows relative bone volume represented as bone volume/tissue volume (BV/TV), connectivity density as the connectivity density per cubic millimetre, trabecular thickness (per mm), and trabecular number represented as number of trabeculae per millimetre (n = 5 mice per group). Mean ± SEM. P values, unpaired two-tailed t-test with Welch's correction for BV/TV and connectivity density for both treatments.

# Reporting Summary

## Statistics

For all statistical analyses, confirm that the following items are present in the figure legend, table legend, main text, or Methods section.

| n/a | Confirmed | |
|---|---|---|
| ☐ | ☒ | The exact sample size (*n*) for each experimental group/condition, given as a discrete number and unit of measurement |
| ☐ | ☒ | A statement on whether measurements were taken from distinct samples or whether the same sample was measured repeatedly |
| ☐ | ☒ | The statistical test(s) used AND whether they are one- or two-sided<br>*Only common tests should be described solely by name; describe more complex techniques in the Methods section.* |
| ☐ | ☒ | A description of all covariates tested |
| ☐ | ☒ | A description of any assumptions or corrections, such as tests of normality and adjustment for multiple comparisons |
| ☐ | ☒ | A full description of the statistical parameters including central tendency (e.g. means) or other basic estimates (e.g. regression coefficient) AND variation (e.g. standard deviation) or associated estimates of uncertainty (e.g. confidence intervals) |
| ☐ | ☒ | For null hypothesis testing, the test statistic (e.g. *F*, *t*, *r*) with confidence intervals, effect sizes, degrees of freedom and *P* value noted<br>*Give P values as exact values whenever suitable.* |
| ☒ | ☐ | For Bayesian analysis, information on the choice of priors and Markov chain Monte Carlo settings |
| ☒ | ☐ | For hierarchical and complex designs, identification of the appropriate level for tests and full reporting of outcomes |
| ☒ | ☐ | Estimates of effect sizes (e.g. Cohen's *d*, Pearson's *r*), indicating how they were calculated |

*Our web collection on statistics for biologists contains articles on many of the points above.*

## Software and code

Policy information about availability of computer code

| Data collection | RNA sequencing by NestSeq500, Confocal imaging were collected from Zeiss LSM780, Yokogawa Confocal microscope and LSM980; Cell sorting and analyzed by FACS Aria II cell sorter with FACSDiva (8.0.2) software (BD Bioscience) |
|---|---|
| Data analysis | No custom software or algorithm was used in this study. All software used in this study for data analysis are either commercially available or open source.<br>Data analysis and statistical tests were done using GraphPad Prism (v8 and v10).<br>Confocal images were analyzed with Volocity 6.3 (Quorum Techonology) or ImageJ2/Fiji (1.53u)<br>FACS data was processed with FlowJo (10.8.1.)<br>GraphPad Prism (v8 and v10) was used for statistical analyses. Figures were generated with Adobe illustrator 2020, 2021, 2024.<br><br>Single cell RNA sequencing data analysis<br>FASTQ-format sequencing raw data was initially processed using UMI-tools (version 1.1.2). The data was then aligned to the mouse reference genome (mm10) using STAR (version 2.7.10a), and quantified with Subread featureCounts (version 2.0.3) to generate an expression matrix. For bone stromal cells, the FASTQ-format sequencing raw data was processed with BD Rhapsody WTA Analysis pipeline (version 1.0) on SevenBridges Genomics online platform (SevenBridges) and expression matrix was used for further data analysis. Data normalization, dimensionality reduction and visualization were performed using Seurat package (version 4.3.0), unless otherwise specified. Plots plotted using Seurat and (https://github.com/SGDDNB/ShinyCell) ShinyCell. |

For manuscripts utilizing custom algorithms or software that are central to the research but not yet described in published literature, software must be made available to editors and reviewers. We strongly encourage code deposition in a community repository (e.g. GitHub). See the Nature Portfolio guidelines for submitting code & software for further information.

## Data

Policy information about availability of data

All manuscripts must include a data availability statement. This statement should provide the following information, where applicable:
- Accession codes, unique identifiers, or web links for publicly available datasets
- A description of any restrictions on data availability
- For clinical datasets or third party data, please ensure that the statement adheres to our policy

Sequencing data that supports the findings of this study have been deposited at Gene Expression Omnibus (GEO, https://www.ncbi.nlm.nih.gov/geo/) under the accession code GSE239627 (https://www.ncbi.nlm.nih.gov/geo/query/acc.cgi?acc=GSE239627). This is a SuperSeries record providing access to all related data, namely bone EC characterization and Dach1OE non-hematopoietic cells. The mouse reference genome (mm10; https://www.ncbi.nlm.nih.gov/datasets/genome/GCF_000001635.20/) was used for mapping reads in this study.
All other information supporting the findings of this study are available within this article or can be obtained from the corresponding author upon request.

## Human research participants

Policy information about studies involving human research participants and Sex and Gender in Research.

| | |
|---|---|
| Reporting on sex and gender | Sex and gender data was not disaggregated due to the small number of samples used in this study. |
| Population characteristics | NA |
| Recruitment | NA |
| Ethics oversight | Collection of human material was conducted with informed consent by patients/parents and with approval from the Swedish Ethical Research Authority and the National Board of Health and Welfare (ethical permissions numbers 2014/276-31/2 and 97-214). |

Note that full information on the approval of the study protocol must also be provided in the manuscript.

# Field-specific reporting

Please select the one below that is the best fit for your research. If you are not sure, read the appropriate sections before making your selection.

☒ Life sciences   ☐ Behavioural & social sciences   ☐ Ecological, evolutionary & environmental sciences

For a reference copy of the document with all sections, see nature.com/documents/nr-reporting-summary-flat.pdf

# Life sciences study design

All studies must disclose on these points even when the disclosure is negative.

| | |
|---|---|
| Sample size | No specific statistical methods were used to predetermine sample size. Sample size were chosen based on previous experience. (ref. Kusumbe et al. Nature 2014; Ramasamy et al., Nature 2014; Sivaraj et al., Elife 2020; Sivaraj et al., Cell Report 2021). |
| Data exclusions | To ensure data quality, cells were filtered based on the criteria: having a number of genes per cell (nFeature_RNA) between 500 and 6000, and a percentage of mitochondrial genes (percent.mito) less than 25. Additionally, genes were filtered to include only those present in a minimum of 3 cells. After filtering, the matrices were normalized using the NormalizeData function with the LogNormalize method and a scale factor of 10,000. Variable genes were identified using the FindVariableFeatures function, selecting the top 2000 genes with the variance stabilizing transformation (VST) method, while also excluding genes related to the cell cycle (GO:0007049). |
| Replication | All experiments were repeated at least three times with an exception for scRNAseq experiments. All other experiments performed independently to ensure reproducibility. All the attempts of replication experiments were successful.<br>scRNA-seq experiments were performed once with pooled animals for each age and treatment group. Integration of these samples led to robust and reliable results across conditions. |
| Randomization | No formal method of randomization was used. All experiments involving wildtype mice were performed using C57Bl6 strain female mice. For mutant studies we used both male and female of same age group, and phenotype were always compared between same age and sex of animals. scRNAseq of 3-week-old bone endothelial cells and Dach1OE scRNAseq utilized both male and female mice. |
| Blinding | Blinding was not be feasible due to logisitic requirements as well as cost and personnel constraints. |

# Reporting for specific materials, systems and methods

We require information from authors about some types of materials, experimental systems and methods used in many studies. Here, indicate whether each material, system or method listed is relevant to your study. If you are not sure if a list item applies to your research, read the appropriate section before selecting a response.

## Materials & experimental systems

| n/a | Involved in the study |
|---|---|
| ☐ | ☒ Antibodies |
| ☐ | ☒ Eukaryotic cell lines |
| ☒ | ☐ Palaeontology and archaeology |
| ☐ | ☒ Animals and other organisms |
| ☒ | ☐ Clinical data |
| ☒ | ☐ Dual use research of concern |

## Methods

| n/a | Involved in the study |
|---|---|
| ☒ | ☐ ChIP-seq |
| ☐ | ☒ Flow cytometry |
| ☒ | ☐ MRI-based neuroimaging |

## Antibodies

**Antibodies used**

All antibody details (clone and manufacturer) are included in methods and also listed below:
Antibody Manufacturer Catalog number Concentration
Primary antibodies used: rat monoclonal anti-Endomucin (V.7C7) (Santa Cruz, sc-65495, 1:100),
goat polyclonal anti-VEGFR3 (R&D Systems, AF743, 1:50),
rabbit polyclonal anti-Cav1 (Cell Signaling, 3238, 1:50),
chicken polyclonal anti-GFP (Abcam, ab13970, 1:200),
rabbit polyclonal anti-Osterix (Abcam, ab22552, 1:300),
rabbit monoclonal anti-Runx2 EPR14334 (Abcam, ab192256, 1:200),
rabbit monoclonal anti-Ki67 (Cell Signaling, 12202, 1:200),
rabbit monoclonal anti-vATPase (Abcam, ab200839, 1:200),
rabbit polyclonal anti-Collagen IV (AbD Serotec, 2150-1470, 1:100),
rabbit polyclonal anti-Collagen IIIA1 (Abcam, ab7778, 1:100),
goat polyclonal anti-Hif1a (R&D Systems, AF1935, 1:100),
rabbit polyclonal anti-Hif1a (Santa Cruz, sc-10790, 1:100),
rabbit monoclonal anti-GPI (Cell Signaling, 94068, 1:200),
rabbit monoclonal anti-CD73 (Cell Signaling, 13160, 1:200),
rabbit monoclonal anti-Hmox1 (Cell Signaling, 86806, 1:200),
rabbit polyclonal anti-Dach1 (Proteintech, 10914-1-AP, 1:200),
rabbit polyclonal anti-NG2 (Millipore, AB5320, 1:200),
goat polyclonal anti-Cxadr (R&D Systems, AF2654, 1:50),
rabbit monoclonal anti-Sox11 (Abcam, ab134107, 1:800),
rat monoclonal anti-Madcam1 (Abcam, ab80680, 1:100),
rabbit polyclonal anti-Fabp4 (Abcam, ab13979, 1:100),
mouse monoclonal anti-Mct4 (Santa Cruz, AF647, 1:100),
rabbit monoclonal anti-Glut1 (Cell Signaling, 12939, 1:100),
goat polyclonal anti-Dll4 (R&D Systems, AF1389, 1:50),
chicken polyclonal anti-Mct1 (Millipore, AB1286-I, 1:100),
rabbit polyclonal anti-Dach1 (Proteintech, 10914-1-AP, 1:500),
goat polyclonal anti-Cdh5 (R&D Systems, AF938, 1:100),
rat anti-CD31 (BD Pharmingen, 553370; 1:100),
goat anti-VEGFR2 (R&D Systems, AF644; 1:100),
Mouse anti-VEGFR3 (R&D Systems, MAB3491, 1:100),
Goat anti-VEGFR2 (R&D Systems, AF357; 1:100),
Anti-rat Alexa Fluor 488 (ThermoFisher Scientific, A21208, 1:100-1:300),
anti-rabbit Alexa Fluor 488 (ThermoFisher Scientific, A21206, 1:100-1:300),
anti-chicken Alexa Fluor 488 (Jackson Laboratories, 703-545-155, 1:100-1:300),
anti-goat Alexa Fluor 546 (ThermoFisher Scientific, A11056, 1:100-1:300),
anti-rabbit Alexa Fluor 546 (ThermoFisher Scientific, A10040, 1:100-1:300),
anti-rat Alexa Fluor 594 (ThermoFisher Scientific, A21209, 1:100-1:300),
anti-rabbit Alexa Fluor 594 (ThermoFisher Scientific, A21207, 1:100-1:300),
anti-goat Alexa Fluor 594 (ThermoFisher Scientific, A11058, 1:100-1:300),
anti-goat Alexa Fluor 647 (ThermoFisher Scientific, A21447, 1:100-1:300),
anti-rabbit Alexa Fluor 647 (ThermoFisher Scientific, A31573, 1:100-1:300),
anti-rat Alexa Fluor 647 (Jackson Laboratories, 712-605-153, 1:100-1:300),
anti-mouse Alexa Fluor 488 (Thermo Fisher Scientific, A21202, 1:500),
Cy™3 AffiniPure™ Goat Anti-Mouse IgG (Jackson ImmunoResearch, 115-165-003; 1:400),
anti-goat Alexa-647 (ThermoFisher Scientific, A32849; 1:400),
anti-rabbit Alexa Fluor 546 (Thermo Fisher Scientific, A10040, 1:500) for HUVECS
DAPI

**Validation**

All antibodies used in the study have been commercially available and previously used by our group (ref. kusumbe et al. Nature 2014;

| Validation | Ramasamy et al., Nature 2014; Sivaraj et al., Elife 2020; Sivaraj et al., Cell Report 2021). The complete information and all validation information for each Ab as well as previous publications that have used each Ab can be found on the manufacturer's website. |

# Eukaryotic cell lines

Policy information about cell lines and Sex and Gender in Research

| Cell line source(s) | primary Human Umbilical Vein Endothelial Cells (HUVECs) : Provitro<br>primary Human Mesenchymal Stem Cells (HMSCs) (Lonza, PT-2501) |
| Authentication | Authentication was performed for each cell lot by vendors and cell identity was confirmed by antibody staining of cultured cells. |
| Mycoplasma contamination | Mycoplasma contamination test results were negative. |
| Commonly misidentified lines<br>(See ICLAC register) | There is no commonly misidentified lines. |

# Animals and other research organisms

Policy information about studies involving animals; ARRIVE guidelines recommended for reporting animal research, and Sex and Gender in Research

| Laboratory animals | All animals used in this study are Mus musculus species, C57/BL6 background strain independent of genotype.<br><br>Flt4-CreERT2 lineage tracing experiments were performed at Uppsala University and approved by the Uppsala Laboratory Animal Ethical Committee. The remaining animals were housed in an animal facility at the Max Planck Institute for Molecular Biomedicine in specific pathogen-free conditions in individually ventilated cages (IVC) with a consistent light/dark cycle, free access to food and water, and controlled temperature and humidity. Mice were routinely genotyped by PCR using allele-specific primers.<br><br>Transgenic mice were generated from our laboratory, Cdh5(PAC)-CreERT2, Bmx-CreERT2 & Cdh5-mTomato-nGFP; Sp7-mcherry mice from Jackson Laboratory; Efnb2-H2BGFP mice from Philippe Soriano Laboratory; Aplnr-CreERT2, Rosa26-Dach1OE, Dach1 flox/flox mice from the Red-Horse Laboratory; Flt4-CreERT2 mice from Sagrario Ortega Laboratory; Dll4 flox/flox mice from Freddy Radtke Laboratory; Rosa26-mTG mice from Jackson Laboratory. 3-week old, 6-week old, 12-week old and 75-week-old mice were used for most of the experiments. Sex of the mice is indicated wherever necessary.<br><br>For mutant experiments, mice were bred to Aplnr-CreERT2 or Cdh5(PAC)CreERT2 to generate inducible mouse model. Cre negative were used as litter mate control.  Most of the experiments were performed at P30 or 12 weeks.<br><br>All animal experiments were conducted following the 3Rs (replacement, reduction, and refinement) and in accordance with institutional guidelines and laws, following protocols (84-02.04.2016.A160, 81-02.04.2018.A171, 81-02.04.2020.A212, 81-02.04.2020.A416 and 81-02.04.2022.A198) approved by the Landesamt für Natur, Umwelt, and Verbraucherschutz of North Rhine-Westphalia, Germany. |
| Wild animals | No wild animals were used in the study. |
| Reporting on sex | Female mice were used for most of the experiments. Male and female mice were used in 3-week-old bone endothelial scRNAseq and Dach1OE scRNAseq at Postnatal day 30. |
| Field-collected samples | No field collected samples were used. |
| Ethics oversight | All the animal experiments described in this study were conducted following the 3Rs (replacement, reduction, and refinement). Mice were housed in specific pathogen-free conditions in individually ventilated cages (IVC) with a consistent light/dark cycle, food, water, and controlled temperature. Experiments were performed at the MPI for Molecular Biomedicine in accordance with institutional guidelines and laws, following protocols (84-02.04.2016.A160, 81-02.04.2018.A171, 81-02.04.2020.A212, 81-02.04.2020.A416 and 81-02.04.2022.A198) approved by the Landesamt für Natur, Umwelt, and Verbraucherschutz (LANUV) of North Rhine-Westphalia, Germany. Flt4-CreERT2 lineage tracing experiments were performed at Uppsala University and approved by the Uppsala Laboratory Animal Ethical Committee. |

Note that full information on the approval of the study protocol must also be provided in the manuscript.

# Flow Cytometry

## Plots

Confirm that:

☒ The axis labels state the marker and fluorochrome used (e.g. CD4-FITC).

☒ The axis scales are clearly visible. Include numbers along axes only for bottom left plot of group (a 'group' is an analysis of identical markers).

☒ All plots are contour plots with outliers or pseudocolor plots.

☒ A numerical value for number of cells or percentage (with statistics) is provided.

## Methodology

Sample preparation

All sample preparation methods are written in the method section and available here.

To enrich for bone endothelial cells, single-cell suspensions were prepared from Cdh5-mT/nG reporter femurs and tibias. First, the surrounding muscle tissue was carefully removed and the epiphysis was detached from both the femur and tibia. Next, the cleaned samples were then collected in a solution of collagenase types I and IV (2 mg/ml). The bones were cut into small pieces and crushed using mortar and pestle. After 20 min of digestion at 37 °C, the crushed samples were transferred to 70μm strainers in 50ml tubes to obtain a single–cell suspension.

The cell suspension was resuspended in blocking solution (1% BSA and 1mM EDTA in PBS without Ca2+/Mg2+), followed by centrifugation at 300×g for 5 min. The cells were washed three times with ice-cold blocking solution and filtered through a 50μm strainers. Subsequently, the cells were resuspended in an appropriate volume of blocking solution. To expedite FACS of endothelial cells, lineage depletion was performed using a lineage cell depletion kit (Miltenyi Biotec, 130-090-858), following the manufacturer's instructions. Lineage-depleted cells were resuspended in 0.5% BSA in PBS, and cell sorting was performed using a FACS Aria II cell sorter (BD Biosciences). Sorted GFP-tomato double-positive cells were collected in a 0.05% blocking buffer for scRNA-seq.

Instrument

FACS Aria II cell sorter (BD Bioscience)

Software

FACSDiva (8.0.2, BD Bioscience) and FlowJo (10.8.1) were used for sorting and analysis.

Cell population abundance

Total cell counts were measured with cell counter.

Gating strategy

Single viable cells were gated initally forward and side scatter.

☒ Tick this box to confirm that a figure exemplifying the gating strategy is provided in the Supplementary Information.

