## [Peer Review File · Nature Cell Biology]

Peer Review Information

Journal: Nature Cell Biology

Manuscript Title: Specialized post-arterial capillaries facilitate adult bone remodeling

Corresponding author name(s): Professor Ralf Adams

Editorial Notes:

Reviewer Comments & Decisions:

Decision Letter, initial version:

*Please delete the link to your author homepage if you wish to forward this email to co-authors.

Dear Professor Adams,

Your manuscript, "Specialized post-arterial capillaries facilitate bone remodeling", has now been seen by 3 referees, who are experts in capillaries, bone, scRNAseq (referee 1); bone formation, skeletal endothelium, transcriptional analysis (referee 2); and bone formation and remodeling (referee 3). As you will see from their comments (attached below) they overall find this work of potential interest, but

have raised substantial concerns, which in our view would need to be addressed with considerable revisions before we can consider publication in Nature Cell Biology.

Nature Cell Biology editors discuss the referee reports in detail within the editorial team, including the chief editor, to identify key referee points that should be addressed with priority, and requests that are overruled as being beyond the scope of the current study. To guide the scope of the revisions, I have listed these points below. I should stress that the referees' concerns overall point to a premature dataset and these points would need to be addressed with experiments and data, and reconsideration of the study for this journal and re-engagement of referees would depend on strength of these revisions.

In particular, it would be essential to:

(A) Carry out additional experiments to further test current claims, clarify data interpretation, and confirm the robustness of the models used throughout the study, as indicated by:

Referee #1:

"(Extended Fig. 4a) These images are hard to interpret. If the authors want to claim the merge of Sp7-mCherry and HIF1a, assigned colors should be changed at least".

"(Fig. 7d) Why rECs are not increased in Dach1-OE should be explained, as this data is inconsistent with immunostaining (Fig. 6f). Do they shift into the dEC cluster? If so, do they retain the feature of Emcn+Flt4-?"

Referee #2:

"The sites of activity of Alpnr-creERT2 activity should be more clearly determined with a reporter for cre activity (such as the mTnG system used elsewhere here). Data from these methods should then be correlated and compared with the various cellular phenotypes and skeletal metabolism phenotypes presented. It is essential that it be robustly confirmed that the Dach1 transgene is only activated in endothelial cells and not other skeletal cell types. Data in fig 6b are seen as not adequate to address this important point, as many cells are DACH1 positive and the difference between the Dach1OE and control mice DACH1 staining is only a minor difference of degree and does not currently make a compelling case for the specificity of the overexpression effect within endothelial cells".

"The Dach1OE experiment is interesting, however it would appear from the scRNA-seq study that the phenotype is likely to be driven by dECs, since this appears to be the most strongly expanded

population. Since dECs do not appear to correspond to type R endothelium, this dilutes the argument that this data implicates type R endothelium in bone metabolism. More broadly, the impact of Dach1 overexpression, some of which may be ectopic in cell types not normally expressing Dach1 (see comment #1 above) can be non-physiologic and difficult to interpret. This weakness could be effectively mitigated if the Dach1 loss-of-function showed a clearly supportive phenotype compatible with the Dach1OE result, but this doesn't appear to be the case in the currently presented data (see main issue #5 below)".

"Both the Dach1 gain-of-function and loss-of-function mouse models appear to invoke changes in multiple endothelial populations. While this data remains interesting for further establishing the function of Dach1 and for further elucidating endothelial to osteoblast-lineage functional connections, it does not appear to make a strong case for the function of type R endothelial cells given that many endothelial subsets are impacted. From this perspective, it is unclear how this data fits into the broader line of investigation focused on type R cells that comes before and after this data".

"The Dach1 loss of function model is very incompletely described with only immunostaining data. What is the phenotype on uCT and regular histology? The phenotype associated with Dach1 loss-of-function is likely the very most important data to link type R cells to bone metabolism and therefore at the conceptual heart of this work, so is concerning that this data is very cursory, does not demonstrate a compelling phenotype, and is relegated to the supplemental figures".

Referee #3:

"I may have misunderstood the alendronate experiment, but was surprised to see an increase in the fraction of type R endothelium in a setting of profound decrease of bone formation and resorption caused by alendronate. Some exploration of that would be useful".

(B) Dissect the mechanistic underpinnings and carry out experiments to test the functional importance of type R endothelial cells, as indicated by referees #2 and #3. We would like to stress that we find it necessary that you provide experimental evidence in this direction.

Referee #2 says:

"However, the key weakness of this report is that there is no causative data that establish the function or importance of type R endothelial cells or mechanisms by which they interact with osteoblast-lineage cells. In particular, the two key experiments in this manuscript are the Dach1 overexpression and loss-of-

function models. However, it appears that many skeletal endothelial cell types are impacted, including cell types previously described by the authors as impacting bone metabolism, so it is difficult to specifically ascribe any phenotypes observed to type R cells. Moreover, the phenotypes seen with Dach1 gain and loss of function appear to be very modest and only superficially studied in the present data. No other data is presented to study a mechanism by which type R cells may impact bone metabolism. These major limitations appear to also be appreciated and acknowledged by the authors given the wording used in the discussion. These limitations result in the present manuscript not being able to make a clear case for the importance or functional relevance of type R endothelial cells".

Referee #3 says:

"The data are extensive and novel, though do not attempt to establish the mechanisms of the links between the changes in endothelium and osteoblasts/clasts. Effects of changes of oxygenation and local gene expression remain to be sorted out with mechanistic experiments perhaps in the future".

(C) All other referee concerns pertaining to strengthening existing data, providing controls, methodological details, clarifications and textual changes, should also be addressed.

(D) Finally please pay close attention to our guidelines on statistical and methodological reporting (listed below) as failure to do so may delay the reconsideration of the revised manuscript. In particular please provide:

We would be happy to consider a revised manuscript that would satisfactorily address these points, unless a similar paper is published elsewhere, or is accepted for publication in Nature Cell Biology in the

meantime.

- ensure that it conforms to our format instructions and publication policies (see below and www.nature.com/nature/authors/).
- provide a point-by-point rebuttal to the full referee reports verbatim, as provided at the end of this letter.
- provide the completed Editorial Policy Checklist (found here <https://www.nature.com/authors/policies/Policy.pdf>), and Reporting Summary (found here <https://www.nature.com/authors/policies/ReportingSummary.pdf>). This is essential for reconsideration of the manuscript and these documents will be available to editors and referees in the event of peer review. For more information see <http://www.nature.com/authors/policies/availability.html> or contact me.

Nature Cell Biology is committed to improving transparency in authorship. As part of our efforts in this direction, we are now requesting that all authors identified as 'corresponding author' on published papers create and link their Open Researcher and Contributor Identifier (ORCID) with their account on the Manuscript Tracking System (MTS), prior to acceptance. ORCID helps the scientific community achieve unambiguous attribution of all scholarly contributions. You can create and link your ORCID from the home page of the MTS by clicking on 'Modify my Springer Nature account'. For more information please visit www.springernature.com/orcid.

[Redacted]

We would like to receive a revised submission within six months. We would be happy to consider a revision even after this timeframe, however if the resubmission deadline is missed and the paper is eventually published, the submission date will be the date when the revised manuscript was received.

We hope that you will find our referees' comments, and editorial guidance helpful. Please do not hesitate to contact me if there is anything you would like to discuss.

Best wishes,

Stelios

Stylianos Lefkopoulos, PhD
He/him/his
Associate Editor
Nature Cell Biology
Springer Nature
Heidelberger Platz 3, 14197 Berlin, Germany

E-mail: stylianos.lefkopoulos@springernature.com

Twitter: @s_lefkopoulos

Reviewers' Comments:

Reviewer #1:

Remarks to the Author:

Bone marrow endothelial cells (BMECs) are a topic of broad interest in various fields including hematology, vascular biology, orthopedics, and regenerative medicine. Nevertheless, compared to endothelial cells in other vascular beds, much less is known about BMECs due to the technical difficulty related to histological analysis of bones.

In the last decade the authors' group have developed a new histological technique of bones and established a landmark concept for the heterogeneity of BMECs, particularly in embryonic and juvenile stages.

The present study intensively studied BMECs in adult stage after completion of bone development, and uncovered a specialized BMECs located in post-arterial capillaries, named as "type R ECs".

Mechanistically the authors showed this new EC subtype contributes to bone remodeling depending on a transcription factor Dach1. In the technical viewpoint, the single cell data is based on a robust number of 62677 ECs, much larger than that of previous reports, which make the data very convincing. The images shown in this paper is of extraordinary quality, and the quantification is solid.

Overall, this paper clearly enhances our understanding of the mechanism, how blood vessels maintain bones as well as the heterogeneity of BMECs. This reviewer only has some minor comments:

1. (Title) This reviewer recommends “adult” should be included somewhere in the title to differentiate from well-studied developmental analysis.
2. (Fig. 1c) Color assignment should be improved. To this reviewer, juvenile and adult cells are hard to be distinguished.
3. (Fig. 1e) What about the expression of tip cell markers like *Esm1*, *Nid2*, and *Angpt2*? If they are enriched in type R ECs, it makes sense as type R ECs look extending lots of filopodia (Fig. 3e).
- 4 (Fig. 1f) It will be helpful if the authors provide the list of marker genes at least in type R ECs (not just top 4) as a supplemental file.
5. (Fig. 1g) The lack of lymphatic markers should be discussed more as it directly conflicts with a recent paper (Biswas et al., Cell 2023). Is it possible to show some positive control in this heatmap such as small contamination of hematopoietic cells known to express *Prox1* and *Lyve1*?
6. (Fig. 3g-j; Extended Fig. 2c-e) Lineage tracing experiments using *Bmx-CreERT2* and *Flt4-CreERT2* show type R ECs are derived from sinusoidal ECs. Could this finding be confirmed by pseudo-time analysis of single cell RNA-seq data?
7. (Extended Fig. 4a) These images are hard to interpret. If the authors want to claim the merge of Sp7-mCherry and HIF1a, assigned colors should be changed at least.
8. (Fig. 6a, b) Why the authors focused on *Dach1* should be explained more. Is that derived from the list of marker genes characterizing type R cells?
9. (Fig. 7d) Why rECs are not increased in *Dach1*-OE should be explained, as this data is inconsistent with immunostaining (Fig. 6f). Do they shift into the dEC cluster? If so, do they retain the feature of *Emcn+Flt4*-?
10. (Extended Fig. 6) In this reviewer’s view, this loss-of-function data is a key for the entire manuscript, so should be moved to main figures. In contrast, hypoxia data without any functional analysis (Fig. 5) could be moved to Extended figures.

Reviewer #2:

Remarks to the Author:

In this report, Mohanakrishnan et al. analyze a very large scRNA-seq dataset focused on skeletal

endothelial cells, most notably finding a population corresponding to post-arteriolar capillaries they term type R cells. Based on these cells expressing Dach1, Dach1 conditional gain- and loss-of-function mice are studied and found to display alterations in skeletal endothelial cells and also bone metabolism. Lastly, treatment with agents relevant to bone physiology (e.g., PTH) is seen to alter the amount of type R endothelial cells. Overall, the question of how skeletal endothelium controls bone formation is very important, underexplored, and appropriate in topic and scope for Nature Cell Biology. There are some important concerns about the site of activity of the *Alpnr-creERT2* system and concerns that the key Dach1 loss and gain of function models are not studied in adequate depth and are lacking basic phenotypic information. Otherwise these studies are well done from a technical perspective. However, the key weakness of this report is that there is no causative data that establish the function or importance of type R endothelial cells or mechanisms by which they interact with osteoblast-lineage cells. In particular, the two key experiments in this manuscript are the Dach1 overexpression and loss-of-function models. However, it appears that many skeletal endothelial cell types are impacted, including cell types previously described by the authors as impacting bone metabolism, so it is difficult to specifically ascribe any phenotypes observed to type R cells. Moreover, the phenotypes seen with Dach1 gain and loss of function appear to be very modest and only superficially studied in the present data. No other data is presented to study a mechanism by which type R cells may impact bone metabolism. These major limitations appear to also be appreciated and acknowledged by the authors given the wording used in the discussion. These limitations result in the present manuscript not being able to make a clear case for the importance or functional relevance of type R endothelial cells.

Major points:

1. The sites of activity of *Alpnr-creERT2* activity should be more clearly determined with a reporter for cre activity (such as the mTnG system used elsewhere here). Data from these methods should then be correlated and compared with the various cellular phenotypes and skeletal metabolism phenotypes presented. It is essential that it be robustly confirmed that the Dach1 transgene is only activated in endothelial cells and not other skeletal cell types. Data in fig 6b are seen as not adequate to address this important point, as many cells are DACH1 positive and the difference between the Dach1OE and control mice DACH1 staining is only a minor difference of degree and does not currently make a compelling case for the specificity of the overexpression effect within endothelial cells.
2. The Dach1OE experiment is interesting, however it would appear from the scRNA-seq study that the phenotype is likely to be driven by dECs, since this appears to be the most strongly expanded population. Since dECs do not appear to correspond to type R endothelium, this dilutes the argument that this data implicates type R endothelium in bone metabolism. More broadly, the impact of Dach1 overexpression, some of which may be ectopic in cell types not normally expressing Dach1 (see comment #1 above) can be non-physiologic and difficult to interpret. This weakness could be effectively mitigated if the Dach1 loss-of-function showed a clearly supportive phenotype compatible with the Dach1OE result, but this doesn't appear to be the case in the currently presented data (see main issue #5 below).

3. Is there a cortical phenotype in Dach1OE mice? The trabecular phenotype presented appears to be modest to moderate at best. Formal quantitative histomorphometry to determine rates of bone formation and analysis of osteoclast numbers/activity is important for both the Dach1 gain and loss of function mice.
4. Both the Dach1 gain-of-function and loss-of-function mouse models appear to invoke changes in multiple endothelial populations. While this data remains interesting for further establishing the function of Dach1 and for further elucidating endothelial to osteoblast-lineage functional connections, it does not appear to make a strong case for the function of type R endothelial cells given that many endothelial subsets are impacted. From this perspective, it is unclear how this data fits into the broader line of investigation focused on type R cells that comes before and after this data.
5. The Dach1 loss of function model is very incompletely described with only immunostaining data. What is the phenotype on uCT and regular histology? The phenotype associated with Dach1 loss-of-function is likely the very most important data to link type R cells to bone metabolism and therefore at the conceptual heart of this work, so is concerning that this data is very cursory, does not demonstrate a compelling phenotype, and is relegated to the supplemental figures.

Minor points:

1. MACS-based cell depletion as employed in the scRNA-seq study can remove cell types of interest. Gating used for the cell sorting should be displayed. It is unlikely that the MACS depletion method used fully removed hematopoietic cells. Similarly, it is possible that CDH5 cre may label non-endothelial populations, though it is appreciated that the authors have used CDH5-based lineage reporters extensively. Were cells filtered from the scRNA-seq data prior to clustering? If so, what criteria were applied?
2. EMCN is also known to label osteoblast-lineage cells. Is this observed in the EMCN immunofluorescence staining? Lower power views of the IF staining in Fig 2 would be helpful for evaluation.
3. Claims that type R vessels are in proximity to osteoblasts and osteoclasts would be more convincing if analyzed in a more systematic manner than the example images in Fig 4 and performed comparing type H or other vessel subsets to type R endothelial cells.
4. For analysis of Xbp1 expression on page 8, it is important to distinguish between spliced and unspliced Xbp1. To this reviewer's knowledge it is Xbp1 splicing and not basal expression that is associated with hypoxia.
5. The analysis of hypoxia associated markers on page 8 does not demonstrate a critical role for type R endothelium in regulating local oxygen tension (though it is appreciated that this is obviously the primary function of vascular endothelium). Instead, this data would seem to at most provide indirect evidence that these cells reside in a hypoxic environment. Similarly, the data on page 8 assigning a role for type R cells in bone turnover on the basis of the transcripts presented is interpreted too strongly. All of this data would be more properly presented as hypothesis-generating and should be better qualified.

Similar issues apply to the discussion.

6. In the scRNA-seq study on page 9-10, was the DACH1 expression level increased when comparing DACH1+ populations from control and Dach1OE mice? More clearly distinguishing the level of Dach1 expression from the number of Dach1+ cells in the presentation of the results would be helpful. Are there differences in the 5'UTR of the Dach1 transgene versus endogenous Dach1 that can be used to discriminate which DACH1 signal reflects endogenous Dach1 and which signal is coming from the transgene?

7. The nomenclature used here is confusing. Are dECs the same as type L cells? Or does “d” in dEC refer to Dach1 overexpressing? There seem to be 3 competing nomenclature systems (type L, dEC and sinusoids), which would appear to create unnecessary confusion. Unless there is a compelling reason otherwise, why not just use the most widely accepted terms (sinusoids)? Even if there is a reason to use another nomenclature (type L, type H, ect...), then recommend sticking with this throughout and eschewing other terms for the sake of consistency and readability.

8. Imprecise and novel terminology is unnecessarily used to characterize the skeletal lineage populations analyzed in Fig 7c. Are these CAR cells? If so, osteoCAR or adipoCAR/MALPs? It is recommended that these populations be more precisely annotated with terminology that reflects the current state of studies in skeletal cell types. Additionally, it is puzzling that relatively few skeletal clusters are detected, when it is now well established that there are many, many more skeletal lineage cell types. Similar issues apply to both the scRNA-seq analysis of skeletal cells in the main text and extended data. Most skeletal scRNA-seq studies resolve more clusters than produced here. How were the clustering settings determined for this analysis? Does clustering into a larger number of populations yield additional “resolution” of skeletal cell types that appear to be biologically meaningful and correspond to established cell types? There is some concern that the biology of effects on skeletal cell types may be partially obscured by “blobbing” via poor resolution of the skeletal populations.

8. Related to the point above, how are skeletal populations classified with anatomic notations (e.g., diaphyseal) in the scRNA-seq analysis? As discussed above, the annotation and clustering of skeletal cells here is problematic and should be reconsidered and better aligned with the underlying biology and contemporary systems of classification for these cells.

9. Page 10, “which indicates that overexpression of the transcription factor shifts these ECs into the dEC”—this interpretation is not warranted from the data. The scRNA-seq data presented is not informative about which population gave rise to the dEC cells, as it can only classify dEC cells by their current cell state and cannot reliably infer their cell of origin.

10. The argument that type R cells are “specialized” as claimed in the title and not just general post-arteriolar capillaries could be better developed, perhaps through comparison to other tissues. Is Dach1 similarly expressed in comparable non-skeletal endothelial populations? Are there non-skeletal phenotypes in the Dach1 gain and loss of function mice? What clearly distinguishes type R cells from generic capillaries?

Reviewer #3:

Remarks to the Author:

The authors characterized the variety of endothelial cells in adult long bones through expression of key genes, single cell RNA sequencing, and changes with changes of expression of DACH1 (known activator of arterial cells), aging and use of drugs to decrease (alendronate) or increase (PTH 1-34). They characterize distinct endothelial cells in the hypoxic environment of the bone marrow sinusoids, metaphyseal sites involved in bone formation, and a similar but distinct group of endothelial cells (that they call R type) seen primarily in older bones. These cell types could be distinguished by single cell sequencing into clusters with distinct patterns of gene expression. The focus on R type endothelial cells to show that they decrease expression of genes associated with hypoxia and express genes expected to favor osteoblast formation over adipocyte formation. Dach1 overexpression increased R type endothelium and associated increased trabecular bone; Dach1 knockout in endothelial cells led to decreased R type endothelium and fewer osteoblast cells. Cortical bones in very old mice show invasion of R type capillaries, osteoblasts, and osteoclasts. Alendronate administration led to an increase in R type endothelium, as did PTH administration.

The originality of the paper lies in its detailed analysis of endothelial types and the effects on osteoblastic cells when the R type of endothelium is manipulated. The data are extensive and novel, though do not attempt to establish the mechanisms of the links between the changes in endothelium and osteoblasts/clasts. Effects of changes of oxygenation and local gene expression remain to be sorted out with mechanistic experiments perhaps in the future. The conclusions appear robust and statistically sound, though the statistical analysis of the beautiful histologic data is limited. (Dramatic figures are shown, without details about how many bones were studied, numbers of various types of cells seen in how many bones, etc). The DACH1 experiments are particularly powerful in showing how changes in the blood vessels lead (by mechanisms not established here) to changes in bone cell numbers.

I may have misunderstood the alendronate experiment, but was surprised to see an increase in the fraction of type R endothelium in a setting of profound decrease of bone formation and resorption caused by alendronate. Some exploration of that would be useful.

Minor points:

1. In Figure 7m, the expression of BMP3 is shown to go up, and it is listed as a growth factor. In fact, multiple papers show that BMP3 suppresses bone formation, probably through actions on Acvr2b (see, for example, Kokabu S et al, Mol Endo 26:87 (2012).) BMP3 increase here presumably tempers to effects of growth factors in this setting.
2. In Extended figure 7e there is a pretty clear increase in EMCN+VEBFR3negative cells in response to alendronate, with an opposite result shown in Extended figure 7g (almost all the capillaries being

EMCN+/VEGFR3+ in response to alendronate (to me, the expected result). Maybe this is just a reflection of the sections chosen, but this is the sort of thing that could benefit from more statistical rigor.

3. I found the abstract easy to follow, but found much of the manuscript slow going, partly because of the multiple terminologies used to describe the vessel types.

ABSTRACT AND MAIN TEXT – please follow the guidelines that are specific to the format of your manuscript, as listed in our Guide to Authors (http://www.nature.com/ncb/pdf/ncb_gta.pdf) Briefly, Nature Cell Biology Articles, Resources and Technical Reports have 3500 words, including a 150 word

abstract, and the main text is subdivided in Introduction, Results, and Discussion sections. Nature Cell Biology Letters have up to 2500 words, including a 180 word introductory paragraph (abstract), and the text is not subdivided in sections.

Methods should be written concisely, but should contain all elements necessary to allow interpretation and replication of the results. As a guideline, Methods sections typically do not exceed 3,000 words. The Methods should be divided into subsections listing reagents and techniques. When citing previous methods, accurate references should be provided and any alterations should be noted. Information must be provided about: antibody dilutions, company names, catalogue numbers and clone numbers for monoclonal antibodies; sequences of RNAi and cDNA probes/primers or company names and catalogue

numbers if reagents are commercial; cell line names, sources and information on cell line identity and authentication. Animal studies and experiments involving human subjects must be reported in detail, identifying the committees approving the protocols. For studies involving human subjects/samples, a statement must be included confirming that informed consent was obtained. Statistical analyses and information on the reproducibility of experimental results should be provided in a section titled “Statistics and Reproducibility”.

All Nature Cell Biology manuscripts submitted on or after March 21 2016 must include a Data availability statement at the end of the Methods section. For Springer Nature policies on data availability see <http://www.nature.com/authors/policies/availability.html>; for more information on this particular policy see <http://www.nature.com/authors/policies/data/data-availability-statements-data-citations.pdf>. The Data availability statement should include:

- Accession codes for primary datasets (generated during the study under consideration and designated as "primary accessions") and secondary datasets (published datasets reanalysed during the study under consideration, designated as "referenced accessions"). For primary accessions data should be made public to coincide with publication of the manuscript. A list of data types for which submission to community-endorsed public repositories is mandated (including sequence, structure, microarray, deep sequencing data) can be found here <http://www.nature.com/authors/policies/availability.html#data>.
- Unique identifiers (accession codes, DOIs or other unique persistent identifier) and hyperlinks for datasets deposited in an approved repository, but for which data deposition is not mandated (see here for details <http://www.nature.com/sdata/data-policies/repositories>).
- At a minimum, please include a statement confirming that all relevant data are available from the authors, and/or are included with the manuscript (e.g. as source data or supplementary information), listing which data are included (e.g. by figure panels and data types) and mentioning any restrictions on availability.
- If a dataset has a Digital Object Identifier (DOI) as its unique identifier, we strongly encourage including this in the Reference list and citing the dataset in the Methods.

We recommend that you upload the step-by-step protocols used in this manuscript to the Protocol Exchange. More details can found at www.nature.com/protocolexchange/about.

All imaging data should be accompanied by scale bars, which should be defined in the legend. Cropped images of gels/blots are acceptable, but need to be accompanied by size markers, and to retain visible background signal within the linear range (i.e. should not be saturated). The boundaries of panels with low background have to be demarked with black lines. Splicing of panels should only be considered if unavoidable, and must be clearly marked on the figure, and noted in the legend with a statement on whether the samples were obtained and processed simultaneously. Quantitative comparisons between samples on different gels/blots are discouraged; if this is unavoidable, it should only be performed for samples derived from the same experiment with gels/blots were processed in parallel, which needs to be stated in the legend.

- We do not recommend using Adobe Photoshop for designing figures, but we can accept Photoshop generated (.PSD or .TIFF) files only if each element included in the figure (text, labels, pictures, graphs, arrows and scale bars) are on separate layers. All text should be editable in 'type layers' and line-art

such as graphs and other simple schematics should be preserved and embedded within 'vector smart objects' - not flattened raster/bitmap graphics.

Supplementary items should relate to a main text figure, wherever possible, and should be mentioned sequentially in the main manuscript, designated as Supplementary Figure, Table, Video, or Note, and numbered continuously (e.g. Supplementary Figure 1, Supplementary Figure 2, Supplementary Table 1,

Supplementary Table 2 etc.).

The total number of Supplementary Figures (not including the “unprocessed scans” Supplementary Figure) should not exceed the number of main display items (figures and/or tables (see our Guide to Authors and March 2012 editorial <http://www.nature.com/ncb/authors/submit/index.html#suppinfo>; <http://www.nature.com/ncb/journal/v14/n3/index.html#ed>). No restrictions apply to Supplementary Tables or Videos, but we advise authors to be selective in including supplemental data.

GUIDELINES FOR EXPERIMENTAL AND STATISTICAL REPORTING

REPORTING REQUIREMENTS – To improve the quality of methods and statistics reporting in our papers we have recently revised the reporting checklist we introduced in 2013. We are now asking all life sciences authors to complete two items: an Editorial Policy Checklist (found here <https://www.nature.com/authors/policies/Policy.pdf>) that verifies compliance with all required editorial policies and a reporting summary (found here <https://www.nature.com/authors/policies/ReportingSummary.pdf>) that collects information on experimental design and reagents. These documents are available to referees to aid the evaluation of the manuscript. Please note that these forms are dynamic ‘smart pdfs’ and must therefore be downloaded and completed in Adobe Reader. We will then flatten them for ease of use by the reviewers. If you would like to reference the guidance text as you complete the template, please access these flattened versions at <http://www.nature.com/authors/policies/availability.html>.

STATISTICS – Wherever statistics have been derived the legend needs to provide the n number (i.e. the sample size used to derive statistics) as a precise value (not a range), and define what this value represents. Error bars need to be defined in the legends (e.g. SD, SEM) together with a measure of

centre (e.g. mean, median). Box plots need to be defined in terms of minima, maxima, centre, and percentiles. Ranges are more appropriate than standard errors for small data sets. Wherever statistical significance has been derived, precise p values need to be provided and the statistical test used needs to be stated in the legend. Statistics such as error bars must not be derived from $n < 3$. For sample sizes of $n < 5$ please plot the individual data points rather than providing bar graphs. Deriving statistics from technical replicate samples, rather than biological replicates is strongly discouraged. Wherever statistical significance has been derived, precise p values need to be provided and the statistical test stated in the legend.

Author Rebuttal to Initial comments

We would like to thank all reviewers for their time, effort and constructive suggestions, which are greatly appreciated and have enabled us to improve the manuscript further. As you will see, we have included a substantial number of new results and have also improved the data presentation. While a detailed point-by-point reply to all comments is provided below, we would like to start by summarizing the most important new results in the revised manuscript:

- Trajectory and pseudo-time analysis of bone ECs based on scRNA-seq data (**Extended Data Fig. 3a-d**)
- Validation of *Aplnr-CreERT2* specificity and fate tracking of *Aplnr*⁺ ECs in femur (**Fig. 4a-d**)
- Immunostaining of bone vessels in human biopsy samples from young and aged patients (**Fig. 5g** and **Extended Data Fig. 5a, b**)
- Immunostaining and quantitation of type R vessels in C57BL6 wild-type female and male femurs at 12 weeks (**Extended Data Fig. 5c-e**)
- Characterization of female and male EC-specific *Dach1* loss-of-function mutants (**Fig. 6d-h** and **Extended Data Fig. 6f-i**)
- Analysis and quantitation of vascular density in retina and intestinal villi of adult EC-specific *Dach1* loss-of-function mutants (**Extended Data Fig. 6a-e**)
- Re-integration and analysis of scRNA-seq EC clusters in *Dach1* gain-of-function mutants (**Extended Data Fig. 7f-j**)
- Immunostaining and quantitation of osteoclasts in *Dach1* loss-of-function mutants (**Extended Data Fig. 6f-h**)
- Data showing the upregulation of arterial markers and Delta-like 4 (Dll4) in *Dach1* gain-of-function mutants and reduction of EMCN⁺ VEGFR3⁺ type R ECs in *Dll4* loss-of-function mutants (**Extended Data Fig. 8a-g**)
- Heatmap of differentially expressed secreted factors in bone EC subpopulations and in vitro experiments showing the effect of selected factors on osteogenesis and osteoclastogenesis (**Extended Data Fig. 8h-l**)
- Overview of the expression of selected markers in bone EC subpopulations (**Extended Data Fig. 3e**)

Point-by-point response to reviewers' comments:

Reviewer #1: Remarks to the Author:

Bone marrow endothelial cells (BMECs) are a topic of broad interest in various fields including hematology, vascular biology, orthopedics, and regenerative medicine. Nevertheless, compared to endothelial cells in other vascular beds, much less is known about BMECs due to the technical difficulty related to histological analysis of bones. In the last decade the authors' group have developed a new histological technique of bones and established a landmark concept for the heterogeneity of BMECs, particularly in embryonic and juvenile stages.

The present study intensively studied BMECs in adult stage after completion of bone development, and uncovered a specialized BMECs located in post-arterial capillaries, named as "type R ECs". Mechanistically the authors showed this new EC subtype contributes to bone remodeling depending on a transcription factor *Dach1*. In the technical viewpoint, the single cell data is based on a robust number of 62677 ECs, much larger than that of previous

reports, which make the data very convincing. The images shown in this paper is of extraordinary quality, and the quantification is solid.

Overall, this paper clearly enhances our understanding of the mechanism, how blood vessels maintain bones as well as the heterogeneity of BMECs. This reviewer only has some minor comments:

Thank you very for this summary and evaluation of our work. We agree that our data will add value to the broader areas related to bone, bone marrow and vascular biology. We have addressed both major and minor comments raised by the reviewers.

1. (Title) This reviewer recommends “adult” should be included somewhere in the title to differentiate from well-studied developmental analysis.

Thank you for this comment. We have changed the title to reflect the role of type R capillaries in adult bone.

2. (Fig. 1c) Color assignment should be improved. To this reviewer, juvenile and adult cells are hard to be distinguished.

Thank you for this feedback. We have modified the color assignment for the cell populations in **Fig. 1c** of the revised manuscript.

3. (Fig. 1e) What about the expression of tip cell markers like *Esm1*, *Nid2*, and *Angpt2*? If they are enriched in type R ECs, it makes sense as type R ECs look extending lots of filopodia (Fig. 3e).

Thank you for this comment. We show below expression data for *Esm1*, *Nid2* and *Angpt2* in bone ECs. *Angpt2* is indeed enriched in Type R vessels, which is very interesting because a previous study (PMID: 28214341) has identified the growth factor as a positive regulator of osteogenesis, which aligns with our finding linking type R capillaries to bone remodeling. Regarding *Nid2* and *Esm1*, which have been previously identified as tip cell markers in the retinal vasculature and elsewhere (PMID: 20705756), transcripts do not appear to be enriched in the rEC population. This could reflect organ-specific differences but might also indicate that these ECs are distinct from tip cells despite morphological similarities such as filopodial protrusions.

Reviewer Fig.: UMAP and DotPlot visualization of *Esm1*, *Nid2* and *Angpt2* expression in bone ECs.

4. (Fig. 1f) It will be helpful if the authors provide the list of marker genes at least in type R ECs (not just top 4) as a supplemental file.

As suggested by the reviewer, we have now included a list of marker genes for all bone EC subtypes (**Supplementary Table 1**).

5. (Fig. 1g) The lack of lymphatic markers should be discussed more as it directly conflicts with a recent paper (Biswas et al., Cell 2023). Is it possible to show some positive control in this heatmap such as small contamination of hematopoietic cells known to express Prox1 and Lyve1?

Thank you very much for this question. In our scRNA-seq dataset of *Cdh5-mTnG* sorted endothelial cells, we observed a total of 11 *Prox1*⁺ and *Lyve1*⁺ cells out of 23,215 ECs, which is 0.047% (see below). Since this dataset is based on 52 bones (femur and tibia) from different age and treatment groups, the tiny number of *Prox1*⁺ *Lyve1*⁺ cells is obviously not substantial enough to support the presence of lymphatics in bone. While we took great care to remove any adhering tissue during sample preparation, it is also difficult to completely exclude small periosteal contaminations, which could include lymphatic vessels that are attached to the outer bone surface (PMID: 32188632). We agree that the presence (or absence) of lymphatic vessels inside bone is an important question, which deserves a separate investigation with suitable reporter alleles and other approaches.

Reviewer Fig.: UMAP visualization of *Prox1*, *Lyve1* and *Prox1-Lyve1* in bone ECs.

6. (Fig. 3g-j; Extended Fig. 2c-e) Lineage tracing experiments using *Bmx-CreERT2* and *Flt4-CreERT2* show type R ECs are derived from sinusoidal ECs. Could this finding be confirmed by pseudo-time analysis of single cell RNA-seq data?

Thank you for this excellent suggestion. We have now included trajectory plots and pseudo-time analysis of our scRNA-seq data (**Extended Data Fig 3a-d**). These analyses support conversion of *Flt4*⁺ bmECs into *C1qtnf9*⁺ rECs. The latter are positioned in close proximity of *Bmx*⁺ aECs, which is consistent with the conversion of type R ECs into arterial endothelium. Pseudo-time analysis of EC subsets with selected marker genes, namely *Aplnr*, *Flt4*, *Bmx* and *C1qtnf9*, also argues for a transcriptional continuum involving bmECs, rECs and aECs *in silico*.

In addition to the *Flt4-CreERT2* fate tracking data, we have included new lineage tracing results with the *Aplnr-CreERT2* line (**Fig 4a-e**). These data confirm that rECs contribute to arterial endothelium, which also helps to explain why the two vessel populations are coupled.

Taken together, trajectory plots and pseudo-time analysis in the revised manuscript are fully consistent with the results obtained from genetic approaches *in vivo*.

7. (Extended Fig. 4a) These images are hard to interpret. If the authors want to claim the merge of Sp7-mCherry and HIF1 α , assigned colors should be changed at least.

We agree with the reviewer and have modified the color scheme (**Extended Data Fig. 7d**). It is now easier to see HIF1 α (in white) in and around Sp7-mCherry⁺ (cyan) trabecular bone, which is reduced after the emergence of type R capillaries in 12-week-old mice.

8. (Fig. 6a, b) Why the authors focused on Dach1 should be explained more. Is that derived from the list of marker genes characterizing type R cells?

Thank you for this question. *Dach1* is strongly enriched in Type R ECs relative to all other bone EC subpopulations (**Fig. 6a**). As suggested by the reviewer, we have included a short statement explaining the rationale for choosing *Dach1* in the revised manuscript.

9. (Fig. 7d) Why rECs are not increased in Dach1-OE should be explained, as this data is inconsistent with immunostaining (Fig. 6f). Do they shift into the dEC cluster? If so, do they retain the feature of Emcn+Flt4-?

Thank you very much for this question. To clarify this important issue, we have re-integrated our *Dach1OE* dataset and obtained clusters that are consistent with our other scRNA-seq data (**Extended Data Fig 7f-i**). The new analysis makes it clear that *Dach1*⁺ cells are found in the aEC and rEC clusters, which is consistent with the increase of CAV1⁺ vessels seen by immunostaining (**Fig. 7b**). The scRNA-seq data also show the presence of *Dach1*⁺ cells in the mpEC and bmEC clusters. Likewise, immunostaining confirms the presence of *Dach1*⁺ cells in the *Dach1OE* sinusoidal vasculature (see **Reviewer Figures** below). In our view, this shows that the overexpression of DACH1 is not sufficient to convert every single EC into type R or arterial endothelium. The result likely reflects that only certain bmECs can acquire the rEC/aEC fate, which may depend on cell-intrinsic (e.g. expression profile) or external factors (e.g. positional or biomechanical clues).

Regarding the molecular properties of the unconverted *Dach1*⁺ bmECs, we find that they retain *Flt4* expression and other features (eg. *Cxadr*) of sinusoidal ECs (see **Reviewer Figures** below). Taken together, it is clear that DACH1 overexpression can induce the conversion into rECs/aECs in some bmECs, whereas other cells retain key features of sinusoidal endothelium. It is for this reason that we no longer use the term dECs in the revised manuscript.

Reviewer Fig.: High-magnification images of femurs immunostained for EMCN (red), CXADR (blue) and DACH1 (yellow) showing the presence of DACH1⁺ (white arrowheads) in EMCN⁺ CXADR⁺ bmECs around bone marrow in control and *Dach1*^{OE}.

Reviewer Fig.: Heatmaps and violin plots showing the expression of Flt4 in all ECs (left) or the bmEC subpopulation (right) from *Dach1* gain-of-function (OE) mutants and littermate controls. Heatmaps show selected additional markers.

10. (Extended Fig. 6) In this reviewer's view, this loss-of-function data is a key for the entire manuscript, so should be moved to main figures. In contrast, hypoxia data without any functional analysis (Fig. 5) could be moved to Extended figures.

We thank the reviewer for this comment and suggestion. We have extended the *Dach1* loss-of-function data and show the results as a main figure of the revised manuscript (**Fig. 6c-h**). In brief, the new data show that EC-specific inactivation of *Dach1* leads to the reduction of type R capillaries and loss of trabecular bone, which is much more pronounced in male mutants than in females (**Fig. 6d, g, h** and **Extended Data Fig. 6i**). For the latter finding, it is important to appreciate that wild-type adult male mice, which are known to have more trabecular bone than age-matched females (PMID: 17488199), also have substantially more type R capillaries (**Extended Data Fig. 5c-e**).

Reviewer #2:

Remarks to the Author:

In this report, Mohanakrishnan et al. analyze a very large scRNA-seq dataset focused on skeletal endothelial cells, most notably finding a population corresponding to post-arteriolar capillaries they term type R cells. Based on these cells expressing *Dach1*, *Dach1* conditional gain- and loss-of-function mice are studied and found to display alterations in skeletal endothelial cells and also bone metabolism. Lastly, treatment with agents relevant to bone physiology (e.g., PTH) is seen to alter the amount of type R endothelial cells. Overall, the question of how skeletal endothelium controls bone formation is very important, underexplored, and appropriate in topic and scope for Nature Cell Biology. There are some important concerns about the site of activity of the *Alpnr*-creERT2 system and concerns that the key *Dach1* loss and gain of function models are not studied in adequate depth and are

lacking basic phenotypic information. Otherwise, these studies are well done from a technical perspective. However, the key weakness of this report is that there is no causative data that establish the function or importance of type R endothelial cells or mechanisms by which they interact with osteoblast-lineage cells. In particular, the two key experiments in this manuscript are the *Dach1* overexpression and loss-of-function models. However, it appears that many skeletal endothelial cell types are impacted, including cell types previously described by the authors as impacting bone metabolism, so it is difficult to specifically ascribe any phenotypes observed to type R cells. Moreover, the phenotypes seen with *Dach1* gain and loss of function appear to be very modest and only superficially studied in the present data. No other data is presented to study a mechanism by which type R cells may impact bone metabolism. These major limitations appear to also be appreciated and acknowledged by the authors given the wording used in the discussion. These limitations result in the present manuscript not being able to make a clear case for the importance or functional relevance of type R endothelial cells.

Thank you very much for your assessment and constructive feedback on our manuscript. We have taken note of the various criticisms and address them in our point-by-point response below. We while discuss limitations and open questions relevant for future studies in the Discussion section of the manuscript, we are convinced that our findings strongly support the importance and functional relevance of type R endothelial cells.

Major points:

1. The sites of activity of *Alpnr-creERT2* activity should be more clearly determined with a reporter for cre activity (such as the mTnG system used elsewhere here). Data from these methods should then be correlated and compared with the various cellular phenotypes and skeletal metabolism phenotypes presented. It is essential that it be robustly confirmed that the *Dach1* transgene is only activated in endothelial cells and not other skeletal cell types. Data in fig 6b are seen as not adequate to address this important point, as many cells are DACH1 positive and the difference between the *Dach1*OE and control mice DACH1 staining is only a minor difference of degree and does not currently make a compelling case for the specificity of the overexpression effect within endothelial cells.

We agree and have included additional data showing the characterization of *Aplnr-CreERT2* activity in the bone vasculature. Our scRNA-seq data shows *Aplnr* expression predominantly in rECs and mpECs but not in arteries (**Fig. 4a**), which is confirmed by *Aplnr-CreERT2*-mediated recombination of a Cre reporter (**Fig. 4b**).

We also show with our scRNA-seq data that *Aplnr-CreERT2*-mediated overexpression of *Dach1* is confined to endothelial cells and absent from bone mesenchymal stromal cells (see **Reviewer Figure** below). *Aplnr* expression is confined to endothelial cells in the Tabula muris database and in our own large dataset of non-hematopoietic cells from bone. This is consistent with the pattern of *Aplnr-CreERT2*-mediated GFP expression in the vasculature (**Fig. 4b, c**). Furthermore, we have used the *Aplnr-CreERT2* line for genetic fate tracking experiments to show that recombined (GFP+) ECs contribute to the arterial endothelium over time (**Fig 4b-e**).

Reviewer Fig.: UMAP plots showing the expression of *Dach1* in control and *Dach1* gain-of-function ECs (left column) but not in bone mesenchymal stromal cells (right column).

Reviewer Fig.: UMAP and violin plots showing that the expression of *Aplnr* is primarily confined to ECs in a large integrated dataset of non-hematopoietic cells from bone (unpublished).

2. The *Dach1*^{OE} experiment is interesting, however it would appear from the scRNA-seq study that the phenotype is likely to be driven by dECs, since this appears to be the most strongly expanded population. Since dECs do not appear to correspond to type R endothelium, this dilutes the argument that this data implicates type R endothelium in bone metabolism. More broadly, the impact of *Dach1* overexpression, some of which may be ectopic in cell types not normally expressing *Dach1* (see comment #1 above) can be non-physiologic and difficult to interpret. This weakness could be effectively mitigated if the *Dach1* loss-of-function showed a clearly supportive phenotype compatible with the *Dach1*^{OE} result, but this doesn't appear to be the case in the currently presented data (see main issue #5 below).

Thank you for your feedback. We apologize for any confusion caused by the term dEC in the original submission, which referred to a *Dach1*⁺ subpopulation of bmECs in *DACH1* gain-of-function mice.

To resolve this important issue, we have re-integrated our *Dach1*^{OE} dataset and obtained clusters that are consistent with our other scRNA-seq data (**Extended Data Fig 7f-i**). The new analysis makes it clear that *Dach1*⁺ cells are found in the aEC and rEC clusters, which is consistent with the increase of *CAV1*⁺ vessels seen by immunostaining (**Fig. 7b**). The scRNA-seq data also show the presence of *Dach1*⁺ cells in the mpEC and bmEC clusters. Likewise, immunostaining confirms the presence of *Dach1*⁺ cells in the *Dach1*^{OE} sinusoidal vasculature (see **Reviewer Figure** in our response to question 1). This shows that the overexpression of *Dach1* is not sufficient to convert every single EC/bmEC into type R or arterial endothelium and likely reflects that only certain bmECs can acquire the rEC/aEC fate. This may depend on cell-intrinsic (e.g. expression profile) or external factors (e.g. positional or biomechanical clues).

We have also extended the *Dach1* loss-of-function data and show it in a main figure of the revised manuscript (**Fig. 6c-h**). In brief, the new data show that EC-specific inactivation of *Dach1* leads to the reduction of type R capillaries and loss of trabecular bone, which is much more pronounced in male mutants than in females (**Fig. 6d, g, h** and **Extended Data Fig. 6i**). For the latter finding, it is important to note that wild-type adult male mice, which are known to have more trabecular bone than age-matched females (PMID: 17488199), also have a substantially larger fraction of type R capillaries (**Extended Data Fig. 5c-e**).

3. Is there a cortical phenotype in *Dach1*OE mice? The trabecular phenotype presented appears to be modest to moderate at best. Formal quantitative histomorphometry to determine rates of bone formation and analysis of osteoclast numbers/activity is important for both the *Dach1* gain and loss of function mice.

Thank you for your feedback. We have now included cortical bone thickness in the histomorphometric analysis of the *Dach1* gain-of-function phenotype (**Fig. 7f**). There is no significant difference. Representative images and quantitation of OSX⁺ cells and osteoclasts are shown in **Extended Data Fig. 7a, b**. Indicating higher bone turnover, both OSX⁺ and ATP6V1B1/B2⁺ cells are increased.

The newly added results for *Dach1* loss-of-function mutants include representative μ CT images and histomorphometric data (**Fig. 6g, h**) together with representative immunostainings and quantitation of OSX⁺ cells and osteoclasts (**Extended Data Fig. 6f-i**). Mutants show a reduction of osteoblast lineage cells, whereas osteoclasts are increased.

Looking at the sum of the data above, it is clear that EC-specific genetic alterations in *Dach1* expression significantly affect the number of type R capillaries but also trabecular bone formation, which most obvious in the distal bone marrow in proximity of the metaphysis.

4. Both the *Dach1* gain-of-function and loss-of-function mouse models appear to invoke changes in multiple endothelial populations. While this data remains interesting for further establishing the function of *Dach1* and for further elucidating endothelial to osteoblast-lineage functional connections, it does not appear to make a strong case for the function of type R endothelial cells given that many endothelial subsets are impacted. From this perspective, it is unclear how this data fits into the broader line of investigation focused on type R cells that comes before and after this data.

Thank you very much for this comment, which we have already addressed in our reply to question 2. In brief, it is correct that not all *Dach1*⁺ bmECs in gain-of-function mice are converted into type R ECs or arterial endothelium. Regarding the molecular properties of the unconverted *Dach1*⁺ bmECs, we find that they retain *Flt4* expression and other features (eg. *Cxadr*) of sinusoidal ECs (see **Reviewer Figures** below). We have no evidence indicating that *Dach1*⁺ bmECs are functionally compromised and it is highly unlikely that these cells are responsible for the increased trabecular bone formation or metabolic alterations in *Dach1* gain-of-function mutants. Likewise, the extension of the *Dach1* loss-of-function data (**Fig. 6c-h**) argues that the phenotypic alterations are a consequence of reduced type R capillaries.

Reviewer Fig.: High-magnification images of femurs immunostained for EMCN (red), CXADR (blue) and DACH1 (yellow) showing the presence of DACH1⁺ (white arrowheads) in EMCN⁺ CXADR⁺ bmECs around bone marrow in control and *Dach1*^{OE}.

Reviewer Fig.: Heatmaps and violin plots showing the expression of Flt4 in all ECs (left) or the bmEC subpopulation (right) from *Dach1* gain-of-function (OE) mutants and littermate controls. Heatmaps show selected additional markers.

5. The *Dach1* loss of function model is very incompletely described with only immunostaining data. What is the phenotype on uCT and regular histology? The phenotype associated with *Dach1* loss-of-function is likely the very most important data to link type R cells to bone metabolism and therefore at the conceptual heart of this work, so is concerning that this data is very cursory, does not demonstrate a compelling phenotype, and is relegated to the supplemental figures.

We thank the reviewer for this comment and suggestion. We have extended the *Dach1* loss-of-function data and show the results as a main figure of the revised manuscript (**Fig. 6c-h**). In brief, the new data show that EC-specific inactivation of *Dach1* leads to the reduction of type R capillaries and loss of trabecular bone, which is much more pronounced in male mutants than in females (**Fig. 6d, g, h** and **Extended Data Fig. 6i**). Regarding the sexual dimorphism of the phenotypic alterations, it is important to appreciate that wild-type adult male mice, which are known to have more trabecular bone than age-matched females (PMID: 17488199), also have substantially more type R capillaries (**Extended Data Fig. 5c-e**).

Minor points:

1 .MACS-based cell depletion as employed the scRNA-seq study can remove cell types of interest. Gating used for the cell sorting should be displayed. It is unlikely that the MACS depletion method used fully removed hematopoietic cells. Similarly, it is possible that CDH5 cre may label non-endothelial populations, though it is appreciated that the authors have used CDH5-based lineage reporters extensively. Were cells filtered from the scRNA-seq data prior to clustering? If so, what criteria were applied?

Thank you for your feedback. FACS gating is shown in **Extended Data Fig. 1a**.

Remaining hematopoietic cells after the MACS step were filtered out as part of the quality control. This information has been added to the Methods.

We can confirm that we have repeatedly validated the EC specificity of the *Cdh5-CreERT2* line over the years. Bone staining of *Cdh5-CreERT2 Rosa26-mTmG* double transgenic mice (after treatment with tamoxifen) shows that GFP signal is clearly confined to the vasculature (see below), which is consistent with the strictly EC-specific expression of *Cdh5* seen in scRNA-seq datasets. Flow cytometry results further support EC-specific activity of the CreERT2 line and absence of recombination in hematopoietic cells (see below) but also other cell types.

Reviewer Fig.: Validation of EC-specific *Cdh5-CreERT2* activity.

2. EMCN is also known to label osteoblast-lineage cells. Is this observed in the EMCN immunofluorescence staining? Lower power views of the IF staining in Fig 2 would be helpful for evaluation.

We do not observe EMCN expression in osteoblast-lineage cells in our data, which is also supported by scRNA-seq data that shows no overt *Emcn* expression in bone mesenchymal stromal cells (see **Reviewer Figure** below). Likewise, EMCN immunostaining labels endothelial cells but is clearly absent from trabecular bone (**Fig. 2c-e**)

Reviewer Fig.: UMAP plots showing the expression of *Emcn* in most EC populations with the exception of aECs (top). In contrast, no substantial *Emcn* expression is detectable in bone mesenchymal stromal cell populations (bottom).

3. Claims that type R vessels are in proximity to osteoblasts and osteoclasts would be more convincing if analyzed in a more systematic manner than the example images in Fig 4 and performed comparing type H or other vessel subsets to type R endothelial cells.

Thank you for your comment. We show numerous examples that type R vessels are specifically located around trabecular bone (Fig. 2d, Fig. 5a-f, Ext. Data Fig. 4a-c). Given that it is well established that bone surface contains osteogenic regions next to osteoclast-containing areas undergoing resorption (PMID: 17705416), it is not at all surprising that rECs are located in direct proximity of osteoprogenitors but also of osteoclasts, as is shown in detail in Fig. 5a-f. Regarding the comparison to type H vessels, which are also known to interact with perivascular osteoprogenitor cells (PMID: 24646994), it is important to recognize that type H are most abundant in postnatal/juvenile bone but are strongly reduced in adult animals. In the latter, type H capillaries are confined to a small region near the growth plate (see below) and are therefore located far away from most trabecular bone, which is typically surrounded by bone marrow (BM) in adult long bone (see below). Thus, type H and type R capillaries are found in different regions of long bone and their abundance is differentially altered during life. Type R capillaries are not yet formed in postnatal bone when type H vessels (mpECs) are most abundant. In turn, type H vessels are scarce/absent in aging bone when type R capillaries are increased. Thus, there are fundamental spatial, temporal and molecular differences between type H and type R vessels. Arteries and sinusoids (type L vessels) are not associated with bone-forming cells and therefore probably not relevant as short-range regulators of osteoprogenitor and osteoclast function.

Reviewer Fig.: Immunostaining of adult femur for CD31 (green), EMCN (red) and OSX (cyan). While arrows indicate type H vessels, yellow arrows mark regions with type R capillaries around trabecular bone.

4. For analysis of *Xbp1* expression on page 8, it is important to distinguish between spliced and unspliced *Xbp1*. To this reviewer's knowledge it is *Xbp1* splicing and not basal expression that is associated with hypoxia.

Thank you very much for this comment. There is literature stating that hypoxia upregulates *Xbp1* at the transcriptional level and also activates splicing of its mRNA (PMID: 15342372). In breast cancer patients, *XBPL* gene expression is highly correlated with HIF1 α and hypoxia-driven signatures (PMID: 24670641). Nevertheless, we are no longer showing the *Xbp1* data in the revised manuscript due to space constraints.

5. The analysis of hypoxia associated markers on page 8 does not demonstrate a critical role for type R endothelium in regulating local oxygen tension (though it is appreciated that this is obviously the primary function of vascular endothelium). Instead, this data would seem to at most provide indirect evidence that these cells reside in a hypoxic environment. Similarly, the data on page 8 assigning a role for type R cells in bone turnover on the basis of the transcripts presented is interpreted too strongly. All of this data would be more properly presented as hypothesis-generating and should be better qualified. Similar issues apply to the discussion.

We agree that marker expression alone cannot “demonstrate” a functional role and therefore we did not use this term in this context. The emergence of type R capillaries during adolescence correlates with downregulation of markers of hypoxia (**Extended Data Fig. 7e**), but we also show that the increase in arteries and post-arterial type R capillaries in *Dach1* gain-of-function mutants leads to altered expression of markers associated with metabolism, oxygen sensing and osteogenesis in mesenchymal stromal cells (**Fig. 7k, l; Extended Data Fig. 7l**).

6. In the scRNA-seq study on page 9-10, was the DACH1 expression level increased when comparing DACH1+ populations from control and *Dach1*OE mice? More clearly distinguishing the level of *Dach1* expression from the number of *Dach1*+ cells in the presentation of the results would be helpful. Are there differences in the 5'UTR of the *Dach1* transgene versus endogenous *Dach1* that can be used to discriminate which DACH1 signal reflects endogenous *Dach1* and which signal is coming from the transgene?

Split view UMAP plots clearly show the overexpression of *Dach1*. The color of each individual point corresponds to the expression level (see below and **Extended Data Fig. 7i**). These results also prove that *Dach1* overexpression is EC-specific and not detectable in bone mesenchymal stromal cells.

Reviewer Fig.: Split view UMAP plots showing the endogenous expression of *Dach1* in control bone (top row) and overexpression of the transcription factor in *Dach1* gain-of-function mutant ECs (bottom row). Note that *Dach1*+ cells are very sparse both in control and mutant bone mesenchymal stromal cells (right column), which is not altered in EC-specific *Dach1* gain-of-function mice.

7. The nomenclature used here is confusing. Are dECs the same as type L cells? Or does “d” in dEC refer to Dach1 overexpressing? There seem to be 3 competing nomenclature systems (type L, dEC and sinusoids), which would appear to create unnecessary confusion. Unless there is a compelling reason otherwise, why not just used the most widely accepted terms (sinusoids)? Even if there is a reason to use another nomenclature (type L, type H, ect...), then recommend sticking with this throughout and eschewing other terms for the sake of consistency and readability.

Thank you very much for this feedback. We agree that the term dEC was confusing and we therefore no longer use it in the revised manuscript. Re-integration of our *Dach1*^{OE} scRNA-seq dataset makes it clear that *Dach1*⁺ cells are found in the aEC and rEC clusters, which is consistent with the increase of CAV1⁺ vessels seen by immunostaining (Fig. 7b). The scRNA-seq data also show the presence of *Dach1*⁺ cells in mpECs and, most prominently, in the bmEC cluster. Likewise, immunostaining confirms the presence of *Dach1*⁺ cells in the *Dach1*^{OE} sinusoidal vasculature (see figure below).

Regarding the molecular properties of the unconverted *Dach1*⁺ bmECs, we find that they retain *Flt4* expression and other features (eg. *Cxadr*) of sinusoidal ECs (see Reviewer Figures below). Taken together, it is clear that DACH1 overexpression can induce the conversion into rECs/aECs in some bmECs, whereas other cells retain key features of sinusoidal endothelium. It is for this reason that we no longer use the term dECs.

Reviewer Fig.: High-magnification images of femurs immunostained for EMCN (red), CXADR (blue) and DACH1 (yellow) showing the presence of DACH1⁺ (white arrowheads) in EMCN⁺ CXADR⁺ bmECs around bone marrow in control and *Dach1*^{OE}.

Reviewer Fig.: Heatmaps and violin plots showing the expression of *Flt4* in all ECs (left) or the bmEC subpopulation (right) from *Dach1* gain-of-function (OE) mutants and littermate controls. Heatmaps show selected additional markers.

8. Imprecise and novel terminology is unnecessarily used to characterize the skeletal lineage populations analyzed in Fig 7c. Are these CAR cells? If so, osteoCAR or adipoCAR/MALPs? It is recommended that these populations be more precisely annotated with terminology that reflects the current state of studies in skeletal cell types. Additionally,

it is puzzling that relatively few skeletal clusters are detected, when it is now well established that there are many, many more skeletal lineage cell types. Similar issues apply to both the scRNA-seq analysis of skeletal cells in the main text and extended data. Most skeletal scRNA-seq studies resolve more clusters than produced here. How were the clustering settings determined for this analysis? Does clustering into a larger number of populations yield additional “resolution” of skeletal cell types that appear to be biologically meaningful and correspond to established cell types? There is some concern that the biology of effects on skeletal cell types may be partially obscured by “blobbing” via poor resolution of the skeletal populations.

Thank you very much for your comments. It is important to note that different scRNA-seq studies of non-hematopoietic cells in bone have used very different nomenclatures, which complicate the direct comparison of results. Adipo-CAR and Osteo-CAR are terms used by the team led by Simon Haas in their excellent study (PMID: 31871321), whereas another landmark publication by the team of David Scadden refers to presumably the same subpopulations as MSC-descendent osteolineage cells (OLCs) and Lepr-MSCs, respectively (PMID: 31130381). Both subpopulations are likely to be part of the LepR⁺ cell cluster reported by the team of Iannis Aifantis (PMID: 30971824). Thus, we currently lack a unified nomenclature for bone MSCs. To clarify this issue in the revised manuscript, we now make clear that the osteo-poised mpMSCs are similar to Osteo-CAR/OLC cells, whereas dpMSCs share many features with the Adipo-CAR/Lepr-MSC population. A more precise and meaningful characterization of bone MSC subsets could be accomplished by integration and reanalysis of the different published datasets, which is, however, not within the scope of the current study. Our objective was the identification of transcriptomic changes in perivascular MSCs in response to DACH1 overexpression in endothelial cells with a focus on changes relating to metabolism, oxygenation and osteogenesis. This is also the reason why we have not included a more extensive subcluster analysis, which would reveal further cell subsets based on differences in gene expression.

With regard to extent of transcriptional changes in different cell subpopulations, it also needs to be considered that not all cells are positioned in direct proximity to vessels or type R capillaries. This makes it even more significant that we can detect transcriptional changes in bone MSCs and in HIF1 α immunostaining in Dach1OE bone sections (**Fig. 7l** and **Ext. Data Fig. 7k, l**).

8. Related to the point above, how are skeletal populations classified with anatomic notations (e.g., diaphyseal) in the scRNA-seq analysis? As discussed above, the annotation and clustering of skeletal cells here is problematic and should be reconsidered and better aligned with the underlying biology and contemporary systems of classification for these cells.

Thank you very much for this comment. As discussed above in our reply to question 7, there is currently no unified nomenclature for bone MSCs. Our nomenclature (PMID: 34260921) takes anatomical locations into account. mpMSCs are located in the metaphysis and dpMSCs in the diaphysis. We appreciate that other publications use different nomenclatures, which can be confusing for readers. As mentioned above, we now make clear that the osteo-poised mpMSCs are similar to Osteo-CAR/OLC cells, whereas dpMSCs share many features with the Adipo-CAR/Lepr-MSC population.

9. Page 10, “which indicates that overexpression of the transcription factor shifts these ECs into the dEC”—this interpretation is not warranted from the data. The scRNA-seq data

presented is not informative about which population gave rise to the dEC cells, as it can only classify dEC cells by their current cell state and cannot reliably infer their cell of origin.

Thank you very much for your feedback. Regarding the nature of *Dach1*⁺ ECs that are not giving rise to type R capillaries and arteries in *Dach1*^{OE} mutants, we find that most of these cells they retain *Flt4* expression and other features (eg. *Cxadr*) of sinusoidal ECs (see above in our response to question 7). Taken together, it is clear that DACH1 overexpression can induce the conversion into rECs/aECs in some bmECs, whereas other cells retain key features of sinusoidal endothelium.

10. The argument that type R cells are “specialized” as claimed in the title and not just general post-arteriolar capillaries could be better developed, perhaps through comparison to other tissues. Is *Dach1* similarly expressed in comparable non-skeletal endothelial populations? Are there non-skeletal phenotypes in the *Dach1* gain and loss of function mice? What clearly distinguishes type R cells from generic capillaries?

The term “specialized” refers to the vasculature in bone. While we have not attempted a broader comparison of endothelial cell transcriptomes from bone relative to other organs, it is already known that there are substantial organ-specific differences in gene expression and this heterogeneity across tissues is largest for capillary ECs (PMID: 32059779). Another landmark study has reported a “seamless continuum” of gene expression changes along the arteriovenous axis in blood vessels from the brain cortex (PMID: 29443965). This aspect is obviously very different in bone. Even though type R vessels are post-arterial capillaries, their gene expression is very distinct from arterial ECs but also from other endothelial subpopulations, which means that rECs are specialized with regard to their transcriptome, anatomical location and function.

Reviewer #3:

Remarks to the Author:

The authors characterized the variety of endothelial cells in adult long bones through expression of key genes, single cell RNA sequencing, and changes with changes of expression of DACH1 (known activator of arterial cells), aging and use of drugs to decrease (alendronate) or increase (PTH 1-34). They characterize distinct endothelial cells in the hypoxic environment of the bone marrow sinusoids, metaphyseal sites involved in bone formation, and a similar but distinct group of endothelial cells (that they call R type) seen primarily in older bones. These cell types could be distinguished by single cell sequencing into clusters with distinct patterns of gene expression. The focus on R type endothelial cells to show that they decrease expression of genes associated with hypoxia and express genes expected to favor osteoblast formation over adipocyte formation. *Dach1* overexpression increased R type endothelium and associated increased trabecular bone; *Dach1* knockout in endothelial cells led to decreased R type endothelium and fewer osteoblast cells. Cortical bones in very old mice show invasion of R type capillaries, osteoblasts, and osteoclasts. Alendronate administration led to an increase in R type endothelium, as did PTH administration.

The originality of the paper lies in its detailed analysis of endothelial types and the effects on osteoblastic cells when the R type of endothelium is manipulated. The data are extensive and novel, though do not attempt to establish the mechanisms of the links between the changes in

endothelium and osteoblasts/clasts. Effects of changes of oxygenation and local gene expression remain to be sorted out with mechanistic experiments perhaps in the future. The conclusions appear robust and statistically sound, though the statistical analysis of the beautiful histologic data is limited. (Dramatic figures are shown, without details about how many bones were studied, numbers of various types of cells seen in how many bones, etc). The DACH1 experiments are particularly powerful in showing how changes in the blood vessels lead (by mechanisms not established here) to changes in bone cell numbers.

Thank you very much for your feedback. We have included additional quantitation in the revised manuscript and the number of replicates/mice is mentioned in the corresponding figure legend. In the Method section, we also state in the section *Image processing and quantifications* that “All figures show maximum intensity projections, and we analyzed at least three mice for each condition, unless explicitly mentioned otherwise.

I may have misunderstood the alendronate experiment, but was surprised to see an increase in the fraction of type R endothelium in a setting of profound decrease of bone formation and resorption caused by alendronate. Some exploration of that would be useful.

First of all, we would like to clarify that Alendronate strongly increases trabecular bone volume, thickness and connectivity density (Ext. Data Fig. 10a), which is consistent with the published literature (PMID: 15664013, PMID: 19113917). Proangiogenic activity of Alendronate has also been reported, which was linked to enhanced expression of vascular endothelial growth factor (VEGF) (PMID: 35879908). Moreover, our own previous work has shown that Alendronate treatment leads to increases in type H vessels and blood flow in aged mice (see below and PMID: 27922003).

Given these previous reports, it is not surprising that type R vessels respond to Alendronate.

Minor points:

1. In Figure 7m, the expression of BMP3 is shown to go up, and it is listed as a growth factor. In fact, multiple papers show that BMP3 suppresses bone formation, probably through actions on *Acvr2b* (see, for example, Kokabu S et al, *Mol Endo* 26:87 (2012).) BMP3 increase here presumably tempers to effects of growth factors in this setting.

Thank you very much for alerting us to this important publication on BMP3, which is most appreciated. Indeed, it is absolutely feasible that type R vessels can influence bone formation and resorption in different directions. In the revised manuscript, we have examined the function of a few selected factors on osteogenesis and osteoclastogenesis *in vitro* (**Extended Data. Fig. 8h-l**). The resulting data support that type R EC-derived molecular signals might regulate bone remodeling, which is probably coupled to the metabolic regulation of cell behavior. Future work will undoubtedly provide further insight into the interplay of blood vessels and other cell types in physiological and pathological bone remodeling.

2. In Extended figure 7e there is a pretty clear increase in EMCN+VEBFR3negative cells in response to alendronate, with an opposite result shown in Extended figure 7g (almost all the capillaries being EMCN+/VEGFR3+ in response to alendronate (to me, the expected result). Maybe this is just a reflection of the sections chosen, but this is the sort of thing that could benefit from more statistical rigor.

Thank you very much for this comment. **Figure 8f-i** provides quantification of EMCN⁺ CAV1⁺ vessels (arteries + type R capillaries), EMCN⁺ VEGFR3⁻ vessels (type R capillaries) and OSX⁺ cells.

3. I found the abstract easy to follow, but found much of the manuscript slow going, partly because of the multiple terminologies used to describe the vessel types.

Thank you very much for this feedback. As mentioned above, we have simplified the terminology and no longer use the term dEC for *Dach1*-overexpressing sinusoidal ECs.

Decision Letter, first revision:

*Please delete the link to your author homepage if you wish to forward this email to co-authors.

Dear Professor Adams,

Your manuscript, "Specialized post-arterial capillaries facilitate adult bone remodeling", has now been seen by the 3 original reviewers. As you will see from their comments (attached below) they find this work of interest, but reviewers #2 and #3 have raised some important points. Although we are also interested in this study, we believe that their concerns should be addressed before we can consider publication in Nature Cell Biology.

Nature Cell Biology editors discuss the referee reports in detail within the editorial team, including the chief editor, and in this case we believe that all the remaining points by referees #2 and #3 should be addressed. We are committed to providing a fair and constructive peer-review process, so please feel free to contact me if you would like to discuss any of the referee comments further.

Finally, please pay close attention to our guidelines on statistical and methodological reporting (listed below) as failure to do so may delay the reconsideration of the revised manuscript. In particular please provide:

- a Supplementary Figure including unprocessed images of all gels/blots in the form of a multi-page pdf file. Please ensure that blots/gels are labeled and the sections presented in the figures are clearly indicated.
- a Supplementary Table including all numerical source data in Excel format, with data for different figures provided as different sheets within a single Excel file. The file should include source data giving rise to graphical representations and statistical descriptions in the paper and for all instances where the figures present representative experiments of multiple independent repeats, the source data of all repeats should be provided.

We therefore invite you to take these points into account when revising the manuscript. In addition, when preparing the revision please:

- ensure that it conforms to our format instructions and publication policies (see below and

<https://www.nature.com/nature/for-authors>).

- provide a point-by-point rebuttal to the full referee reports verbatim, as provided at the end of this letter.

- provide the completed Reporting Summary (found here <https://www.nature.com/documents/nr-reporting-summary.pdf>). This is essential for reconsideration of the manuscript and will be available to editors and referees in the event of peer review. For more information see <http://www.nature.com/authors/policies/availability.html> or contact me.

When submitting the revised version of your manuscript, please pay close attention to our [href="https://www.nature.com/nature-portfolio/editorial-policies/image-integrity">Digital Image Integrity Guidelines](https://www.nature.com/nature-portfolio/editorial-policies/image-integrity). and to the following points below:

Nature Cell Biology is committed to improving transparency in authorship. As part of our efforts in this direction, we are now requesting that all authors identified as 'corresponding author' on published papers create and link their Open Researcher and Contributor Identifier (ORCID) with their account on the Manuscript Tracking System (MTS), prior to acceptance. ORCID helps the scientific community achieve unambiguous attribution of all scholarly contributions. You can create and link your ORCID from the home page of the MTS by clicking on 'Modify my Springer Nature account'. For more information please visit www.springernature.com/orcid.

This journal strongly supports public availability of data. Please place the data used in your paper into a public data repository, or alternatively, present the data as Supplementary Information. If data can only be shared on request, please explain why in your Data Availability Statement, and also in the correspondence with your editor. Please note that for some data types, deposition in a public repository is mandatory - more information on our data deposition policies and available repositories appears below.

[Redacted]

We would like to receive the revision within four weeks. If submitted within this time period, reconsideration of the revised manuscript will not be affected by related studies published elsewhere, or accepted for publication in Nature Cell Biology in the meantime. We would be happy to consider a revision even after this timeframe, but in that case we will consider the published literature at the time of resubmission when assessing the file.

We hope that you will find our referees' comments, and editorial guidance helpful. Please do not hesitate to contact me if there is anything you would like to discuss.

Best wishes,

Stelios

Stylios Lefkopoulos, PhD
He/him/his
Senior Editor, Nature Cell Biology
Springer Nature
Heidelberger Platz 3, 14197 Berlin, Germany

E-mail: stylios.lefkopoulos@springernature.com
Twitter: [@s_lefkopoulos](https://twitter.com/s_lefkopoulos)
LinkedIn: [linkedin.com/in/stylios-lefkopoulos-81b007a0](https://www.linkedin.com/in/stylios-lefkopoulos-81b007a0)

Reviewers' Comments:

Reviewer #1:

Remarks to the Author:

In this revised paper, authors have adequately addressed my previous concerns and strengthened the data. Now the paper is acceptable.

Reviewer #2:

Remarks to the Author:

Overall, the manuscript revisions do address the vast majority of issues raised in response to the initial submission. In particular, the further data provided on Dach1 loss-of-function does overall provide a more compelling argument for the relevance of the type R endothelial cells to bone metabolism.

The one remaining concern is that dynamic histomorphometry is the standard method to show alterations in bone formation, and the data provided in Figs 7f, 6g,h and ext data Fig 6f, i or Ext Data Fig 7a,b appear to not include the requested dynamic histomorphometry demonstrating the rate of bone formation or measures of bone resorption activity (serum markers of bone resorption) in any of the key Dach1 gain-of-function or loss-of-function models. Providing data directly showing alterations in bone formation and resorption in one of the Dach1 models is seen as central for demonstrating an impact of type R capillaries on bone metabolism. Also reporting of all histomorphometry data should use the standard nomenclature established in Dempster et al. JBMR 2012.

Other issues have been adequately addressed.

Reviewer #3:

Remarks to the Author:

This manuscript makes an important contribution to the literature, but one topic continues to need greater clarification: the increase in type R capillaries in the setting of alendronate treatment. The title of this paper is, "Specialized post-arterial capillaries facilitate adult bone remodeling". Remodeling is the process in which bone is digested by osteoclast and then replaced by the action of osteoblasts. Alendronate dramatically decreases bone remodeling by decreasing the number and activity of osteoblasts, while, through blocking osteoclast action, also decreasing bone resorption. Yet type R capillaries are increased after alendronate administration. Since there are no real mechanistic studies in this paper, it is not clear why these R type capillaries increase in this setting of decreased remodeling. The cause of the increase in type R capillaries is NOT an increase in bone remodeling. The authors point out that alendronate increases bone mass and connectivity, properties associated with the increase in the organic and inorganic matrix materials. That increase in matrix probably makes oxygenation of the

bone cells more challenging. My guess is that the R type capillaries in this setting are fulfilling the need for more oxygenation of the dispersed bone cells, despite the dramatic decrease in bone remodeling. The authors should clarify the specific role of R type capillaries in this setting, particularly if they insist on their misleading title of the paper.

GUIDELINES FOR SUBMISSION OF NATURE CELL BIOLOGY ARTICLES

ARTICLE FORMAT

ABSTRACT – should not exceed 150 words and should be unreferenced. This paragraph is the most visible part of the paper and should briefly outline the background and rationale for the work, and accurately summarize the main results and conclusions. Key genes, proteins and organisms should be specified to ensure discoverability of the paper in online searches.

TEXT – the main text consists of the Introduction, Results, and Discussion sections and must not exceed 3500 words including the abstract. The Introduction should expand on the background relating to the work. The Results should be divided in subsections with subheadings, and should provide a concise and accurate description of the experimental findings. The Discussion should expand on the findings and their implications. All relevant primary literature should be cited, in particular when discussing the background and specific findings.

REFERENCES – are limited to a total of 70 in the main text and Methods combined. They must be numbered sequentially as they appear in the main text, tables and figure legends and Methods and must follow the precise style of Nature Cell Biology references. References only cited in the Methods should be numbered consecutively following the last reference cited in the main text. References only associated with Supplementary Information (e.g. in supplementary legends) do not count toward the total reference limit and do not need to be cited in numerical continuity with references in the main text. Only published papers can be cited, and each publication cited should be included in the numbered reference list, which should include the manuscript titles. Footnotes are not permitted.

Methods should be written concisely, but should contain all elements necessary to allow interpretation and replication of the results. As a guideline, Methods sections typically do not exceed 3,000 words. The Methods should be divided into subsections listing reagents and techniques. When citing previous

methods, accurate references should be provided and any alterations should be noted. Information must be provided about: antibody dilutions, company names, catalogue numbers and clone numbers for monoclonal antibodies; sequences of RNAi and cDNA probes/primers or company names and catalogue numbers if reagents are commercial; cell line names, sources and information on cell line identity and authentication. Animal studies and experiments involving human subjects must be reported in detail, identifying the committees approving the protocols. For studies involving human subjects/samples, a statement must be included confirming that informed consent was obtained. Statistical analyses and information on the reproducibility of experimental results should be provided in a section titled "Statistics and Reproducibility".

All Nature Cell Biology manuscripts submitted on or after March 21 2016, must include a Data availability statement as a separate section after Methods but before references, under the heading "Data Availability". For Springer Nature policies on data availability see <http://www.nature.com/authors/policies/availability.html>; for more information on this particular policy see <http://www.nature.com/authors/policies/data/data-availability-statements-data-citations.pdf>. The Data availability statement should include:

- Accession codes for primary datasets (generated during the study under consideration and designated as "primary accessions") and secondary datasets (published datasets reanalysed during the study under consideration, designated as "referenced accessions"). For primary accessions data should be made public to coincide with publication of the manuscript. A list of data types for which submission to community-endorsed public repositories is mandated (including sequence, structure, microarray, deep sequencing data) can be found here <http://www.nature.com/authors/policies/availability.html#data>.
- Unique identifiers (accession codes, DOIs or other unique persistent identifier) and hyperlinks for datasets deposited in an approved repository, but for which data deposition is not mandated (see here for details <http://www.nature.com/sdata/data-policies/repositories>).
- At a minimum, please include a statement confirming that all relevant data are available from the authors, and/or are included with the manuscript (e.g. as source data or supplementary information), listing which data are included (e.g. by figure panels and data types) and mentioning any restrictions on availability.
- If a dataset has a Digital Object Identifier (DOI) as its unique identifier, we strongly encourage including this in the Reference list and citing the dataset in the Methods.

We recommend that you upload the step-by-step protocols used in this manuscript to the Protocol Exchange. More details can found at www.nature.com/protocolexchange/about.

DISPLAY ITEMS – main display items are limited to 6-8 main figures and/or main tables. For Supplementary Information see below.

FIGURES – Colour figure publication costs \$395 per colour figure. All panels of a multi-panel figure must be logically connected and arranged as they would appear in the final version. Unnecessary figures and figure panels should be avoided (e.g. data presented in small tables could be stated briefly in the text instead).

All imaging data should be accompanied by scale bars, which should be defined in the legend. Cropped images of gels/blots are acceptable, but need to be accompanied by size markers, and to retain visible background signal within the linear range (i.e. should not be saturated). The boundaries of panels with low background have to be demarked with black lines. Splicing of panels should only be considered if unavoidable, and must be clearly marked on the figure, and noted in the legend with a statement on whether the samples were obtained and processed simultaneously. Quantitative comparisons between samples on different gels/blots are discouraged; if this is unavoidable, it has to be performed for samples derived from the same experiment with gels/blots were processed in parallel, which needs to be stated in the legend.

Regardless of format, all figures must be vector graphic compatible files, not supplied in a flattened raster/bitmap graphics format, but should be fully editable, allowing us to highlight/copy/paste all text and move individual parts of the figures (i.e. arrows, lines, x and y axes, graphs, tick marks, scale bars etc). The only parts of the figure that should be in pixel raster/bitmap format are photographic images or 3D rendered graphics/complex technical illustrations.

Unprocessed scans of all key data generated through electrophoretic separation techniques need to be presented in a supplementary figure that should be labeled and numbered as the final supplementary figure, and should be mentioned in every relevant figure legend. This figure does not count towards the total number of figures and is the only figure that can be displayed over multiple pages, but should be provided as a single file, in PDF or TIFF format. Data in this figure can be displayed in a relatively informal style, but size markers and the figures panels corresponding to the presented data must be indicated.

The total number of Supplementary Figures (not including the “unprocessed scans” Supplementary Figure) should not exceed the number of main display items (figures and/or tables (see our Guide to Authors and March 2012 editorial <http://www.nature.com/ncb/authors/submit/index.html#suppinfo>; <http://www.nature.com/ncb/journal/v14/n3/index.html#ed>). No restrictions apply to Supplementary Tables or Videos, but we advise authors to be selective in including supplemental data.

GUIDELINES FOR EXPERIMENTAL AND STATISTICAL REPORTING

REPORTING REQUIREMENTS – We ask authors to complete a Reporting Summary that collects information on experimental design and reagents. We hope this will aid in your evaluation of the paper. The Reporting Summary can be found here <https://www.nature.com/documents/nr-reporting-summary.pdf>) Please note that these forms are dynamic ‘smart pdfs’ and must therefore be downloaded and completed in Adobe Reader. We will then flatten them for ease of use. If you would like to reference the guidance text as you complete the template, please access these flattened versions at <http://www.nature.com/authors/policies/availability.html>.

STATISTICS – Wherever statistics have been derived the legend needs to provide the n number (i.e. the sample size used to derive statistics) as a precise value (not a range), and define what this value represents. Error bars need to be defined in the legends (e.g. SD, SEM) together with a measure of centre (e.g. mean, median). Box plots need to be defined in terms of minima, maxima, centre, and

percentiles. Ranges are more appropriate than standard errors for small data sets. Wherever statistical significance has been derived, precise p values need to be provided and the statistical test used needs to be stated in the legend. Statistics such as error bars must not be derived from $n < 3$. For sample sizes of $n < 5$ please plot the individual data points rather than providing bar graphs. Deriving statistics from technical replicate samples, rather than biological replicates is strongly discouraged. Wherever statistical significance has been derived, precise p values need to be provided and the statistical test stated in the legend.

Author Rebuttal, first revision:

We would like to thank all reviewers for this positive assessment of our revision and the additional suggestions. Point-to-point responses to the remaining questions are provided below.

Point-by-point response to reviewers' comments:

Reviewer #1: Remarks to the Author:

In this revised paper, authors have adequately addressed my previous concerns and strengthened the data. Now the paper is acceptable.

We sincerely appreciate the reviewer's positive comments and valuable suggestions, which have significantly enhanced our manuscript.

Reviewer #2: Remarks to the Author:

Overall, the manuscript revisions do address the vast majority of issues raised in response to the initial submission. In particular, the further data provided on Dach1 loss-of-function does overall provide a more compelling argument for the relevance of the type R endothelial cells to bone metabolism.

The one remaining concern is that dynamic histomorphometry is the standard method to show alterations in bone formation, and the data provided in Figs 7f, 6g,h and ext data Fig 6f, i or Ext Data Fig 7a,b appear to not include the requested dynamic histomorphometry demonstrating the rate of bone formation or measures of bone resorption activity (serum markers of bone resorption) in any of the key Dach1 gain-of-function or loss-of-function models. Providing data directly showing alterations in bone formation and resorption in one of the Dach1 models is seen as central for demonstrating an impact of type R capillaries on bone metabolism. Also reporting of all histomorphometry data should use the standard nomenclature established in Dempster et al. JBMR 2012.

Other issues have been adequately addressed.

Thank you very much for your insightful comments and constructive feedback on our revised manuscript. We are very grateful that the important role of type R capillaries in bone remodeling is recognized. We also agree with the importance of dynamic histomorphometry to demonstrate changes in bone formation. To address this issue, we have now included new data showing double-fluorescence bone labelling by consecutive injection of Calcein Green and Alizarin Red. Calculation of bone formation rates shows that mineral apposition rate (MAR) and bone formation rate (BFR/BS) are strongly increased in endothelial cell-specific Dach1 gain-of-function trabecular bone compared to littermate controls, whereas MAR remains unchanged for compact bone (see below and **Extended Data Fig.7d-f**). In these MAR calculations, we addressed missing Calcein Green labeling in control trabecular bone by imputing 0.1 $\mu\text{m}/\text{day}$ as a minimum value, as recommended by the ASBMR Histomorphometry Nomenclature Committee (Dempster et al. 2013, PMID: 23197339) and mentioned in the revised Methods. In addition, we have ensured that all histomorphometry data are reported using the standard nomenclature, as recommended by the reviewer.

Reviewer #3: Remarks to the Author:

This manuscript makes an important contribution to the literature, but one topic continues to need greater clarification: the increase in type R capillaries in the setting of alendronate treatment. The title of this paper is, “Specialized post-arterial capillaries facilitate adult bone remodeling”. Remodeling is the process in which bone is digested by osteoclast and then replaced by the action of osteoblasts. Alendronate dramatically decreases bone remodeling by decreasing the number and activity of osteoblasts, while, through blocking osteoclast action, also decreasing bone resorption. Yet type R capillaries are increased after alendronate administration. Since there are no real mechanistic studies in this paper, it is not clear why these R type capillaries increase in this setting of decreased remodeling. The cause of the increase in type R capillaries is NOT an increase in bone remodeling. The authors point out that alendronate increases bone mass and connectivity, properties associated with the increase in the organic and inorganic matrix materials. That increase in matrix probably makes oxygenation of the bone cells more challenging. My guess is that the R type capillaries in this setting are fulfilling the need for more oxygenation of the dispersed bone cells, despite the dramatic decrease in bone remodeling. The authors should clarify the specific role of R type capillaries in this setting, particularly if they insist on their misleading title of the paper.

Thank you very much for this feedback. We greatly appreciate the comment regarding the role of type R capillaries in the context of Alendronate treatment. The reviewer is absolutely correct in noting that Alendronate blocks bone remodeling by inhibiting osteoclast activity, leading to increased bone mass. Our findings in this context are likely to reflect that the metabolic need associated with the Alendronate-induced increase in trabecular bone requires additional blood supply provided by arteries and post-arterial type R capillaries. In addition, it is possible that

compensatory mechanisms try to restore bone turnover in response to inhibited bone resorption.

As recommended by the reviewer, we have clarified this important issue in the revised manuscript. At the end of the last paragraph on page 14, we write “This result suggests that type R capillaries respond to an increase in trabecular bone even when osteoclast-mediated resorption and thereby bone turnover are pharmacologically blocked.” On page 18 in the Discussion, we state: “Increased oxygen demand and elevated metabolic needs might also explain the higher abundance of type R vessels in response to pharmacological inhibition of osteoclast activity, a condition that increases bone mass but suppresses bone turnover.”

We trust that these statements have addressed the reviewer’s concern. We also appreciate that future work, stimulated by our pioneering study, will have to provide further insight into the induction and molecular regulation of type R vessels in different physiological and pathological contexts.

Decision Letter, second revision:

30th July 2024

Dear Dr. Adams,

Thank you for submitting your revised manuscript "Specialized post-arterial capillaries facilitate adult bone remodeling" (NCB-A51946B). It has now been seen by the original referees #2 and #3 and their comments are below. The reviewers find that the paper has improved in revision, and therefore we'll be happy in principle to publish it in Nature Cell Biology, pending minor revisions to comply with our editorial and formatting guidelines.

If the current version of your manuscript is in a PDF format, please email us a copy of the file in an editable format (Microsoft Word or LaTeX)-- we cannot proceed with PDFs at this stage.

We are now performing detailed checks on your paper and will send you a checklist detailing our editorial and formatting requirements in about 2 weeks. Please do not upload the final materials and make any revisions until you receive this additional information from us.

Thank you again for your interest. in Nature Cell Biology Please do not hesitate to contact me if you have any questions.

Best wishes,
Stelios

Stylianos Lefkopoulos, PhD
He/him/his
Senior Editor, Nature Cell Biology
Springer Nature
Heidelberger Platz 3, 14197 Berlin, Germany

E-mail: stylianos.lefkopoulos@springernature.com
Twitter: @s_lefkopoulos
LinkedIn: [linkedin.com/in/stylianos-lefkopoulos-81b007a0](https://www.linkedin.com/in/stylianos-lefkopoulos-81b007a0)

Reviewer #2 (Remarks to the Author):

The authors have now adequately addressed all comments on the manuscript.

Reviewer #3 (Remarks to the Author):

The authors have now addressed my concerns.

Decision Letter, final checks:

Our ref: NCB-A51946B

21st August 2024

Dear Dr. Adams,

Thank you for your patience as we've prepared the guidelines for final submission of your Nature Cell Biology manuscript, "Specialized post-arterial capillaries facilitate adult bone remodeling" (NCB-A51946B). Please carefully follow the step-by-step instructions provided in the attached file, and add a response in each row of the table to indicate the changes that you have made. Ensuring that each point is addressed will help to ensure that your revised manuscript can be swiftly handed over to our production team.

In recognition of the time and expertise our reviewers provide to Nature Cell Biology's editorial process, we would like to formally acknowledge their contribution to the external peer review of your manuscript entitled "Specialized post-arterial capillaries facilitate adult bone remodeling". For those reviewers who give their assent, we will be publishing their names alongside the published article.

Nature Cell Biology offers a Transparent Peer Review option for new original research manuscripts submitted after December 1st, 2019. As part of this initiative, we encourage our authors to support increased transparency into the peer review process by agreeing to have the reviewer comments, author rebuttal letters, and editorial decision letters published as a Supplementary item. When you submit your final files please clearly state in your cover letter whether or not you would like to participate in this initiative. Please note that failure to state your preference will result in delays in accepting your manuscript for publication.

Cover suggestions

COVER ARTWORK: We welcome submissions of artwork for consideration for our cover. For more information, please see our guide for cover artwork.

Nature Cell Biology has now transitioned to a unified Rights Collection system which will allow our Author Services team to quickly and easily collect the rights and permissions required to publish your work. Approximately 10 days after your paper is formally accepted, you will receive an email in providing you with a link to complete the grant of rights. If your paper is eligible for Open Access, our Author Services team will also be in touch regarding any additional information that may be required to arrange payment for your article.

Please note that *Nature Cell Biology* is a Transformative Journal (TJ). Authors may publish their research with us through the traditional subscription access route or make their paper immediately open access through payment of an article-processing charge (APC). Authors will not be required to make a final decision about access to their article until it has been accepted. Find out more about Transformative Journals

[Redacted]

Best regards,

Kendra Donahue
Staff
Nature Cell Biology

On behalf of

Stylianos Lefkopoulos, PhD
He/him/his
Senior Editor, Nature Cell Biology
Springer Nature
Heidelberger Platz 3, 14197 Berlin, Germany

E-mail: stylianos.lefkopoulos@springernature.com
Twitter: @s_lefkopoulos
LinkedIn: [linkedin.com/in/stylianos-lefkopoulos-81b007a0](https://www.linkedin.com/in/stylianos-lefkopoulos-81b007a0)

Reviewer #2:

Remarks to the Author:

The authors have now adequately addressed all comments on the manuscript.

Reviewer #3:

Remarks to the Author:

The authors have now addressed my concerns.

Final Decision Letter:

Dear Dr Adams,

I am pleased to inform you that your manuscript, "Specialized post-arterial capillaries facilitate adult

bone remodeling", has now been accepted for publication in Nature Cell Biology. Congratulations!

You may wish to make your media relations office aware of your accepted publication, in case they consider it appropriate to organize some internal or external publicity. Once your paper has been scheduled you will receive an email confirming the publication details. This is normally 3-4 working days in advance of publication. If you need additional notice of the date and time of publication, please let the production team know when you receive the proof of your article to ensure there is sufficient time

to coordinate. Further information on our embargo policies can be found here:
<https://www.nature.com/authors/policies/embargo.html>

Please note that *Nature Cell Biology* is a Transformative Journal (TJ). Authors may publish their research with us through the traditional subscription access route or make their paper immediately open access through payment of an article-processing charge (APC). Authors will not be required to make a final decision about access to their article until it has been accepted. Find out more about Transformative Journals

If you have not already done so, we strongly recommend that you upload the step-by-step protocols used in this manuscript to protocols.io (<https://protocols.io>), an open online resource that allows researchers to share their detailed experimental know-how. All uploaded protocols are made freely available and are assigned DOIs for ease of citation. Protocols and Nature Portfolio journal papers in which they are used can be linked to one another, and this link is clearly and prominently visible in the online versions of both. Authors who performed the specific experiments can act as primary authors for the Protocol as they will be best placed to share the methodology details, but the Corresponding Author of the present research paper should be included as one of the authors. By uploading your Protocols onto protocols.io, you are enabling researchers to more readily reproduce or adapt the methodology you use, as well as increasing the visibility of your protocols and papers. You can also establish a dedicated workspace to collect your lab Protocols. Further information can be found at

<https://www.protocols.io/help/publish-articles>.

With kind regards,
Stelios

Stylianos Lefkopoulos, PhD
He/him/his
Senior Editor, Nature Cell Biology
Springer Nature
Heidelberger Platz 3, 14197 Berlin, Germany

E-mail: stylianos.lefkopoulos@springernature.com
Twitter: @s_lefkopoulos
LinkedIn: [linkedin.com/in/stylianos-lefkopoulos-81b007a0](https://www.linkedin.com/in/stylianos-lefkopoulos-81b007a0)
